# Adaptive SGD with Polyak stepsize and Line-search: Robust Convergence and Variance Reduction

**Xiaowen Jiang**
CISPA*
xiaowen.jiang@cispa.de

**Sebastian U. Stich**
CISPA*
stich@cispa.de

## Abstract

The recently proposed stochastic Polyak stepsize (SPS) and stochastic line-search (SLS) for SGD have shown remarkable effectiveness when training over-parameterized models. However, two issues remain unsolved in this line of work.

First, in non-interpolation settings, both algorithms only guarantee convergence to a neighborhood of a solution which may result in a worse output than the initial guess. While artificially decreasing the adaptive stepsize has been proposed to address this issue (Orvieto et al. [44]), this approach results in slower convergence rates under interpolation. Second, intuitive line-search methods equipped with variance-reduction (VR) fail to converge (Dubois-Taine et al. [14]). So far, no VR methods successfully accelerate these two stepsizes with a convergence guarantee.

In this work, we make two contributions: Firstly, we propose two new robust variants of SPS and SLS, called AdaSPS and AdaSLS, which achieve optimal asymptotic rates in both strongly-convex or convex and interpolation or non-interpolation settings, except for the case when we have both strong convexity and non-interpolation. AdaSLS requires no knowledge of problem-dependent parameters, and AdaSPS requires only a lower bound of the optimal function value as input. Secondly, we propose a novel VR method that can use Polyak stepsizes or line-search to achieve acceleration. When it is equipped with AdaSPS or AdaSLS, the resulting algorithms obtain the optimal rate for optimizing convex smooth functions. Finally, numerical experiments on synthetic and real datasets validate our theory and demonstrate the effectiveness and robustness of our algorithms.

## 1 Introduction

Stochastic Gradient Descent (SGD) [46] and its variants [7] are among the most preferred algorithms for training modern machine learning models. These methods only compute stochastic gradients in each iteration, which is often more efficient than computing a full batch gradient. However, the performance of SGD is highly sensitive to the choice of the stepsize. Common strategies use a fixed stepsize schedule, such as keeping it constant or decreasing it over time. Unfortunately, the theoretically optimal schedules are disparate across different function classes [8], and usually depend on problem parameters that are often unavailable, such as the Lipschitz constant of the gradient. As a result, a heavy tuning of the stepsize parameter is required, which is typically expensive in practice.

Instead of fixing the stepsize schedule, adaptive SGD methods adjust the stepsize on the fly [15, 26]. These algorithms often require less hyper-parameter tuning and still enjoy competitive performance in practice. Stochastic Polyak Stepsize (SPS) [34, 6, 45] is one of such recent advances. It has received rapid growing interest due to two factors: (i) the only required parameter is the individual optimal function value which is often available in many machine learning applications and (ii) its adaptivity

---

*CISPA Helmholtz Center for Information Security, Saarbrücken, Germany

37th Conference on Neural Information Processing Systems (NeurIPS 2023).

utilizes the local curvature and smoothness information allowing the algorithm to accelerate and converge quickly when training over-parametrized models. Stochastic Line-Search (SLS) [52] is another adaptive stepsize that offers exceptional performance when the interpolation condition holds. In contrast to SPS, the knowledge of the optimal function value is not required for SLS, at the cost of additional function value evaluations per iteration.

An ideal adaptive stepsize should not only require fewer hyper-parameters but should also enjoy *robust* convergence, in the sense that they can automatically adapt to the optimization setting (interpolation vs. non-interpolation). This will bring great convenience to the users in practice as they no longer need to choose which method to use (or runinning both of them at double the cost). Indeed, in many real-world scenarios, it can be challenging to ascertain whether a model is effectively interpolating the data or not [11]. For instance, the feature dimension of the rcv1 dataset [10] is twice larger than the number of data points. A logistic regression model with as many parameters as the feature dimension may tend to overfit the data points. But the features are actually sparse and the model is not interpolated. Another example is federated learning [23] where millions of clients jointly train a machine learning model on their mobile devices, which usually cannot support huge-scale models. Due to the fact that each client's data is stored locally, it becomes impractical to check the interpolation condition.

While SPS and SLS are promising adaptive methods, they are not robust since both methods cannot converge to the solution when interpolation does not hold. Orvieto et al. [44] address this issue for SPS by applying an artificially decreasing rule and the resulting algorithm DecSPS is able to converge as quickly as SGD with the optimal stepsize schedule. However, the convergence rates of DecSPS in interpolation regimes are much slower than SPS. For SLS, no solution has been proposed.

If the user is certain that the underlying problem is non-interpolated, then applying variance-reduction (VR) techniques can further accelerate the convergence [22, 40, 13, 27, 49, 33]. While gradient descent with Polayak stepsize and line-search perform well in the deterministic settings, there exists no method that successfully adapt these stepsizes in VR methods. Mairal [36] and Schmidt et al. [49] proposed to use stochastic line-search with VR. However, no theoretical guarantee is shown. Indeed, this is a challenging open question as Dubois-Taine et al. [14] provides a counter-example where classical line-search methods fail in the VR setting. As such, it remains unclear whether we can accelerate SGD with stepsizes from Polyak and line-search family in non-interpolated settings.

## 1.1 Main contributions

In this work, we provide solutions to the aforementioned two challenges and contribute new theoretical insights on Polyak stepsize and line-search methods. We summarize our main contributions as follows:

- In Section 3, we propose the first robust adaptive methods that simultaneously achieve the best-known asymptotic rates in both strongly-convex or convex and interpolation or non-interpolation settings except for the case when we have strongly-convexity and non-interpolation. The first method called AdaSPS, a variant of SPS, requires only a lower bound of the optimal function value as input (similar to DecSPS) while AdaSLS, the second method based on SLS, is parameter-free. In the non-interpolated setting, we prove for both algorithms an $\mathcal{O}(1/\varepsilon^2)$ convergence rate for convex functions which matches the classical DecSPS and AdaGrad [15] results, whereas SPS and SLS cannot converge in this case. In the interpolated regime, we establish fast $\mathcal{O}(\log(1/\varepsilon))$ and $\mathcal{O}(1/\varepsilon)$ rates under strong convexity and convexity conditions respectively, without knowledge of any problem-dependent parameters. In contrast, DecSPS converges at the slower $\mathcal{O}(1/\varepsilon^2)$ rate and for AdaGrad, the Lipschitz constant is needed to set its stepsize [56].

- In Section 4, we design a new variance-reduction method that is applicable to both Polyak stepsizes or line-search methods. We prove that to reach an $\varepsilon$-accuracy, the total number of gradient evaluations required in expectation is $\widetilde{\mathcal{O}}(n + 1/\varepsilon)$ for convex functions which matches the rate of AdaSVRG [14]. With our newly proposed decreasing probability strategy, the artificially designed multi-stage inner-outer-loop structure is not needed, which makes our methods easier to analyze.

  Our novel VR-framework is based on proxy function sequences and can recover the standard VR methods [22] as a special case. We believe that this technique can be of independent interest to the optimization community and may motivate more personalized VR techniques in the future.

| Stepsize | Interpolation | | | Non-interpolation | | |
|---|---|---|---|---|---|---|
| | strongly-convex | convex | required input | strongly-convex | convex$^a$ | required input |
| SPS/SPS$_{\max}$ [34] | $\mathcal{O}(\log(\frac{1}{\varepsilon}))$ | $\mathcal{O}(\frac{1}{\varepsilon})$ | $f^\star_{i_t}$ | $\varepsilon \geq \Omega(\sigma^2_{f,B})$ | $\varepsilon \geq \Omega(\sigma^2_{f,B})$ | $f^\star_{i_t}$ |
| SLS [52] | $\mathcal{O}(\log(\frac{1}{\varepsilon}))$ | $\mathcal{O}(\frac{1}{\varepsilon})$ | None | $\varepsilon \geq \Omega(\sigma^2_{f,B})$ | $\varepsilon \geq \Omega(\sigma^2_{f,B})$ | None |
| DecSPS [44] | $\mathcal{O}(\frac{1}{\varepsilon^2})$ | $\mathcal{O}(\frac{1}{\varepsilon^2})$ | $\ell^\star_{i_t}$ | $\mathcal{O}(\frac{1}{\varepsilon^2})$ | $\mathcal{O}(\frac{1}{\varepsilon^2})$ | $\ell^\star_{i_t}$ |
| AdaSPS (this work) | $\mathcal{O}(\log(\frac{1}{\varepsilon}))$ | $\mathcal{O}(\frac{1}{\varepsilon})$ | $f^\star_{i_t}$ | $\mathcal{O}(\frac{1}{\varepsilon^2})$ | $\mathcal{O}(\frac{1}{\varepsilon^2})$ | $\ell^\star_{i_t}$ |
| AdaSLS (this work) | $\mathcal{O}(\log(\frac{1}{\varepsilon}))$ | $\mathcal{O}(\frac{1}{\varepsilon})$ | None | $\mathcal{O}(\frac{1}{\varepsilon^2})$ | $\mathcal{O}(\frac{1}{\varepsilon^2})$ | None |

$^a$The assumption of bounded iterates is also required except for SPS and SLS.

Table 1: Summary of convergence behaviors of the considered adaptive stepsizes for smooth functions. For SPS/SPS$_{\max}$ and SLS in non-interpolation settings, $\Omega(\cdot)$ indicates the size of the neighborhood that they can converge to. In the other cases, the $\mathcal{O}(\cdot)$ complexity provides the total number of gradient evaluations required for each algorithm to reach an $\mathcal{O}(\varepsilon)$ suboptimality. For convex functions, the suboptimality is defined as $\mathbb{E}[f(\bar{\mathbf{x}}_T) - f^\star]$ and for strongly convex functions, the suboptimality is defined as $\mathbb{E}[||\mathbf{x}_T - \mathbf{x}^\star||^2]$.

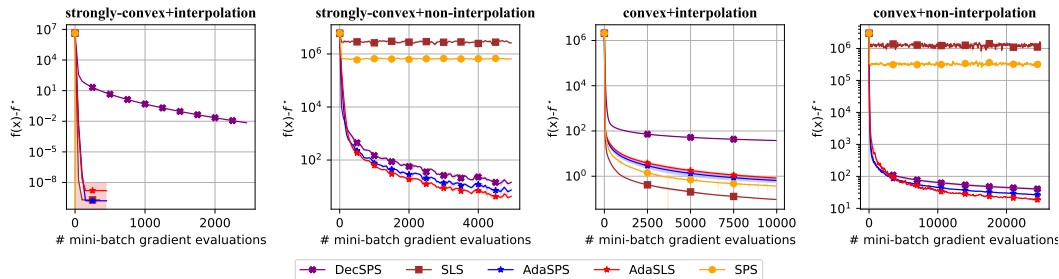

Figure 1: Illustration of the robust convergence of AdaSPS and AdaSLS on synthetic data with quadratic loss. SPS and SLS have superior performance on the two interpolated problems but cannot converge when the interpolation condition does not hold. DecSPS suffers from a slow convergence on both interpolated problems. (Repeated 3 times. The solid lines and the shaded area represent the mean and the standard deviation.)

## 1.2 Related work

Line-search procedures has been successfully applied to accelerate large-scale machine learning training. Following [52], Galli et al. [16] propose to relax the condition of monotone decrease of objective function for training over-parameterized models. Kunstner et al. [30] extends backtracking line-search to a multidimensional variant which provides better diagonal preconditioners. In recent years, adaptive stepsizes from the AdaGrad family have become widespread and are particularly successful when training deep neural networks. Plenty of contributions have been made to analyze variants of AdaGrad for different classes of functions [15, 51, 43, 55, 56], among which Vaswani et al. [53] first propose to use line-search to set the stepsize for AdaGrad to enhance its practical performance. More recently, variance reduction has successfully been applied to AdaGrad stepsize and faster convergence rates have been established for convex and non-convex functions [14, 25].

Another promising direction is the Polyak stepsize (PS) [45] originally designed as a subgradient method for solving non-smooth convex problems. Hazan and Kakade [20] show that PS indeed gives simultaneously the optimal convergence result for a more general class of convex functions. Nedić and Bertsekas [38] propose several variants of PS as incremental subgradient methods and they also discuss the method of dynamic estimation of the optimal function value when it is not known. Recently, more effort has been put into extending deterministic PS to the stochastic setting [47, 6, 42]. However, theoretical guarantees of the algorithms still remain elusive until the emergence of SPS/SPS$_{\max}$ [34]. Subsequently, further improvements and new variants such as DecSPS [44] and SPS with a moving target [17] have been introduced. A more recent line of work interprets stochastic Polyak stepsize as a subsampled Newton Raphson method and interesting algorithms have been designed based on the first-order local expansion [17, 18] as well as the second-order expansion [31]. Wang et al. [54] propose to set the stepsize for SGD with momentum using Polyak stepsize. Abdukhakimov et al. [1] employ more general preconditioning techniques to SPS.

There has been a recent line of work attempting to develop methods that can adapt to both the interpolation setting and the general growth condition beyond strong convexity. Using the iterative halving technique from [5], Cheng and Duchi [11] propose AdaStar-G which gives the desired property if the Lipschitz constant and the diameter of the parameter domain are known. How to remove these requirements is an interesting open question for the future research.

## 2 Problem setup and background

### 2.1 Notations

In this work, we consider solving the finite-sum smooth convex optimization problem:

$$\min_{\mathbf{x} \in \mathbb{R}^d} \left[ f(\mathbf{x}) = \frac{1}{n} \sum_{i=1}^{n} f_i(\mathbf{x}) \right] . \tag{1}$$

This type of problem appears frequently in the modern machine learning applications [19], where each $f_i(\mathbf{x})$ represents the loss of a model on the $i$-th data point parametrized by the parameter $\mathbf{x}$. Stochastic Gradient Descent (SGD) [46] is one of the most popular methods for solving the problem (1). At each iteration, SGD takes the form:

$$\mathbf{x}_{t+1} = \mathbf{x}_t - \eta_t \nabla f_{i_t}(\mathbf{x}_t) , \tag{2}$$

where $\eta_t$ is the stepsize parameter, $i_t \subseteq [n]$ is a random set of size $B$ sampled independently at each iteration $t$ and $\nabla f_{i_t}(\mathbf{x}) = \frac{1}{B} \sum_{i \in i_t} \nabla f_i(\mathbf{x})$ is the minibatch gradient.

Throughout the paper, we assume that there exists a non-empty set of optimal points $\mathcal{X}^\star \subset \mathbb{R}^d$, and we use $f^\star$ to denote the optimal value of $f$ at a point $\mathbf{x}^\star \in \mathcal{X}^\star$. We use $f_{i_t}^\star$ to denote the infimum of minibatch function $f_{i_t}(x)$, i.e. $f_{i_t}^\star = \inf_{\mathbf{x} \in \mathbb{R}^d} \frac{1}{B} \sum_{i \in i_t} f_i(\mathbf{x})$. We assume that all the individual functions $\{f_i(\mathbf{x})\}$ are $L$-smooth. Finally, we denote the optimal objective difference, first introduced in [34], by $\sigma_{f,B}^2 = f^\star - \mathbb{E}_{i_t}[f_{i_t}^\star]$. The definitions for the interpolation condition can be defined and studied in various ways [48, 9, 4, 11]. Here, we adopt the notion from [34]. The problem (1) is said to be **interpolated** if $\sigma_{f,1}^2 = 0$, which implies that $\sigma_{f,B}^2 = 0$ for all $B \leq n$ since $\sigma_{f,B}^2$ is non-increasing w.r.t $B$. Note interpolation implies that the global minimizer of $f$ is also a minimizer of each individual function $f_i$.

### 2.2 SGD with stochastic polyak stepsize

Loizou et al. [34] propose to set the stepsize $\eta_t$ as: $\eta_t = 2 \frac{f_{i_t}(\mathbf{x}_t) - f_{i_t}^\star}{\|\nabla f_{i_t}(\mathbf{x}_t)\|^2}$, which is well known as the Stochastic Polyak stepsize (SPS). In addition to SPS, they also propose a bounded variant $\text{SPS}_{\max}$ which has the form $\eta_t = \min\left\{2 \frac{f_{i_t}(\mathbf{x}_t) - f_{i_t}^\star}{\|\nabla f_{i_t}(\mathbf{x}_t)\|^2}, \gamma_b\right\}$ where $\gamma_b > 0$. Both algorithms require the input of the exact $f_{i_t}^\star$ which is often unavailable when the batch size $B > 1$ or when the interpolation condition does not hold. Orvieto et al. [44] removes the requirement for $f_{i_t}^\star$ and propose to set $\eta_t$ as: $\eta_t = \frac{1}{\sqrt{t+1}} \min\left\{ \frac{f_{i_t}(\mathbf{x}_t) - \ell_{i_t}^\star}{\|\nabla f_{i_t}(\mathbf{x}_t)\|^2}, \sqrt{t}\eta_{t-1} \right\}$ for $t \geq 1$ (DecSPS), where $\eta_0 > 0$ is a constant and $\ell_{i_t}^\star$ is an input lower bound such that $\ell_{i_t}^\star \leq f_{i_t}^\star$. In contrast to the exact optimal function value, a lower bound $\ell_{i_t}^\star$ is often available in practice, in particular for machine learning problems when the individual loss functions are non-negative. We henceforth denote the estimation error by:

$$\text{err}_{f,B}^2 := \mathbb{E}_{i_t}[f_{i_t}^\star - \ell_{i_t}^\star] . \tag{3}$$

For convex smooth functions, SPS achieves a fast convergence up to a neighborhood of size $\Omega(\sigma_{f,B}^2)$ and its variant $\text{SPS}_{\max}$ converges up to $\Omega(\sigma_{f,B}^2 \gamma_b / \alpha)$ where $\alpha = \min\{\frac{1}{L}, \gamma_b\}$. Note that the size of the neighborhood cannot be further reduced by choosing an appropriate $\gamma_b$. In contrast, DecSPS converges at the rate of $\mathcal{O}(1/\sqrt{T})$ which matches the standard result for SGD with decreasing stepsize. However, the strictly decreasing $\Theta(1/\sqrt{t})$ stepsize schedule hurts its performance in interpolated settings. For example, DecSPS has a much slower $\mathcal{O}(1/\sqrt{T})$ convergence rate compared with the fast $\mathcal{O}(\exp(-T\mu/L))$ rate of SPS when optimizing strongly-convex objectives. Therefore, both algorithms do not have the *robust* convergence property (achieving fast convergence guarantees in both interpolated and non-interpolated regimes) and we aim to fill this gap. See Figure 1 for a detailed illustration of the non-robustness of SPS and DecSPS.

# 3 Adaptive SGD with polyak stepsize and line-search

In this section, we introduce and analyze two adaptive algorithms to solve problem (1).

## 3.1 Proposed methods

**AdaSPS.** Our first stepsize is defined as the following:

$$\eta_t = \min\left\{ \frac{f_{i_t}(\mathbf{x}_t) - \ell_{i_t}^\star}{c_p ||\nabla f_{i_t}(\mathbf{x}_t)||^2} \frac{1}{\sqrt{\sum_{s=0}^t f_{i_s}(\mathbf{x}_s) - \ell_{i_s}^\star}}, \eta_{t-1}\right\}, \quad \text{with } \eta_{-1} = +\infty, \quad \text{(AdaSPS)}$$

where $\ell_{i_t}^\star$ is an input parameter that must satisfy $\ell_{i_t}^\star \leq f_{i_t}^\star$ and $c_p > 0$ is an input constant to adjust the magnitude of the stepsize (we discuss suggested choices in Section 5).

AdaSPS can be seen as an extension of DecSPS. However, unlike the strict $\Theta(1/\sqrt{t})$ decreasing rule applied in DecSPS, AdaSPS accumulates the function value difference during the optimization process which enables it to dynamically adapt to the underlying unknown interpolation settings.

**AdaSLS.** We provide another stepsize that can be applied even when a lower bound estimation is unavailable. The method is based on line-search and thus is completely parameter-free, but requires additional function value evaluations in each iteration:

$$\eta_t = \min\left\{ \frac{\gamma_t}{c_l \sqrt{\sum_{s=0}^t \gamma_s ||\nabla f_{i_s}(\mathbf{x}_s)||^2}}, \eta_{t-1}\right\}, \quad \text{with } \eta_{-1} = +\infty, \quad \text{(AdaSLS)}$$

where $c_l > 0$ is an input constant, and the scale $\gamma_t$ is obtained via stardard Armijo backtracking line-search (see Algorithm 4 for further implementation details in the Appendix D) such that the following conditions are satisfied:

$$f_{i_t}(\mathbf{x}_t - \gamma_t \nabla f_{i_t}(\mathbf{x}_t)) \leq f_{i_t}(\mathbf{x}_t) - \rho \gamma_t ||\nabla f_{i_t}(\mathbf{x}_t)||^2 \quad \text{and} \quad \gamma_t \leq \gamma_{\max}, \ 0 < \rho < 1, \quad (4)$$

for line-search parameters $\gamma_{\max}$ and $\rho$. By setting the decreasing factor $\beta \geq \frac{1}{2}$ defined in Algorithm 4, one can show that $\gamma_t \geq \min(\frac{1-\rho}{L}, \gamma_{\max})$. We give a formal proof in Lemma 16 in Appendix A.2.

**Discussion.** Our adaptation mechanism in AdaSPS/AdaSLS is reminiscent of AdaGrad type methods, in particular to AdaGrad-Norm, the scalar version of AdaGrad, that aggregates the gradient norm in the denominator and takes the form $\eta_t = \frac{c_g}{\sqrt{\sum_{s=0}^t ||\nabla f_{i_s}(\mathbf{x}_s)||^2 + b_0^2}}$ where $c_g > 0$ and $b_0^2 \geq 0$.

The primary distinction between AdaSPS and AdaSLS compared to AdaGrad-Norm is the inclusion of an additional component that captures the curvature information at each step, and not using squared gradient norms in AdaSPS. In contrast to the strict decreasing behavior of AdaGrad-Norm, AdaSPS and AdaSLS can automatically mimic a constant stepsize when navigating a flatter region.

Vaswani et al. [53] suggest using line-search to set the stepsize for AdaGrad-Norm which takes the form $\eta_t = \frac{\gamma_t}{\sqrt{\sum_{s=0}^t ||\nabla f_{i_s}(\mathbf{x}_s)||^2}}$ where $\gamma_t \leq \gamma_{t-1}$ is required for solving non-interpolated convex problems. While this stepsize is similar to AdaSLS, the scaling of the denominator gives a suboptimal convergence rate as we demonstrate in the following section.

## 3.2 Convergence rates

In this section, we present the convergence results for AdaSPS and AdaSLS. We list the helpful lemmas in Appendix A. The proofs can be found in Appendix B.

**General convex.** We denote $\mathcal{X}$ to be a convex compact set with diameter $D$ such that there exists a solution $\mathbf{x}^\star \in \mathcal{X}$ and $\sup_{\mathbf{x},\mathbf{y} \in \mathcal{X}} ||\mathbf{x} - \mathbf{y}||^2 \leq D^2$. We let $\Pi_{\mathcal{X}}$ denote the Euclidean projection onto $\mathcal{X}$. For general convex stochastic optimization, it seems inevitable that adaptive methods require the bounded iterates assumption or an additional projection step to prove convergence due to the lack of knowledge of problem-dependent parameters [12, 15]. Here, we employ the latter solution by running projected stochastic gradient descent (PSGD):

$$\mathbf{x}_{t+1} = \Pi_{\mathcal{X}}(\mathbf{x}_t - \eta_t \nabla f_{i_t}(\mathbf{x}_t)). \quad (5)$$

**Theorem 1** (General convex). *Assume that $f$ is convex, each $f_i$ is $L$-smooth and $\mathcal{X}$ is a convex compact feasible set with diameter $D$, PSGD with AdaSPS or AdaSLS converges as:*

$$
\begin{aligned}
(AdaSPS): \ \mathbb{E}[f(\bar{\mathbf{x}}_T) - f^\star] &\leq \frac{\tau_p^2}{T} + \frac{\tau_p \sqrt{\sigma_{f,B}^2 + \mathrm{err}_{f,B}^2}}{\sqrt{T}} \ , \\
(AdaSLS): \ \mathbb{E}[f(\bar{\mathbf{x}}_T) - f^\star] &\leq \frac{\tau_l^2}{T} + \frac{\tau_l \sigma_{f,B}}{\sqrt{T}} \ ,
\end{aligned}
\tag{6}
$$

*where $\bar{\mathbf{x}}_T = \frac{1}{T}\sum_{t=0}^{T-1}\mathbf{x}_t$, $\tau_p = (2c_p L D^2 + \frac{1}{c_p})$ and $\tau_l = \max\left\{\frac{L}{(1-\rho)\sqrt{\rho}}, \frac{1}{\gamma_{\max}\sqrt{\rho}}\right\} c_l D^2 + \frac{1}{c_l\sqrt{\rho}}$.*

As a consequence of Theorem 1, if $\mathrm{err}_{f,B}^2 = \sigma_{f,B}^2 = 0$, then PSGD with AdaSPS or AdaSLS converges as $\mathcal{O}(\frac{1}{T})$. Suppose $\gamma_{\max}$ is sufficiently large, then picking $c_p^\star = \frac{1}{\sqrt{2LD^2}}$ and $c_l^\star = \frac{\sqrt{1-\rho}}{\sqrt{LD^2}}$ gives a $\mathcal{O}(\frac{LD^2}{T})$ rate under the interpolation condition, which is slightly worse than $\frac{L||\mathbf{x}_0 - \mathbf{x}^\star||^2}{T}$ obtained by SPS and SLS but is better than $\mathcal{O}(\frac{LD^2}{\sqrt{T}})$ obtained by DecSPS. If otherwise $\sigma_{f,B}^2 > 0$, then AdaSPS, AdaSLS, and DecSPS converge as $\mathcal{O}(1/\sqrt{T})$ which matches the rate of Vanilla SGD with decreasing stepsize. Finally, AdaGrad-Norm gives a similar rate in both cases while AdaGrad-Norm with line-search [53] shows a suboptimal rate of $\mathcal{O}(\frac{L^3 D^4}{T} + \frac{D^2 L^{3/2}\sigma}{\sqrt{T}})$. It is worth noting that SPS, DecSPS and SLS require an additional assumption on individual convexity.

**Theorem 2** (Individual convex+interpolation). *Assume that $f$ is convex, each $f_i$ is convex and $L$-smooth, and that $\mathrm{err}_{f,B}^2 = \sigma_{f,B}^2 = 0$, by setting $c_p = \frac{c_p^{scale}}{\sqrt{f_{i_0}(\mathbf{x}_0) - f_{i_0}^\star}}$ and $c_l = \frac{c_l^{scale}}{\rho\sqrt{\gamma_0||\nabla f_{i_0}(\mathbf{x}_0)||^2}}$ with constants $c_p^{scale} \geq 1$ and $c_l^{scale} \geq 1$, then for any $T \geq 1$, SGD (no projection) with AdaSPS or AdaSLS converges as:*

$$
(AdaSPS) \quad \mathbb{E}[f(\bar{\mathbf{x}}_T) - f^\star] \leq \left(4L(c_p^{scale})^2 \, \mathbb{E}_{i_0}\left[\frac{||\mathbf{x}_0 - \mathbf{x}^\star||^2}{f_{i_0}(\mathbf{x}_0) - f^\star}\right]\right) \frac{L||\mathbf{x}_0 - \mathbf{x}^\star||^2}{T} \ ,
\tag{7}
$$

*and*

$$
(AdaSLS) \quad \mathbb{E}[f(\bar{\mathbf{x}}_T) - f^\star] \leq \left(\frac{(c_l^{scale})^2}{\rho^3 L \min^2\{\frac{1-\rho}{L}, \gamma_{\max}\}} \, \mathbb{E}_{i_0}\left[\frac{||\mathbf{x}_0 - \mathbf{x}^\star||^2}{\gamma_0||\nabla f_{i_0}(\mathbf{x}_0)||^2}\right]\right) \frac{L||\mathbf{x}_0 - \mathbf{x}^\star||^2}{T} \ .
\tag{8}
$$

*where $\bar{\mathbf{x}}_T = \frac{1}{T}\sum_{t=1}^{T}\mathbf{x}_t$.*

The result implies that the bounded iterates assumption is not needed if we have both individual convexity and interpolation by picking $c_p$ and $c_l$ to satisfy certain conditions that do not depend on unknown parameters. To our knowledge, no such result exists for stepsizes from the AdaGrad family. It is worth noting that the min operator defined in AdaSPS or AdaSLS is not necessary in the proof.

**Remark 3.** *We note that for non-interpolated problems, AdaSPS only requires the knowledge of $\ell_{i_t}^\star$ while the exact $f_{i_t}^\star$ is needed under the interpolation condition. We argue that in many standard machine learning problems, simply picking zero will suffice. For instance, $f_{i_t}^\star = 0$ for over-parameterized logistic regression and after adding a regularizer, $\ell_{i_t}^\star = 0$.*

**Strongly convex.** We now present two algorithmic behaviors of AdaSPS and AdaSLS for strongly convex functions. In particular, We show that 1) the projection step can be removed as shown in DecSPS, and 2) if the interpolation condition holds, the min operator is not needed and the asymptotic linear convergence rate is preserved. The full statement of Lemma 4 can be found in Appendix B.2.

**Lemma 4** (Bounded iterates). *Let each $f_i$ be $\mu$-strongly convex and $L$-smooth. For any $t = 0, \ldots, T$, the iterates of SGD with AdaSPS or AdaSLS satisfy: $||\mathbf{x}_t - \mathbf{x}^\star||^2 \leq D_{\max}$, for a constant $D_{\max}$ specified in the appendix in Equation (B.16).*

**Corollary 5** (Individual strongly convex). *Assume each $f_i$ is $\mu$-strongly convex and $L$-smooth, Theorem 1 holds with PSGD and $D$ replaced by SGD and $D_{\max}$ defined in Lemma 4.*

Although it has not been formally demonstrated that AdaGrad/AdaGrad-Norm can relax the assumption on bounded iterates for strongly convex functions, we believe that with a similar proof technique, this property still holds for AdaGrad/AdaGrad-Norm.

We next show that AdaSPS and AdaSLS achieve linear convergence under the interpolation condition.

**Theorem 6** (Strongly convex + individual convex + interpolation). *Consider SGD with AdaSPS (AdaSPS) or AdaSLS (AdaSLS) stepsize. Suppose that each $f_i$ is convex and $L$-smooth, $f$ is $\mu$-strongly convex and that $\sigma_{f,B}^2 = \text{err}_{f,B}^2 = 0$. If we let $c_p = \frac{c_p^{scale}}{\sqrt{f_{i_0}(\mathbf{x}_0) - f_{i_0}^\star}}$ and $c_l = \frac{c_l^{scale}}{\rho\sqrt{\gamma_0 ||\nabla f_{i_0}(\mathbf{x}_0)||^2}}$ with constants $c_p^{scale} \geq 1$ and $c_l^{scale} \geq 1$, then AdaSPS or AdaSLS converges as:*

$$(AdaSPS) \quad \mathbb{E}[||\mathbf{x}_{T+1} - \mathbf{x}^\star||^2] \leq \mathbb{E}_{i_0}\left[\left(1 - \frac{(f_{i_0}(\mathbf{x}_0) - f^\star)\mu}{(2c_p^{scale}L||\mathbf{x}_0 - \mathbf{x}^\star||)^2}\right)^T\right]||\mathbf{x}_0 - \mathbf{x}^\star||^2, \quad (9)$$

*and*

$$(AdaSLS) \quad \mathbb{E}[||\mathbf{x}_{T+1} - \mathbf{x}^\star||^2] \leq \mathbb{E}_{i_0}\left[\left(1 - \frac{\mu\rho^3 \min^2\{\frac{1-\rho}{L}, \gamma_{\max}\}\gamma_0 ||\nabla f_{i_0}(\mathbf{x}_0)||^2}{(c_l^{scale}||\mathbf{x}_0 - \mathbf{x}^\star||)^2}\right)^T\right]||\mathbf{x}_0 - \mathbf{x}^\star||^2. \quad (10)$$

The proof of Theorem 6 is presented in Appendix B.3. We now compare the above results with the other stepsizes. Under the same settings, DecSPS has a slower $\mathcal{O}(1/\sqrt{T})$ rate due to the usage of $\Theta(1/\sqrt{t})$ decay stepsize. While AdaGrad-Norm does have a linear acceleration phase when the accumulator grows large, to avoid an $\mathcal{O}(1/\varepsilon)$ slow down, the parameters of AdaGrad-Norm have to satisfy $c_g < b_0/L$, which requires the knowledge of Lipschitz constant [56]. Instead, the conditions on $c_p$ and $c_l$ for AdaSPS and AdaSLS only depend on the function value and gradient norm at $\mathbf{x}_0$ which can be computed at the first iteration. SPS, SLS, and Vannilia-SGD with constant stepsize achieve faster linear convergence rate of order $\mathcal{O}\left(\exp(-\frac{\mu}{L}T)\right)$. It is worth noting that Vannila-SGD can further remove the individual convexity assumption.

**Discussion.** In non-interpolation regimes, AdaSPS and AdaSLS only ensure a slower $\mathcal{O}(1/\sqrt{T})$ convergence rate compared with $\mathcal{O}(1/T)$ rate achieved by vanilla SGD with $\Theta(1/t)$ decay stepsize when optimizing strongly-convex functions [7]. To our knowledge, no parameter-free adaptive stepsize exists that achieves such a fast rate under the same assumptions. Therefore, developing an adaptive algorithm that can adapt to both convex and strongly-convex functions would be a significant further contribution.

## 4   AdaSPS and AdaSLS with variance-reduction

Combining variance-reduction (VR) with adaptive Polyak-stepsize and line-search to achieve acceleration is a natural idea that has been explored in the last decade [49, 36]. However, it remains an open challenge as no theoretical guarantee has been proven yet. Indeed, Dubois-Taine et al. [14] provide a counter-example for intuitive line-search methods. In Appendix E we provide counter-examples of the classical SPS and its variants. The reason behind the failure is the biased curvature information provided by $f_{i_t}$ that prevents global convergence. In this section, we introduce a novel framework to address this issue. Since there exists many variance-reduced stochastic gradient estimators, we focus on the classical SVRG estimator in this section, and our framework also applies to other estimators such as SARAH [40].

### 4.1   Algorithm design: achieving variance-reduction without interpolation

It is known that adaptive methods such as SPS or SLS converge linearly on problems where the interpolation condition holds, i.e. $f(\mathbf{x})$ with $\sigma_{f,B} = 0$.

For problems that do not satisfy the interpolation condition, our approach is to transition the problem to an equivalent one that satisfies the interpolation condition. One such transformation is to shift each individual function by the gradient of $f_i(\mathbf{x})$ at $\mathbf{x}^\star$, i.e. $F_i(\mathbf{x}) = f_i(\mathbf{x}) - \mathbf{x}^T\nabla f_i(\mathbf{x}^\star)$. In this case $f(\mathbf{x})$ can be written as $f(\mathbf{x}) = \frac{1}{n}\sum_{i=1}^n F_i(\mathbf{x})$ due to the fact that $\frac{1}{n}\sum_{i=1}^n \nabla f_i(\mathbf{x}^\star) = 0$. Note that $\nabla F_i(\mathbf{x}^\star) = \nabla f_i(\mathbf{x}^\star) - \nabla f_i(\mathbf{x}^\star) = 0$ which implies that each $F_i(\mathbf{x})$ shares the same minimizer and thus the interpolation condition is satisfied ($\sigma_{f,1}^2 = 0$). However, $\nabla f_i(\mathbf{x}^\star)$ is usually not available at hand. This motivates us to design the following algorithm.

---

**Algorithm 1** (Loopless) AdaSVRPS and AdaSVRLS

---

**Require:** $\mathbf{x}_0 \in \mathbb{R}^d$, $\mu_F > 0$, $c_p > 0$ or $c_l > 0$

1: set $\mathbf{w}_0 = \mathbf{x}_0$, $\eta_{-1} = +\infty$
2: **for** $t = 0$ to $T - 1$ **do**
3:     uniformly sample $i_t \subseteq [n]$
4:     set $F_{i_t}(\mathbf{x}) = f_{i_t}(\mathbf{x}) + \mathbf{x}^T(\nabla f(\mathbf{w}_t) - \nabla f_{i_t}(\mathbf{w}_t)) + \frac{\mu_F}{2}||\mathbf{x} - \mathbf{x}_t||^2$
5:     $\eta_t = \min\left\{\frac{F_{i_t}(\mathbf{x}_t) - F_{i_t}^\star}{c_p||\nabla F_{i_t}(\mathbf{x}_t)||^2} \frac{1}{\sqrt{\sum_{s=0}^t F_{i_s}(\mathbf{x}_s) - F_{i_s}^\star}}, \eta_{t-1}\right\}$ (AdaSVRPS)
6:     $\eta_t = \min\left\{\gamma_t \frac{1}{c_l\sqrt{\sum_{s=0}^t \gamma_s||\nabla F_{i_s}(\mathbf{x}_s)||^2}}, \eta_{t-1}\right\}$ (AdaSVRLS)[2]
7:     $\mathbf{x}_{t+1} = \Pi_{\mathcal{X}}\left(\mathbf{x}_t - \eta_t \nabla F_{i_t}(\mathbf{x}_t)\right)$
8:     $\mathbf{w}_{t+1} = \begin{cases} \mathbf{w}_t & \text{with probability } 1 - p_{t+1} \\ \mathbf{x}_t & \text{with probability } p_{t+1} \end{cases}$
9: **return** $\bar{\mathbf{x}}_T = \frac{1}{T}\sum_{t=0}^{T-1} \mathbf{x}_t$

---

## 4.2 Algorithms and convergence

Inspired by the observation, we attempt to reduce the variance of the functions $\sigma_{f,B}^2$ by constructing a sequence of random functions $\{F_{i_t}(\mathbf{x})\}$ such that $\sigma_{\frac{1}{n}\sum_{i=1}^n F_{i_t}(\mathbf{x}),B}^2 \to 0$ as $\mathbf{x}_t \to \mathbf{x}^\star$. However, directly applying SPS or SLS to $\{F_{i_t}(\mathbf{x})\}$ still requires the knowledge of the Lipschitz constant to guarantee convergence. This problem can be solved by using our proposed AdaSPS and AdaSLS. The whole procedure of the final algorithm is summarized in Algorithm 1.

At each iteration of Algorithm 1, we construct a proxy function by adding two quantities to the minibatch function $f_{i_t}(\mathbf{x})$, where $\frac{\mu_F}{2}||\mathbf{x} - \mathbf{x}_t||^2$ is a proximal term that helps improve the inherent stochasticity due to the partial information obtained from $f_{i_t}(\mathbf{x})$. The additional inner product quantity is used to draw closer the minimizers of $f_{i_t}(\mathbf{x})$ and $f(\mathbf{x})$. Following [27, 33], the full gradient is computed with a coin flip probability. Note that Algorithm 1 still works with $\eta_t$ replaced with SVRG and AdaSVRG stepsize since $\nabla F_{i_t}(\mathbf{x}_t) = \nabla f_{i_t}(\mathbf{x}_t) - \nabla f_{i_t}(\mathbf{w}_t) + \nabla f(\mathbf{w}_t)$, and thus this framework can be seen as a generalization of the standard VR methods. A similar idea can also be found in the works on federated learning with variance-reduction [32, 2, 24, 37, 50].

**Theorem 7.** *Assume each $f_i$ is convex and $L$ smooth and $\mathcal{X}$ is a convex compact feasible set with diameter $D$. Let $p_t = \frac{1}{at+1}$ with $0 \leq a < 1$. Algorithm 1 converges as:*

$$(AdaSVRPS) \qquad \mathbb{E}[f(\bar{\mathbf{x}}_T) - f^\star] \leq \frac{1 + \frac{2L}{(1-a)\mu_F}}{T}\left(2c_p(L + \mu_F)D^2 + \frac{1}{c_p}\right)^2, \qquad (11)$$

$$(AdaSVRLS) \quad \mathbb{E}[f(\bar{\mathbf{x}}_T) - f^\star] \leq \frac{1 + \frac{2L}{(1-a)\mu_F}}{T}\left(\max\left\{\frac{L + \mu_F}{(1-\rho)\sqrt{\rho}}, \frac{1}{\gamma_{\max}\sqrt{\rho}}\right\}c_l D^2 + \frac{1}{c_l\sqrt{\rho}}\right)^2, \quad (12)$$

*where $\bar{\mathbf{x}}_T = \frac{1}{T}\sum_{t=0}^{T-1} \mathbf{x}_t$.*

Suppose $\gamma_{\max}$ is sufficiently large, then picking $\mu_F^\star = \mathcal{O}(L)$, $c_p^\star = \mathcal{O}(\frac{1}{\sqrt{LD^2}})$ and $c_l^\star = \mathcal{O}(\frac{\sqrt{1-\rho}}{\sqrt{LD^2}})$ yields an $\mathcal{O}(\frac{LD^2}{T})$ rate which matches the $\mathcal{O}(\frac{L||\mathbf{x}_0 - \mathbf{x}^\star||^2}{T})$ rate of full-batch gradient descent except for a larger term $D^2$ due to the lack of knowledge of the Lipschitz constant.

**Corollary 8.** *Under the setting of Theorem 7, given an arbitrary accuracy $\varepsilon$, the total number of gradient evaluations required to have $\mathbb{E}[f(\bar{\mathbf{x}}_T) - f^\star] \leq \varepsilon$ in expectation is $\mathcal{O}(\log(1/\varepsilon)n + 1/\varepsilon)$.*

The proved efficiency of stochastic gradient calls matches the optimal rates of SARAH [40]/SVRG and AdaSVRG [14] but removes the artificially designed inner and outer loop size. However, note that Algorithm 1 requires an additional assumption on individual convexity. Unfortunately, we believe this

---

[2]where $\gamma_t$ is obtained via the Armijo backtracking line-search (Algorithm 4) which satisfies: $F_{i_t}(\mathbf{x}_t - \gamma_t\nabla F_{i_t}(\mathbf{x}_t)) \leq F_{i_t}(\mathbf{x}_t) - \rho\gamma_t||\nabla F_{i_t}(\mathbf{x}_t)||^2$ and $\gamma_t \leq \gamma_{\max}$.

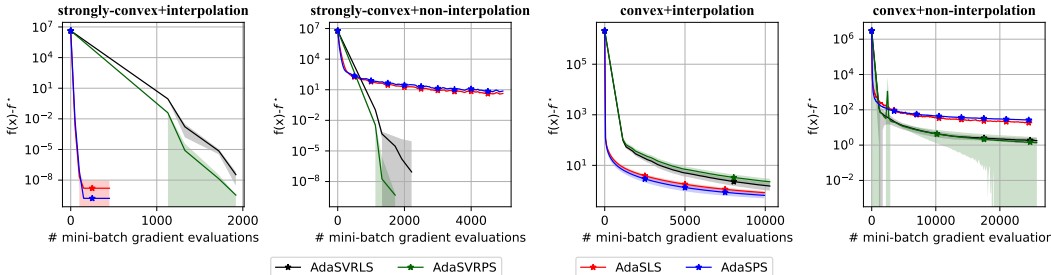

Figure 2: Illustration of the accelerated convergence of AdaSVRPS and AdaSVRLS on quadratic loss without interpolation. Both algorithms require less gradient evaluations than AdaSPS and AdaSLS for optimizing non-interpolated problems. However, they are less efficient for solving interpolated problems. (Repeated 3 times. The solid lines and the shaded area represent the mean and the standard deviation.)

assumption is necessary for the VR methods to work for the algorithms in the Polyak and line-search family since SPS/SLS has to assume the same condition for the proof in the interpolation settings.

**Discussion.** The classical SVRG with Armijo line-search (presented as Algorithm 6 in [14]) employs the same gradient estimator as SVRG but chooses its stepsize based on the returning value of line-search on the individual function $f_i$. Similarly, SVRG with classical Polyak stepsize uses the individual curvature information of $f_i$ to set the stepsize for the global variance-reduced gradient. Due to the misleading curvature information provided by the biased function $f_i$, both methods have convergence issues. In constrast, Algorithm 1 reduces the bias by adding a correction term $\mathbf{x}^T(\nabla f(\mathbf{w}_t) - \nabla f_{i_t}(\mathbf{w}_t))$ with global information to $f_i$ and then applying line-search or Polyak-stepsize on the variance-reduced functions $F_{i_t}$. This difference essentially guarantees the convergence.

## 5 Numerical evaluation

In this section, we illustrate the main properties of our proposed methods in numerical experiments. A detailed description of the experimental setup can be found in Appendix F. We report the theoretically justified hyperparameter $c_p^{\text{scale}}$ or $c_l^{\text{scale}}$ as defined in Theorem 6 rather than $c_p$ or $c_l$ in the following.

**Synthetic data.** We illustrate the robustness property on a class of synthetic problems. We consider the minimization of a quadratic of the form: $f(\mathbf{x}) = \frac{1}{n}\sum_{i=1}^n f_i(\mathbf{x})$ where $f_i(\mathbf{x}) = \frac{1}{2}(\mathbf{x}-\mathbf{b}_i)^T A_i(\mathbf{x}-\mathbf{b}_i)$, $\mathbf{b}_i \in \mathbb{R}^d$ and $A_i \in \mathbb{R}^{d \times d}$ is a diagonal matrix. We use $n = 50$, $d = 1000$. We can control the convexity of the problem by choosing different matrices $A_i$, and control interpolation by either setting all $\{\mathbf{b}_i\}$ to be identical or different. We generate a strongly convex instance where the eigenvalues of $\nabla^2 f(\mathbf{x})$ are between 1 and 10, and a general convex instance by setting some of the eigenvalues to small values close to zero (while ensuring that each $\nabla^2 f_i(\mathbf{x})$ is positive semi-definite). The exact procedure to generate these problems is described in Appendix F.

For all methods, we use a batch size $B = 1$. We compare AdaSPS and AdaSLS against DecSPS [44], SPS [34] and SLS [52] to illustrate the robustness property. More comparisons can be found in Appendix F.1. We fix $c_p^{\text{scale}} = c_l^{\text{scale}} = 1$ for AdaSPS/AdaSLS and use the optimal parameters for DecSPS and SPS. In Figure 1, we observe that SPS does not converge in the non-interpolated settings and DecSPS suffers from a slow $\mathcal{O}(1/\sqrt{T})$ convergence on the two interpolated problems. AdaSPS and AdaSLS show the desired convergence rates across all cases which matches the theory. When the problems are non-interpolated, AdaSVRPS and AdaSVRLS illustrate faster convergence, which can be seen in Figure 2.

**Binary classification on LIBSVM datasets.** We experiment with binary classification on four diverse datasets from [10]. We consider the standard regularized logistic loss: $f(\mathbf{x}) = \frac{1}{n}\sum_{i=1}^n \log(1 + \exp(-y_i \cdot \mathbf{a}_i^T\mathbf{x})) + \frac{1}{2n}||\mathbf{x}||^2$ where $(\mathbf{a}_i, y_i) \in \mathbb{R}^{d+1}$ are features and labels. We defer the study of variance-reduction methods to Appendix F.2 for clarity of presentation. We benchmark against popular optimization algorithms including Adam [26], SPS [34], DecSPS [44], SLS [52] and AdaGrad-Norm [15]. We fix $c_p^{\text{scale}} = c_l^{\text{scale}} = 1$ for AdaSPS/AdaSLS and pick the best learning rate from $\{10^i\}_{i=-4,..,3}$ for SGD, Adam and AdaGrad-Norm. We observe that Adam, SPS and SLS have remarkable performances on duke with $n = 48$ and $d = 7129$, which satisfies interpolation. AdaSPS

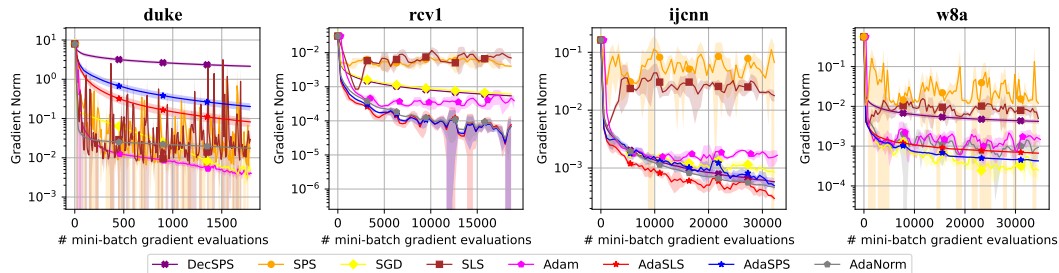

Figure 3: Comparison of AdaSPS/AdaSLS against six other popular optimizers on four LIBSVM datasets, with batch size $B = 1$ for duke, $B = 64$ for rcv1, $B = 64$ for ijcnn and $B = 128$ for w8a. AdaSPS and AdaSLS have competitive performance on rcv1, ijcnn, and w8a while SPS, SLS, and Adam converge fast on duke. (Repeated 3 times. The solid lines and the shaded area represent the mean and the standard deviation.)

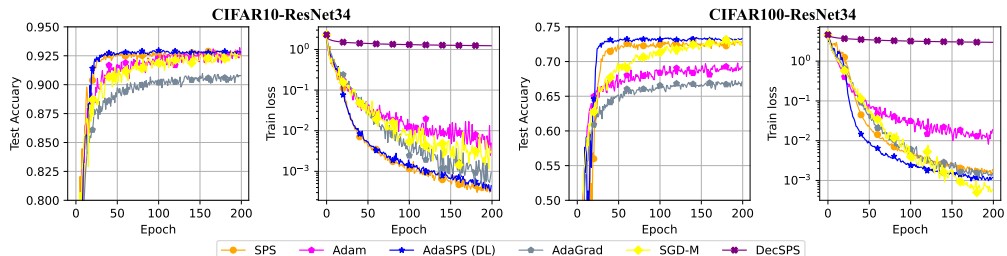

Figure 4: Comparison of the considered optimizers on multi-class classification tasks with CIFAR10 and CIFAR100 datasets using ResNet34 with softmax loss.

and AdaSLS consistently perform reasonably well on the other three larger datasets. It is worth noting that the hyper-parameters $c_p^{\text{scale}}$ and $c_l^{\text{scale}}$ are fixed for all the datasets, which is desired in practice.

**Deep learning task.** We provide a heuristic extension of AdaSPS to over-parameterized non-convex optimization tasks to illustrate its potential. We benchmark the convergence and generalization performance of AdaSPS (DL) 5 for the multi-class classification tasks on CIFAR10 [28] and CIFAR100 [29] datasets using ResNet-34 [21]. More experimental details can be found in Appendix G. We demonstrate the effectiveness of AdaSPS (DL) in Figure 4.

**Discussion.** AdaSPS and AdaSLS consistently demonstrate robust convergence across all tasks and achieve performance on par with, if not better than, the best-tuned algorithms. Consequently, it is reliable and convenient for practical use.

# 6 Conclusion and future work

We proposed new variants of SPS and SLS algorithms and demonstrated their robust and fast convergence in both interpolated and non-interpolated settings. We further accelerate both algorithms for convex optimization with a novel variance reduction technique. Interesting future directions may include: accelerating AdaSPS and AdaSLS with momentum, developing effective robust adaptive methods for training deep neural networks, designing an adaptive algorithm that gives a faster rate $\mathcal{O}(1/T)$ under strong convexity, extensions to distributed and decentralized settings.

# Acknowledgments

We appreciate the fruitful discussion with Anton Rodomanov.

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

# Appendix

## A Technical Preliminaries

### A.1 Basic Definitions

We use the following definitions throughout the paper.

**Definition 1** (convexity). *A differentiable function $f : \mathbb{R}^d \to \mathbb{R}$ is convex if $\forall \, \mathbf{x}, \mathbf{y} \in \mathbb{R}^d$,*

$$f(\mathbf{y}) \geq f(\mathbf{x}) + \langle \nabla f(\mathbf{x}), \mathbf{y} - \mathbf{x} \rangle \; . \tag{A.1}$$

**Definition 2** (strong convexity). *A differentiable function $f : \mathbb{R}^d \to \mathbb{R}$ is $\mu$-strongly convex if $\forall \, \mathbf{x}, \mathbf{y} \in \mathbb{R}^d$,*

$$f(\mathbf{y}) \geq f(\mathbf{x}) + \langle \nabla f(\mathbf{x}), \mathbf{y} - \mathbf{x} \rangle + \frac{\mu}{2} ||\mathbf{x} - \mathbf{y}||^2 \; . \tag{A.2}$$

**Definition 3** ($L$-smooth). *Let function $f : \mathbb{R}^d \to \mathbb{R}$ be differentiable. $f$ is smooth if there exists $L > 0$ such that $\forall \, \mathbf{x}, \mathbf{y} \in \mathbb{R}^d$,*

$$||\nabla f(\mathbf{x}) - \nabla f(\mathbf{y})|| \leq L||\mathbf{x} - \mathbf{y}|| \; . \tag{A.3}$$

### A.2 Useful Lemmas

We frequently use the following helpful lemmas for the proof.

**Lemma 9** (Nesterov [39], Lemma 1.2.3). *Definition 3 implies that there exists a quadratic upper bound on $f$:*

$$f(\mathbf{y}) \leq f(\mathbf{x}) + \langle \nabla f(\mathbf{x}), \mathbf{y} - \mathbf{x} \rangle \, | + \frac{L}{2} ||\mathbf{y} - \mathbf{x}||^2 \, , \forall \mathbf{x}, \mathbf{y} \in \mathbb{R}^d \; . \tag{A.4}$$

**Lemma 10** (Nesterov [39], Theorem 2.1.5). *If a convex function $f$ satisfies (A.4), then it holds that:*

$$f(\mathbf{y}) \geq f(\mathbf{x}) + \langle \nabla f(\mathbf{x}), \mathbf{y} - \mathbf{x} \rangle \, | + \frac{1}{2L} ||\nabla f(\mathbf{y}) - \nabla f(\mathbf{x})||^2 \, , \forall \mathbf{x}, \mathbf{y} \in \mathbb{R}^d. \tag{A.5}$$

**Lemma 11** (Ward et al. [55]). *For any non-negative sequence $a_0, ..., a_T$, the following holds:*

$$\sqrt{\sum_{t=0}^{T} a_t} \leq \sum_{t=0}^{T} \frac{a_t}{\sqrt{\sum_{i=0}^{t} a_i}} \leq 2 \sqrt{\sum_{t=0}^{T} a_t} \; . \tag{A.6}$$

*If $a_0 \geq 1$, then the following holds:*

$$\sum_{t=0}^{T} \frac{a_t}{\sum_{i=0}^{t} a_i} \leq \log(\sum_{t=0}^{T} a_t) + 1 \; . \tag{A.7}$$

*Proof.* To show equation (A.6), we proceed with the proof by induction. For $t = 0$, (A.6) holds trivially since $\sqrt{a_0} \leq \sqrt{a_0} \leq 2\sqrt{a_0}$. Assume equation (A.6) holds for $T - 1$. For RHS, we have:

$$\sum_{t=0}^{T-1} \frac{a_t}{\sum_{i=0}^{t} a_i} + \frac{a_T}{\sqrt{\sum_{i=0}^{T} a_i}} \leq 2 \sqrt{\sum_{t=0}^{T-1} a_t} + \frac{a_T}{\sqrt{\sum_{i=0}^{T} a_i}}$$

$$= 2 \sqrt{\sum_{t=0}^{T} a_t - a_T + \frac{a_T}{\sqrt{\sum_{t=0}^{T} a_t}}} \tag{A.8}$$

$$\leq 2 \sqrt{\sum_{t=0}^{T} a_t} \; .$$

where the last inequality is due to the fact that $2\sqrt{x-y} + \frac{y}{\sqrt{x}} \leq 2\sqrt{x}$ for any $x \geq y \geq 0$. For LHS, we have:

$$\sum_{t=0}^{T-1} \frac{a_t}{\sum_{i=0}^{t} a_i} + \frac{a_T}{\sqrt{\sum_{i=0}^{T} a_i}} \geq \sqrt{\sum_{t=0}^{T-1} a_t + \frac{a_T}{\sqrt{\sum_{i=0}^{T} a_i}}}$$

$$= \sqrt{\sum_{t=0}^{T} a_t - a_T + \frac{a_T}{\sqrt{\sum_{t=0}^{T} a_t}}} \qquad (A.9)$$

$$\geq \sqrt{\sum_{t=0}^{T} a_t} .$$

where the last inequality is due to the fact that $\sqrt{x-y} + \frac{y}{\sqrt{x}} \geq \sqrt{x}$ for any $x \geq y \geq 0$.

We next show equation (A.7) by induction. For $t = 0$, equation (A.7) trivially holds since $1 \leq \log(a_0) + 1$. Assume (A.7) holds for $T - 1$, we have:

$$\sum_{t=0}^{T} \frac{a_t}{\sum_{i=0}^{t} a_i} \leq \log(\sum_{t=0}^{T-1} a_t) + 1 + \frac{a_T}{\sum_{i=0}^{T} a_i}$$

$$\leq \log(\sum_{t=0}^{T} a_t) + 1 . \qquad (A.10)$$

where the last inequality is due to the fact that $\log(x-y) + \frac{y}{x} \leq \log(x)$ for any $x \geq y \geq 0$ since $e^{\frac{y}{x}} \leq \frac{1+\frac{y}{x}}{1-\frac{y^2}{x^2}}$. $\qquad\square$

**Lemma 12** (Dubois-Taine et al. [14, Lemma 5]). *If $x^2 \leq a(x+b)$ for $a \geq 0$ and $b \geq 0$, then it holds that:*

$$x \leq a + \sqrt{ab} . \qquad (A.11)$$

The following Lemma is an extension of Lemma 5 in [44].

**Lemma 13.** *Let $z_{t+1} \leq (1 - a\eta_t)z_t + \eta_t b$ and $z_t \geq 0$ where $a > 0$, $b > 0$ and $\eta_t > 0, \eta_{t+1} \leq \eta_t, \forall t \geq 0$. It holds that:*

$$z_t \leq \max\{\frac{b}{a}, z_0, \eta_0 b\}, \ \forall t \geq 0 . \qquad (A.12)$$

*Proof.* Since $\eta_t$ is non-increasing, $1 - a\eta_t \leq 0$ is non-decreasing. For any $t \geq 0$ such that $1 - a\eta_t \leq 0$, we have $z_{t+1} \leq \eta_t b \leq \eta_0 b$. If $1 - a\eta_t \leq 0$ for all $t \geq 0$, then the proof is done. Otherwise, let us assume there exists a first index $j$ such that $1 - a\eta_j > 0$ and we have $z_j \leq \max\{z_0, \eta_0 b\} := \tilde{z}_0$. We proceed with the proof starting with the index $j$ by induction. For $t = j$, the lemma trivially holds. Let us assume $z_t \leq \max\{\frac{b}{a}, \tilde{z}_0\}$ for $t > j$. If $\frac{b}{a} \geq \tilde{z}_0$, then by induction, we have:

$$z_{t+1} \leq (1 - a\eta_t)\frac{b}{a} + \eta_t b = \frac{b}{a} . \qquad (A.13)$$

If instead $\frac{b}{a} \leq \tilde{z}_0$, then by induction, we have:

$$z_{t+1} \leq (1 - a\eta_t)\tilde{z}_0 + \eta_t b = \tilde{z}_0 - \eta_t(a\tilde{z}_0 - b) \leq \tilde{z}_0 . \qquad (A.14)$$

Combining the above cases concludes the proof. $\qquad\square$

The following lemma is commonly used in the works on Polyak stepsize [34, 20].

**Lemma 14.** *Suppose a function $f$ is $L$-smooth and $\mu$-strongly convex, then the following holds:*

$$\frac{1}{2L} \leq \frac{f(\mathbf{x}) - f^\star}{||\nabla f(\mathbf{x})||^2} \leq \frac{1}{2\mu} . \qquad (A.15)$$

The following lemma provides upper and lower bounds for the stepsize of AdaSPS.

**Lemma 15.** *Suppose each $f_i$ is $L$-smooth, then the stepsize of AdaSPS* (AdaSPS) *satisfies:*

$$\frac{1}{2c_p L} \frac{1}{\sqrt{\sum_{s=0}^{t} f_{i_s}(\mathbf{x}_s) - \ell_{i_s}^\star}} \leq \eta_t \leq \frac{f_{i_t}(\mathbf{x}_t) - \ell_{i_t}^\star}{c_p||\nabla f_{i_t}(\mathbf{x}_t)||^2} \frac{1}{\sqrt{\sum_{s=0}^{t} f_{i_s}(\mathbf{x}_s) - \ell_{i_s}^\star}} . \qquad (A.16)$$

*Proof.* The upper bound follows from the definition of the stepsize (AdaSPS). To prove the lower bound, we note that the stepsize (AdaSPS) is composed of two parts where the first component $\frac{f_{i_s}(\mathbf{x}_s)-\ell^\star_{i_s}}{c_p||\nabla f_{i_s}(\mathbf{x}_s)||^2} \geq \frac{1}{2c_p L}$ for all $0 \leq s \leq t$ due to (A.15), and the second component is always decreasing. Finally, recall that $\eta_{-1} = +\infty$ and thus the proof is completed. $\qquad\square$

The following lemma provides upper and lower bounds for the stepsize of AdaSLS. We refer to Appendix D for details of the line-search procedure.

**Lemma 16.** *Suppose each $f_i$ is $L$-smooth, then the stepsize of AdaSLS (AdaSLS) satisfies:*

$$\min\Big\{\frac{1-\rho}{L}, \gamma_{\max}\Big\} \frac{1}{c_l\sqrt{\sum_{s=0}^t \gamma_s||\nabla f_{i_s}(\mathbf{x}_s)||^2}} \leq \eta_t \leq \frac{\gamma_t}{c_l\sqrt{\sum_{s=0}^t \gamma_s||\nabla f_{i_s}(\mathbf{x}_s)||^2}} . \tag{A.17}$$

*Proof.* The upper bound is due to the definition of the stepsize (AdaSLS). We next prove the lower bound. From the smoothness definition, the following holds for all $\gamma_t$:

$$f_{i_t}(\mathbf{x}_t - \gamma_t \nabla f_{i_t}(\mathbf{x}_t)) \overset{(A.4)}{\leq} f_{i_t}(\mathbf{x}_t) - \gamma_t||\nabla f_{i_t}(\mathbf{x}_t)||^2 + \frac{L}{2}\gamma_t^2||\nabla f_{i_t}(\mathbf{x}_t)||^2 . \tag{A.18}$$

For any $0 < \gamma_t \leq \frac{2(1-\rho)}{L}$, we have:

$$f_{i_t}(\mathbf{x}_t - \gamma_t \nabla f_{i_t}(\mathbf{x}_t)) \leq f_{i_t}(\mathbf{x}_t) - \rho\gamma_t||\nabla f_{i_t}(\mathbf{x}_t)||^2 , \tag{A.19}$$

which satisfies the line-search condition (4). From the procedure of Backtracking line-search (Alg. 4), if $\gamma_{\max} \leq \frac{1-\rho}{L}$, then $\gamma_t = \gamma_{\max}$ is accepted. Otherwise, since we require the decreasing factor $\beta$ **to be no smaller than** $\frac{1}{2}$ in Algorithm 4, we must have $\gamma_t \geq \frac{2(1-\rho)}{2L}$. Therefore, $\gamma_t$ is always lower bounded by $\min\{\frac{1-\rho}{L}, \gamma_{\max}\}$. The second component of AdaSLS is always decreasing, and recall that $\eta_{-1} = +\infty$. The proof is thus completed. $\qquad\square$

# B   Proofs of main results

## B.1   Proof of Theorem 1

*Proof.* We follow a common proof routine for the general convex optimization [15, 14, 44]. Using the update rule of PSGD (5), we have:

$$
\begin{aligned}
||\mathbf{x}_{t+1} - \mathbf{x}^\star||^2 &= ||\Pi_{\mathcal{X}}(\mathbf{x}_t - \eta_t \nabla f_{i_t}(\mathbf{x}_t)) - \Pi_{\mathcal{X}}(\mathbf{x}^\star)||^2 \\
&\leq ||\mathbf{x}_t - \eta_t \nabla f_{i_t}(\mathbf{x}_t) - \mathbf{x}^\star||^2 \\
&= ||\mathbf{x}_t - \mathbf{x}^\star||^2 - 2\eta_t\langle \nabla f_{i_t}(\mathbf{x}_t), \mathbf{x}_t - \mathbf{x}^\star\rangle + \eta_t^2||\nabla f_{i_t}(\mathbf{x}_t)||^2 .
\end{aligned} \tag{B.1}
$$

Dividing by $2\eta_t$ and rearranging gives:

$$
\begin{aligned}
&\langle \nabla f_{i_t}(\mathbf{x}_t), \mathbf{x}_t - \mathbf{x}^\star\rangle \\
&\leq \frac{||\mathbf{x}_t - \mathbf{x}^\star||^2}{2\eta_t} - \frac{||\mathbf{x}_{t+1} - \mathbf{x}^\star||^2}{2\eta_t} + \frac{\eta_t}{2}||\nabla f_{i_t}(\mathbf{x}_t)||^2 \\
&= \frac{||\mathbf{x}_t - \mathbf{x}^\star||^2}{2\eta_t} - \frac{||\mathbf{x}_{t+1} - \mathbf{x}^\star||^2}{2\eta_{t+1}} + \frac{||\mathbf{x}_{t+1} - \mathbf{x}^\star||^2}{2\eta_{t+1}} - \frac{||\mathbf{x}_{t+1} - \mathbf{x}^\star||^2}{2\eta_t} + \frac{\eta_t}{2}||\nabla f_{i_t}(\mathbf{x}_t)||^2 .
\end{aligned} \tag{B.2}
$$

Summing from $t = 0$ to $t = T - 1$, we get:

$$
\begin{aligned}
&\sum_{t=0}^{T-1}\langle \nabla f_{i_t}(\mathbf{x}_t), \mathbf{x}_t - \mathbf{x}^\star\rangle \\
&\leq \sum_{t=0}^{T-1}\frac{||\mathbf{x}_t - \mathbf{x}^\star||^2}{2\eta_t} - \frac{||\mathbf{x}_{t+1} - \mathbf{x}^\star||^2}{2\eta_{t+1}} + \frac{||\mathbf{x}_{t+1} - \mathbf{x}^\star||^2}{2\eta_{t+1}} - \frac{||\mathbf{x}_{t+1} - \mathbf{x}^\star||^2}{2\eta_t} + \frac{\eta_t}{2}||\nabla f_{i_t}(\mathbf{x}_t)||^2 \\
&\leq \frac{||\mathbf{x}_0 - \mathbf{x}^\star||^2}{2\eta_0} - \frac{||\mathbf{x}_T - \mathbf{x}^\star||^2}{2\eta_T} + \frac{||\mathbf{x}_T - \mathbf{x}^\star||^2}{2\eta_T} - \frac{||\mathbf{x}_T - \mathbf{x}^\star||^2}{2\eta_{T-1}} + \sum_{t=0}^{T-2}(\frac{1}{2\eta_{t+1}} - \frac{1}{2\eta_t})D^2 + \sum_{t=0}^{T-1}\frac{\eta_t}{2}||\nabla f_{i_t}(\mathbf{x}_t)||^2 \\
&\leq \frac{||\mathbf{x}_0 - \mathbf{x}^\star||^2}{2\eta_0} - \frac{||\mathbf{x}_T - \mathbf{x}^\star||^2}{2\eta_T} + \frac{||\mathbf{x}_T - \mathbf{x}^\star||^2}{2\eta_T} + \frac{D^2}{2\eta_{T-1}} + \sum_{t=0}^{T-1}\frac{\eta_t}{2}||\nabla f_{i_t}(\mathbf{x}_t)||^2
\end{aligned}
$$

$$= \frac{||\mathbf{x}_0 - \mathbf{x}^\star||^2}{2\eta_0} + \frac{D^2}{2\eta_{T-1}} + \sum_{t=0}^{T-1} \frac{\eta_t}{2} ||\nabla f_{i_t}(\mathbf{x}_t)||^2 , \tag{B.3}$$

where in the second inequality, we use the decreasing property of the stepsize $\eta_t$ which guarantees $\frac{1}{2\eta_t} - \frac{1}{2\eta_{t-1}} \geq 0$, and we use the fact that $||\mathbf{x}_t - \mathbf{x}^\star||^2 \leq D^2$ because of the projection step in (5). For clarity, we next separate the proof for AdaSPS and AdaSLS.

**AdaSPS:** We upper bound the last two terms by using Lemma 15 and we obtain:

$$\sum_{t=0}^{T-1} \frac{\eta_t}{2} ||\nabla f_{i_t}(\mathbf{x}_t)||^2 \overset{(A.16)}{\leq} \sum_{t=0}^{T-1} \frac{f_{i_t}(\mathbf{x}_t) - \ell_{i_t}^\star}{2c_p \sqrt{\sum_{s=0}^t f_{i_s}(\mathbf{x}_s) - \ell_{i_s}^\star}} \overset{(A.6)}{\leq} \frac{1}{c_p} \sqrt{\sum_{s=0}^{T-1} f_{i_s}(\mathbf{x}_s) - \ell_{i_s}^\star} , \tag{B.4}$$

and

$$\frac{D^2}{2\eta_{T-1}} \overset{(A.16)}{\leq} c_p L D^2 \sqrt{\sum_{s=0}^{T-1} f_{i_s}(\mathbf{x}_s) - \ell_{i_s}^\star} . \tag{B.5}$$

Using $\frac{||\mathbf{x}_0 - \mathbf{x}^\star||^2}{2\eta_0} \leq \frac{D^2}{2\eta_{T-1}}$ and plugging (B.4) and (B.5) back to (B.3) gives:

$$\sum_{t=0}^{T-1} \langle \nabla f_{i_t}(\mathbf{x}_t), \mathbf{x}_t - \mathbf{x}^\star \rangle \leq (2c_p L D^2 + \frac{1}{c_p}) \sqrt{\sum_{s=0}^{T-1} f_{i_s}(\mathbf{x}_s) - \ell_{i_s}^\star} . \tag{B.6}$$

Taking the expectation on both sides, we have:

$$\sum_{t=0}^{T-1} \mathbb{E}[\langle \nabla f(\mathbf{x}_t), \mathbf{x}_t - \mathbf{x}^\star \rangle] \leq (2c_p L D^2 + \frac{1}{c_p}) \mathbb{E}\Big[ \sqrt{\sum_{s=0}^{T-1} f_{i_s}(\mathbf{x}_s) - \ell_{i_s}^\star} \Big]$$
$$= (2c_p L D^2 + \frac{1}{c_p}) \mathbb{E}\Big[ \sqrt{\sum_{s=0}^{T-1} f_{i_s}(\mathbf{x}_s) - f_{i_s}(\mathbf{x}^\star) + f_{i_s}(\mathbf{x}^\star) - \ell_{i_s}^\star} \Big] . \tag{B.7}$$

Using the convexity assumption of $f$ and applying Jensen's inequality to the square root function, we get:

$$\sum_{t=0}^{T-1} \mathbb{E}[f(\mathbf{x}_t) - f^\star] \leq (2c_p L D^2 + \frac{1}{c_p}) \sqrt{\sum_{s=0}^{T-1} \mathbb{E}[f(\mathbf{x}_s) - f^\star] + \sigma_{f,B}^2 + \mathrm{err}_{f,B}^2} , \tag{B.8}$$

where $\mathrm{err}_{f,B}^2 = \mathbb{E}_{i_s}[f_{i_s}^\star - \ell_{i_s}^\star]$. Let $\tau := 2c_p L D^2 + \frac{1}{c_p}$. Taking the square gives:

$$\Big( \sum_{t=0}^{T-1} \mathbb{E}[f(\mathbf{x}_t) - f^\star] \Big)^2 \leq \tau^2 \Big( \sum_{t=0}^{T-1} \mathbb{E}[f(\mathbf{x}_t) - f^\star] + T(\sigma_{f,B}^2 + \mathrm{err}_{f,B}^2) \Big) . \tag{B.9}$$

We next apply Lemma 12 with $x = \sum_{t=0}^{T-1} \mathbb{E}[f(\mathbf{x}_t) - f(\mathbf{x}^\star)]$, $a = \tau^2$ and $b = T(\sigma_{f,B}^2 + \mathrm{err}_{f,B}^2)$:

$$\sum_{t=0}^{T-1} \mathbb{E}[f(\mathbf{x}_t) - f^\star] \leq \tau^2 + \tau \sqrt{\sigma_{f,B}^2 + \mathrm{err}_{f,B}^2} \sqrt{T} . \tag{B.10}$$

We conclude by dividing both sides by $T$ and using Jensen's inequality:

$$\mathbb{E}[f(\bar{\mathbf{x}}_T) - f^\star] \leq \frac{\sum_{t=0}^{T-1} \mathbb{E}[f(\mathbf{x}_t) - f^\star]}{T} \leq \frac{\tau^2}{T} + \frac{\tau \sqrt{\sigma_{f,B}^2 + \mathrm{err}_{f,B}^2}}{\sqrt{T}} . \tag{B.11}$$

where $\bar{\mathbf{x}}_T = \frac{1}{T} \sum_{t=0}^{T-1} \mathbf{x}_t$.

**AdaSLS:** The proof is almost the same as AdaSPS. We omit procedures with the same proof reasons for simplicity. We first use Lemma 16 to obtain:

$$\sum_{t=0}^{T-1} \frac{\eta_t}{2} ||\nabla f_{i_t}(\mathbf{x}_t)||^2 \overset{(A.17)}{\leq} \sum_{t=0}^{T-1} \frac{\gamma_t ||\nabla f_{i_t}(\mathbf{x}_t)||^2}{2c_l \sqrt{\sum_{s=0}^t \gamma_s ||\nabla f_{i_s}(\mathbf{x}_s)||^2}} \overset{(A.6)}{\leq} \frac{1}{c_l} \sqrt{\sum_{s=0}^{T-1} \gamma_s ||\nabla f_{i_s}(\mathbf{x}_s)||^2} , \tag{B.12}$$

and

$$\frac{D^2}{2\eta_{T-1}} \overset{(A.17)}{\leq} \frac{c_l \sqrt{\sum_{s=0}^{T-1} \gamma_s ||\nabla f_{i_s}(\mathbf{x}_s)||^2} D^2}{2 \min\left\{ \frac{1-\rho}{L}, \gamma_{\max} \right\}} = \frac{\max\{ \frac{L}{1-\rho}, \frac{1}{\gamma_{\max}} \} c_l D^2}{2} \sqrt{\sum_{s=0}^{T-1} \gamma_s ||\nabla f_{i_s}(\mathbf{x}_s)||^2} . \tag{B.13}$$

Inequality (B.3) can then be further bounded by:

$$\sum_{t=0}^{T-1}\langle\nabla f_{i_t}(\mathbf{x}_t),\mathbf{x}_t-\mathbf{x}^\star\rangle \le \left(\max\left\{\frac{L}{1-\rho},\frac{1}{\gamma_{\max}}\right\}c_l D^2+\frac{1}{c_l}\right)\sqrt{\sum_{s=0}^{T-1}\gamma_s||\nabla f_{i_s}(\mathbf{x}_s)||^2}$$

$$\le\left(\max\left\{\frac{L}{(1-\rho)\sqrt{\rho}},\frac{1}{\gamma_{\max}\sqrt{\rho}}\right\}c_l D^2+\frac{1}{c_l\sqrt{\rho}}\right)\sqrt{\sum_{s=0}^{T-1}f_{i_s}(\mathbf{x}_s)-f_{i_s}^\star}. \tag{B.14}$$

where we used line-search condition (4) and the fact that $f_{i_s}(\mathbf{x}_t-\gamma_t\nabla f_{i_s}(\mathbf{x}_s))\ge f_{i_s}^\star$.

Let $\tau:=\max\left\{\frac{L}{(1-\rho)\sqrt{\rho}},\frac{1}{\gamma_{\max}\sqrt{\rho}}\right\}c_l D^2+\frac{1}{c_l\sqrt{\rho}}$. We arrive at:

$$\mathbb{E}[f(\bar{\mathbf{x}}_T)-f^\star]\le\frac{\tau^2}{T}+\frac{\tau\sigma_{f,B}}{\sqrt{T}}. \tag{B.15}$$

where $\bar{\mathbf{x}}_T=\frac{1}{T}\sum_{t=0}^{T-1}\mathbf{x}_t$. $\qquad\square$

## B.2 Full statement and proof for Lemma 4

**Lemma 17** (Bounded iterates). *Let each $f_i$ be $\mu$-strongly convex and $L$-smooth. For any $t\in\mathbb{N}$, the iterates of SGD with AdaSPS or AdaSLS satisfy:*

$$||\mathbf{x}_t-\mathbf{x}^\star||^2\le D_{\max}:=\max\left\{||\mathbf{x}_0-\mathbf{x}^\star||^2,\frac{2\sigma_{\max}^2+b}{\mu},(2\sigma_{\max}^2+b)\eta_0\right\}, \tag{B.16}$$

*where $\sigma_{\max}^2:=\max_{i_t}\{f_{i_t}(\mathbf{x}^\star)-\ell_{i_t}^\star\}$, $b:=1/\left(4c_p^3\sqrt{f_{i_0}(\mathbf{x}_0)-\ell_{i_0}^\star}\right)$ for AdaSPS and $\sigma_{\max}^2:=\max_{i_t}\{f_{i_t}(\mathbf{x}^\star)-f_{i_t}^\star\}$, $b:=1/\left(4c_l^3\rho^2\sqrt{\gamma_0||\nabla f_{i_0}(\mathbf{x}_0)||^2}\right)$ for AdaSLS.*

*Proof.* By strong convexity of $f_{i_t}$, the iterates generated by SGD satisfy:

$$||\mathbf{x}_{t+1}-\mathbf{x}^\star||^2=||\mathbf{x}_t-\mathbf{x}^\star||^2-2\eta_t\langle\nabla f_{i_t}(\mathbf{x}_t),\mathbf{x}_t-\mathbf{x}^\star\rangle+\eta_t^2||\nabla f_{i_t}(\mathbf{x}_t)||^2$$

$$\overset{(A.2)}{\le}||\mathbf{x}_t-\mathbf{x}^\star||^2-2\eta_t(f_{i_t}(\mathbf{x}_t)-f_{i_t}(\mathbf{x}^\star)+\frac{\mu}{2}||\mathbf{x}_t-\mathbf{x}^\star||^2)+\eta_t^2||\nabla f_{i_t}(\mathbf{x}_t)||^2 \tag{B.17}$$

$$=(1-\eta_t\mu)||\mathbf{x}_t-\mathbf{x}^\star||^2-2\eta_t(f_{i_t}(\mathbf{x}_t)-f_{i_t}(\mathbf{x}^\star))+\eta_t^2||\nabla f_{i_t}(\mathbf{x}_t)||^2.$$

We next separate the proofs for clarity.

**AdaSPS**: Plugging in the upper bound of $\eta_t$ in Lemma 15, we obtain:

$$||\mathbf{x}_{t+1}-\mathbf{x}^\star||^2$$

$$\overset{(A.16)}{\le}(1-\eta_t\mu)||\mathbf{x}_t-\mathbf{x}^\star||^2-2\eta_t(f_{i_t}(\mathbf{x}_t)-f_{i_t}(\mathbf{x}^\star))+\eta_t\frac{f_{i_t}(\mathbf{x}_t)-\ell_{i_t}^\star}{c_p\sqrt{\sum_{s=0}^t f_{i_s}(\mathbf{x}_t)-\ell_{i_s}^\star}}$$

$$=(1-\eta_t\mu)||\mathbf{x}_t-\mathbf{x}^\star||^2+2\eta_t(f_{i_t}(\mathbf{x}^\star)-\ell_{i_t}^\star)-2\eta_t(f_{i_t}(\mathbf{x}_t)-\ell_{i_t}^\star)+\eta_t\frac{f_{i_t}(\mathbf{x}_t)-\ell_{i_t}^\star}{c_p\sqrt{\sum_{s=0}^t f_{i_s}(\mathbf{x}_t)-\ell_{i_s}^\star}}$$

$$\le(1-\eta_t\mu)||\mathbf{x}_t-\mathbf{x}^\star||^2+2\eta_t\sigma_{\max}^2-2\eta_t\underbrace{(f_{i_t}(\mathbf{x}_t)-\ell_{i_t}^\star)}_{\ge 0}+\eta_t\frac{f_{i_t}(\mathbf{x}_t)-\ell_{i_t}^\star}{c_p\sqrt{\sum_{s=0}^t f_{i_s}(\mathbf{x}_t)-\ell_{i_s}^\star}}, \tag{B.18}$$

where $\sigma_{\max}^2:=\max_{i_t}\{f_{i_t}(\mathbf{x}^\star)-\ell_{i_t}^\star\}$.

We now split the proof into two cases. Firstly, if $c_p\sqrt{\sum_{s=0}^t f_{i_s}(\mathbf{x}_t)-\ell_{i_s}^\star}\le\frac{1}{2}$, then it follows that:

$$f_{i_t}(\mathbf{x}_t)-\ell_{i_t}^\star\le(\frac{1}{2c_p})^2\quad\text{and}\quad\sum_{s=0}^t f_{i_s}(\mathbf{x}_t)-\ell_{i_s}^\star\ge f_{i_0}(\mathbf{x}_0)-\ell_{i_0}^\star. \tag{B.19}$$

Plugging in the above bounds, we have:

$$||\mathbf{x}_{t+1}-\mathbf{x}^\star||^2\le(1-\eta_t\mu)||\mathbf{x}_t-\mathbf{x}^\star||^2+\eta_t(2\sigma_{\max}^2+\frac{1}{4c_p^3\sqrt{f_{i_0}(\mathbf{x}_0)-\ell_{i_0}^\star}}). \tag{B.20}$$

We conclude by applying Lemma 13 with $z_t = ||\mathbf{x}_t - \mathbf{x}^\star||^2$, $a = \mu$, $b = (2\sigma_{\max}^2 + \frac{1}{4c_p^3\sqrt{f_{i_0}(\mathbf{x}_0) - \ell_{i_0}^\star}})$. Secondly, if instead $c_p\sqrt{\sum_{s=0}^t f_{i_s}(\mathbf{x}_t) - \ell_{i_s}^\star} \geq \frac{1}{2}$, then we have:

$$-2\eta_t(f_{i_t}(\mathbf{x}_t) - \ell_{i_t}^\star) + \eta_t \frac{f_{i_t}(\mathbf{x}_t) - \ell_{i_t}^\star}{c_p\sqrt{\sum_{s=0}^t f_{i_s}(\mathbf{x}_t) - \ell_{i_s}^\star}} \leq 0 \, , \tag{B.21}$$

and consequently we can apply Lemma 13 with $z_t = ||\mathbf{x}_t - \mathbf{x}^\star||^2$, $a = \mu$, $b = 2\sigma_{\max}^2$.

**AdaSLS**: Similarly, by plugging the upper bound of $\eta_t$ in Lemma 16, we obtain:

$$||\mathbf{x}_{t+1} - \mathbf{x}^\star||^2$$
$$\overset{(A.17)}{\leq} (1 - \eta_t\mu)||\mathbf{x}_t - \mathbf{x}^\star||^2 - 2\eta_t(f_{i_t}(\mathbf{x}_t) - f_{i_t}(\mathbf{x}^\star)) + \eta_t \frac{\gamma_t||\nabla f_{i_t}(\mathbf{x}_t)||^2}{c_l\sqrt{\sum_{s=0}^t \gamma_s|||\nabla f_{i_s}(\mathbf{x}_s)||^2}}$$
$$\leq (1 - \eta_t\mu)||\mathbf{x}_t - \mathbf{x}^\star||^2 + 2\eta_t(f_{i_t}(\mathbf{x}^\star) - f_{i_t}^\star) - 2\eta_t(f_{i_t}(\mathbf{x}_t) - f_{i_t}^\star) + \eta_t \frac{f_{i_t}(\mathbf{x}_t) - f_{i_t}^\star}{c_l\rho\sqrt{\sum_{s=0}^t \gamma_s|||\nabla f_{i_s}(\mathbf{x}_s)||^2}}$$
$$\leq (1 - \eta_t\mu)||\mathbf{x}_t - \mathbf{x}^\star||^2 + 2\eta_t\sigma_{\max}^2 - 2\eta_t \underbrace{(f_{i_t}(\mathbf{x}_t) - f_{i_t}^\star)}_{\geq 0} + \eta_t \frac{f_{i_t}(\mathbf{x}_t) - f_{i_t}^\star}{c_l\rho\sqrt{\sum_{s=0}^t \gamma_s|||\nabla f_{i_s}(\mathbf{x}_s)||^2}} \, , \tag{B.22}$$

where $\sigma_{\max}^2 = \max_{i_t}\{f_{i_t}(\mathbf{x}^\star) - f_{i_t}^\star\}$. We can then compare $c_l\rho\sqrt{\sum_{s=0}^t \gamma_s|||\nabla f_{i_s}(\mathbf{x}_s)||^2}$ with $\frac{1}{2}$ and apply Lemma 13 correspondingly. $\qquad\square$

## B.3   Proofs for Theorem 2 and 6

*Proof.* For clarity, we separate the proofs for AdaSPS and AdaSLS.

**AdaSPS:** Plugging in the upper bound of $\eta_t$ in Lemma 15, we have:

$$||\mathbf{x}_{t+1} - \mathbf{x}^\star||^2 \overset{(A.16)}{\leq} ||\mathbf{x}_t - \mathbf{x}^\star||^2 - 2\eta_t\langle\nabla f_{i_t}(\mathbf{x}_t), \mathbf{x}_t - \mathbf{x}^\star\rangle + \eta_t \frac{f_{i_t}(\mathbf{x}_t) - f^\star}{c_p\sqrt{\sum_{s=0}^t f_{i_s}(\mathbf{x}_s) - f^\star}} \, . \tag{B.23}$$

Since $c_p\sqrt{f_{i_0}(\mathbf{x}_0) - f^\star} \geq 1$, (B.23) can be reduced to:

$$||\mathbf{x}_{t+1} - \mathbf{x}^\star||^2 \leq ||\mathbf{x}_t - \mathbf{x}^\star||^2 - 2\eta_t\langle\nabla f_{i_t}(\mathbf{x}_t), \mathbf{x}_t - \mathbf{x}^\star\rangle + \eta_t(f_{i_t}(\mathbf{x}_t) - f^\star) \, . \tag{B.24}$$

By convexity of $f_{i_t}$, we get:

$$||\mathbf{x}_{t+1} - \mathbf{x}^\star||^2 \overset{(A.1)}{\leq} ||\mathbf{x}_t - \mathbf{x}^\star||^2 - \eta_t\langle\nabla f_{i_t}(\mathbf{x}_t), \mathbf{x}_t - \mathbf{x}^\star\rangle \, . \tag{B.25}$$

Note that $\langle\nabla f_{i_t}(\mathbf{x}_t), \mathbf{x}_t - \mathbf{x}^\star\rangle \geq 0$ and $\eta_t$ is non-increasing, we thus get:

$$||\mathbf{x}_{t+1} - \mathbf{x}^\star||^2 \leq ||\mathbf{x}_t - \mathbf{x}^\star||^2 - \eta_{T-1}\langle\nabla f_{i_t}(\mathbf{x}_t), \mathbf{x}_t - \mathbf{x}^\star\rangle \, . \tag{B.26}$$

We first show that $\eta_{T-1}$ is always lower bounded. From equation (B.25) and using convexity of $f_{i_t}$, we get:

$$\eta_t(f_{i_t}(\mathbf{x}_t) - f^\star) \leq ||\mathbf{x}_t - \mathbf{x}^\star||^2 - ||\mathbf{x}_{t+1} - \mathbf{x}^\star||^2 \, . \tag{B.27}$$

Summing from $t = 0$ to $t = T - 1$, we get:

$$\sum_{t=0}^{T-1} \eta_t(f_{i_t}(\mathbf{x}_t) - f^\star) \leq ||\mathbf{x}_0 - \mathbf{x}^\star||^2 \, . \tag{B.28}$$

Using the lower bound of $\eta_t$, we get:

$$\frac{1}{2c_pL}\sqrt{\sum_{s=0}^{T-1} f_{i_s}(\mathbf{x}_s) - f^\star} \overset{(A.6)}{\leq} \frac{1}{2c_pL}\sum_{t=0}^{T-1} \frac{f_{i_t}(\mathbf{x}_t) - f^\star}{\sqrt{\sum_{s=0}^t f_{i_s}(\mathbf{x}_s) - f^\star}} \overset{(A.16)}{\leq} \sum_{t=0}^{T-1} \eta_t(f_{i_t}(\mathbf{x}_t) - f^\star) \, , \tag{B.29}$$

This reveals that:

$$\eta_{T-1} \overset{(A.16)}{\geq} \frac{1}{2c_pL}\frac{1}{\sqrt{\sum_{s=0}^{T-1} f_{i_s}(\mathbf{x}_s) - f^\star}} \geq \frac{1}{(2c_pL||\mathbf{x}_0 - \mathbf{x}^\star||)^2} \, . \tag{B.30}$$

Plugging in the lower bound of $\eta_{T-1}$ to (B.26), we obtain:

$$||\mathbf{x}_{t+1} - \mathbf{x}^\star||^2 \leq ||\mathbf{x}_t - \mathbf{x}^\star||^2 - \frac{1}{(2c_p L||\mathbf{x}_0 - \mathbf{x}^\star||)^2}\langle \nabla f_{i_t}(\mathbf{x}_t), \mathbf{x}_t - \mathbf{x}^\star\rangle . \tag{B.31}$$

Plugging in $c_p = \frac{c_p^{\text{scale}}}{\sqrt{f_{i_0}(\mathbf{x}_0) - f^\star}}$, we get

$$||\mathbf{x}_{t+1} - \mathbf{x}^\star||^2 \leq ||\mathbf{x}_t - \mathbf{x}^\star||^2 - \frac{f_{i_0}(\mathbf{x}_0) - f^\star}{(2c_p^{\text{scale}} L||\mathbf{x}_0 - \mathbf{x}^\star||)^2}\langle \nabla f_{i_t}(\mathbf{x}_t), \mathbf{x}_t - \mathbf{x}^\star\rangle . \tag{B.32}$$

**case 1: $f$ is convex**.

For any $t \geq 1$, we take expectation conditional on $i_0$ on both sides and get:

$$\mathbb{E}[||\mathbf{x}_{t+1} - \mathbf{x}^\star||^2|i_0] \leq \mathbb{E}[||\mathbf{x}_t - \mathbf{x}^\star||^2|i_0] - \frac{f_{i_0}(\mathbf{x}_0) - f^\star}{(2c_p^{\text{scale}} L||\mathbf{x}_0 - \mathbf{x}^\star||)^2}\mathbb{E}[\langle \nabla f_{i_t}(\mathbf{x}_t), \mathbf{x}_t - \mathbf{x}^\star\rangle|i_0]$$

$$\overset{(\text{A.1})}{\leq} \mathbb{E}[||\mathbf{x}_t - \mathbf{x}^\star||^2|i_0] - \frac{f_{i_0}(\mathbf{x}_0) - f^\star}{(2c_p^{\text{scale}} L||\mathbf{x}_0 - \mathbf{x}^\star||)^2}\mathbb{E}[f(\mathbf{x}_t) - f^\star|i_0] . \tag{B.33}$$

Summing up from $t = 1$ to $t = T$ and dividing by $T$, we obtain:

$$\frac{1}{T}\sum_{t=1}^{T}\mathbb{E}[f(\mathbf{x}_t) - f^\star|i_0] \leq 4L(c_p^{\text{scale}})^2\frac{||\mathbf{x}_0 - \mathbf{x}^\star||^2}{f_{i_0}(\mathbf{x}_0) - f^\star}\frac{L\,\mathbb{E}[||\mathbf{x}_1 - \mathbf{x}^\star||^2|i_0]}{T} . \tag{B.34}$$

Note that $\mathbb{E}[||\mathbf{x}_1 - \mathbf{x}^\star||^2|i_0] = ||\mathbf{x}_1 - \mathbf{x}^\star||^2 \leq ||\mathbf{x}_0 - \mathbf{x}^\star||^2$ due to (B.31). We thus get:

$$\frac{1}{T}\sum_{t=1}^{T}\mathbb{E}[f(\mathbf{x}_t) - f^\star|i_0] \leq \left(4L(c_p^{\text{scale}})^2\frac{||\mathbf{x}_0 - \mathbf{x}^\star||^2}{f_{i_0}(\mathbf{x}_0) - f^\star}\right)\frac{L||\mathbf{x}_0 - \mathbf{x}^\star||^2}{T} \tag{B.35}$$

Taking expectation w.r.t $i_0$ on both sides, we arrive at:

$$\frac{1}{T}\sum_{t=1}^{T}\mathbb{E}[f(\mathbf{x}_t) - f^\star] \leq A\frac{L||\mathbf{x}_0 - \mathbf{x}^\star||^2}{T} \text{ with } A := 4L(c_p^{\text{scale}})^2\,\mathbb{E}_{i_0}\Big[\frac{||\mathbf{x}_0 - \mathbf{x}^\star||^2}{f_{i_0}(\mathbf{x}_0) - f^\star}\Big] . \tag{B.36}$$

**case 2: $f$ is strongly-convex**

For any $t \geq 1$, we take expectation conditional on $i_0$ on both sides of (B.32) and get:

$$\mathbb{E}[||\mathbf{x}_{t+1} - \mathbf{x}^\star||^2|i_0] \leq \mathbb{E}[||\mathbf{x}_t - \mathbf{x}^\star||^2|i_0] - \frac{f_{i_0}(\mathbf{x}_0) - f^\star}{(2c_p^{\text{scale}} L||\mathbf{x}_0 - \mathbf{x}^\star||)^2}\mathbb{E}[\langle \nabla f_{i_t}(\mathbf{x}_t), \mathbf{x}_t - \mathbf{x}^\star\rangle|i_0]$$

$$\overset{(\text{A.2})}{\leq} \mathbb{E}[||\mathbf{x}_t - \mathbf{x}^\star||^2|i_0] - \frac{f_{i_0}(\mathbf{x}_0) - f^\star}{(2c_p^{\text{scale}} L||\mathbf{x}_0 - \mathbf{x}^\star||)^2}\mathbb{E}[f(\mathbf{x}_t) - f^\star + \frac{\mu}{2}||\mathbf{x}_t - \mathbf{x}^\star||^2|i_0] . \tag{B.37}$$

Note that $f(\mathbf{x}_t) - f^\star \geq \frac{\mu}{2}||\mathbf{x}_t - \mathbf{x}^\star||^2$ due to strong convexity, we obtain:

$$\mathbb{E}[||\mathbf{x}_{t+1} - \mathbf{x}^\star||^2|i_0] \leq \left(1 - \frac{(f_{i_0}(\mathbf{x}_0) - f^\star)\mu}{(2c_p^{\text{scale}} L||\mathbf{x}_0 - \mathbf{x}^\star||)^2}\right)\mathbb{E}[||\mathbf{x}_t - \mathbf{x}^\star||^2|i_0] . \tag{B.38}$$

For any $T \geq 1$, unrolling gives:

$$\mathbb{E}[||\mathbf{x}_{T+1} - \mathbf{x}^\star||^2|i_0] \leq \left(1 - \frac{(f_{i_0}(\mathbf{x}_0) - f^\star)\mu}{(2c_p^{\text{scale}} L||\mathbf{x}_0 - \mathbf{x}^\star||)^2}\right)^T\mathbb{E}[||\mathbf{x}_1 - \mathbf{x}^\star||^2|i_0]$$

$$\leq \left(1 - \frac{(f_{i_0}(\mathbf{x}_0) - f^\star)\mu}{(2c_p^{\text{scale}} L||\mathbf{x}_0 - \mathbf{x}^\star||)^2}\right)^T||\mathbf{x}_0 - \mathbf{x}^\star||^2 . \tag{B.39}$$

The claim of Theorem 6 follows by taking expectation w.r.t $i_0$ on both sides.

**AdaSLS**: We only highlight the difference from AdaSPS. Plugging in the upper bound of $\eta_t$ from Lemma 16 and using the line-search condition, we get:

$$||\mathbf{x}_{t+1} - \mathbf{x}^\star||^2 \overset{(\text{A.17})}{\leq} ||\mathbf{x}_t - \mathbf{x}^\star||^2 - 2\eta_t\langle \nabla f_{i_t}(\mathbf{x}_t), \mathbf{x}_t - \mathbf{x}^\star\rangle + \eta_t\frac{\gamma_t||\nabla f_{i_t}(\mathbf{x}_t)||^2}{c_l\sqrt{\sum_{s=0}^{t}\gamma_s||\nabla f_{i_s}(\mathbf{x}_s)||^2}}$$

$$\leq ||\mathbf{x}_t - \mathbf{x}^\star||^2 - 2\eta_t\langle \nabla f_{i_t}(\mathbf{x}_t), \mathbf{x}_t - \mathbf{x}^\star\rangle + \eta_t\frac{f_{i_t}(\mathbf{x}_t) - f^\star}{c_l\rho\sqrt{\sum_{s=0}^{t}\gamma_s||\nabla f_{i_s}(\mathbf{x}_s)||^2}} . \tag{B.40}$$

Since $c_l\rho\sqrt{\sum_{s=0}^{t}\gamma_s||\nabla f_{i_s}(\mathbf{x}_s)||^2} \geq 1$, we obtain the same equation as (B.26). To find a lower bound of $\eta_{T-1}$, we rearrange (B.25) as:

$$\eta_t(f_{i_t}(\mathbf{x}_t) - f^\star) \leq ||\mathbf{x}_t - \mathbf{x}^\star||^2 - ||\mathbf{x}_{t+1} - \mathbf{x}^\star||^2 , \tag{B.41}$$

the left-hand-side of which can be lower bounded by:

$$\eta_t(f_{i_t}(\mathbf{x}_t) - f^\star) \geq \eta_t\rho\gamma_t||\nabla f_{i_t}(\mathbf{x}_t)||^2 \overset{(A.17)}{\geq} \min\{\frac{1-\rho}{L}, \gamma_{\max}\}\frac{\rho\gamma_t||\nabla f_{i_t}(\mathbf{x}_t)||^2}{c_l\sqrt{\sum_{s=0}^{t}\gamma_s||\nabla f_{i_s}(\mathbf{x}_s)||^2}} . \tag{B.42}$$

Summing over $t=0$ to $t = T-1$ gives:

$$\min\{\frac{1-\rho}{L}, \gamma_{\max}\}\frac{\rho}{c_l}\sqrt{\sum_{s=0}^{T-1}\gamma_s||\nabla f_{i_s}(\mathbf{x}_s)||^2} \overset{(A.6)}{\leq} \min\{\frac{1-\rho}{L}, \gamma_{\max}\}\frac{\rho}{c_l}\sum_{t=0}^{T-1}\frac{\gamma_t||\nabla f_{i_t}(\mathbf{x}_t)||^2}{\sqrt{\sum_{s=0}^{t}\gamma_s||\nabla f_{i_s}(\mathbf{x}_s)||^2}} \leq ||\mathbf{x}_0 - \mathbf{x}^\star||^2 . \tag{B.43}$$

This implies that:

$$\eta_{T-1} \overset{(A.17)}{\geq} \frac{\min\{\frac{1-\rho}{L}, \gamma_{\max}\}}{c_l\sqrt{\sum_{s=0}^{T-1}\gamma_s||\nabla f_{i_s}(\mathbf{x}_s)||^2}} \overset{(B.43)}{\geq} \frac{\rho\min^2\{\frac{1-\rho}{L}, \gamma_{\max}\}}{c_l^2||\mathbf{x}_0 - \mathbf{x}^\star||^2} . \tag{B.44}$$

After plugging the above into (B.26), the remaining proof follows from the same routine as shown for AdaSPS. $\square$

## Proofs for Loopless Variance-Reduction

## B.4 Statement and Proof of Lemma 18

The following Lemma provides us with the guarantee that as $\mathbf{w}_t, \mathbf{x}_t \to \mathbf{x}^\star$, $\mathbb{E}_{i_t}[F_{i_t}(\mathbf{x}_t) - F_{i_t}^\star] \to 0$, which implies diminishing variance.

**Lemma 18.** *Assume each $f_i$ is convex and $L$-smooth, for any $t \geq 0$, the iterates generated by Algorithm 1 satisfy:*

$$\mathbb{E}_{i_t}[F_{i_t}(\mathbf{x}_t) - F_{i_t}^\star] \leq f(\mathbf{x}_t) - f^\star + \frac{1}{2\mu_F}\mathbb{E}_{i_t}\left[||\nabla f_{i_t}(\mathbf{w}_t) - \nabla f_{i_t}(\mathbf{x}^\star)||^2\right] . \tag{B.45}$$

*Proof.* By definition of $F_{i_t}(\mathbf{x})$, we have:

$$\begin{aligned}
&F_{i_t}(\mathbf{x}_t) - F_{i_t}^\star \\
&= F_{i_t}(\mathbf{x}_t) - F_{i_t}(\mathbf{x}^\star) + F_{i_t}(\mathbf{x}^\star) - F_{i_t}^\star \\
&= f_{i_t}(\mathbf{x}_t) - f_{i_t}(\mathbf{x}^\star) + (\mathbf{x}_t - \mathbf{x}^\star)^T(\nabla f(\mathbf{w}_t) - \nabla f_{i_t}(\mathbf{w}_t)) - \frac{\mu_F}{2}||\mathbf{x}_t - \mathbf{x}^\star||^2 + F_{i_t}(\mathbf{x}^\star) - F_{i_t}^\star .
\end{aligned} \tag{B.46}$$

By $\mu_F$-strong convexity of $F_{i_t}(\mathbf{x})$, we obtain:

$$\begin{aligned}
F_{i_t}(\mathbf{x}^\star) - F_{i_t}^\star &\overset{(A.15)}{\leq} \frac{1}{2\mu_F}||\nabla F_{i_t}(\mathbf{x}^\star)||^2 \\
&= \frac{1}{2\mu_F}||\nabla f_{i_t}(\mathbf{x}^\star) - \nabla f_{i_t}(\mathbf{w}_t) + \nabla f(\mathbf{w}_t) + \mu_F(\mathbf{x}^\star - \mathbf{x}_t)||^2 .
\end{aligned} \tag{B.47}$$

Plugging (B.47) into (B.46), taking expectation w.r.t the randomness $i_t$ on both sides gives:

$$\begin{aligned}
&\mathbb{E}_{i_t}[F_{i_t}(\mathbf{x}_t) - F_{i_t}^\star] \\
&\leq f(\mathbf{x}_t) - f(\mathbf{x}^\star) - \frac{\mu_F}{2}||\mathbf{x}_t - \mathbf{x}^\star||^2 + \mathbb{E}_{i_t}[\frac{1}{2\mu_F}||\nabla f_{i_t}(\mathbf{x}^\star) - \nabla f_{i_t}(\mathbf{w}_t) + \nabla f(\mathbf{w}_t) + \mu_F(\mathbf{x}^\star - \mathbf{x}_t)||^2] \\
&= f(\mathbf{x}_t) - f(\mathbf{x}^\star) - \frac{\mu_F}{2}||\mathbf{x}_t - \mathbf{x}^\star||^2 + \frac{\mu_F}{2}||\mathbf{x}_t - \mathbf{x}^\star||^2 + \frac{1}{2\mu_F}\mathbb{E}_{i_t}[||\nabla f_{i_t}(\mathbf{x}^\star) - \nabla f_{i_t}(\mathbf{w}_t) + \nabla f(\mathbf{w}_t)||^2] \\
&\quad + \frac{1}{\mu_F}\mathbb{E}_{i_t}[\langle\nabla f_{i_t}(\mathbf{x}^\star) - \nabla f_{i_t}(\mathbf{w}_t) + \nabla f(\mathbf{w}_t), \mu_F(\mathbf{x}^\star - \mathbf{x}_t)\rangle] \\
&= f(\mathbf{x}_t) - f(\mathbf{x}^\star) + \frac{1}{2\mu_F}\mathbb{E}_{i_t}[||\nabla f_{i_t}(\mathbf{x}^\star) - \nabla f_{i_t}(\mathbf{w}_t) + \nabla f(\mathbf{w}_t)||^2] \\
&\leq f(\mathbf{x}_t) - f(\mathbf{x}^\star) + \frac{1}{2\mu_F}\mathbb{E}_{i_t}[||\nabla f_{i_t}(\mathbf{x}^\star) - \nabla f_{i_t}(\mathbf{w}_t)||^2] .
\end{aligned} \tag{B.48}$$

$\square$

## B.5 Proof of Theorem 7

*Proof.* Recall the proof technique that gives equation (B.6) and (B.14) in Theorem 1. Following the same routine, we arrive at:

$$\sum_{t=0}^{T-1} \langle \nabla F_{i_t}(\mathbf{x}_t), \mathbf{x}_t - \mathbf{x}^\star \rangle \leq \tau \sqrt{\sum_{t=0}^{T-1} F_{i_t}(\mathbf{x}_t) - F_{i_t}^\star} \; . \tag{B.49}$$

where $\tau = (2c_p(L + \mu_F)D^2 + \frac{1}{c_p})$ for AdaSVRPS and $\tau = \max\left\{\frac{L+\mu_F}{(1-\rho)\sqrt{\rho}}, \frac{1}{\gamma_{\max}\sqrt{\rho}}\right\}c_l D^2 + \frac{1}{c_l\sqrt{\rho}}$ for AdaSVRLS. The difference is due to the fact that $F_{i_t}(\mathbf{x})$ is $(L + \mu_F)$-smooth. Taking the expectation, using the fact that $\mathbb{E}[\nabla F_{i_t}(\mathbf{x}_t)] = \mathbb{E}[\nabla f_{i_t}(\mathbf{x}_t) + \nabla f(\mathbf{w}_t) - \nabla f_{i_t}(\mathbf{w}_t)] = \nabla f(\mathbf{x}_t)$ and applying Lemma 18, we end up with:

$$\sum_{t=0}^{T-1} \mathbb{E}[f(\mathbf{x}_t) - f^\star] \stackrel{(B.48)}{\leq} \tau \sqrt{\sum_{t=0}^{T-1} \mathbb{E}[f(\mathbf{x}_t) - f^\star] + \frac{1}{2\mu_F} \sum_{t=0}^{T-1} \mathbb{E}[||\nabla f_{i_t}(\mathbf{w}_t) - \nabla f_{i_t}(\mathbf{x}^\star)||^2]} \; . \tag{B.50}$$

Taking the square gives:

$$\left(\sum_{t=0}^{T-1} \mathbb{E}[f(\mathbf{x}_t) - f^\star]\right)^2 \leq \tau^2 \left(\sum_{t=0}^{T-1} \mathbb{E}[f(\mathbf{x}_t) - f^\star] + \frac{1}{2\mu_F} \sum_{t=0}^{T-1} \mathbb{E}[||\nabla f_{i_t}(\mathbf{w}_t) - \nabla f_{i_t}(\mathbf{x}^\star)||^2]\right) \; . \tag{B.51}$$

Define the Lyapunov function: $\mathcal{Z}_{t+1} = \frac{1}{2(1-a)} \frac{\tau^2}{p_{t+1}\mu_F} ||\nabla f_{i_{t+1}}(\mathbf{w}_{t+1}) - \nabla f_{i_{t+1}}(\mathbf{x}^\star)||^2$. It follows that:

$$\mathbb{E}[\mathcal{Z}_{t+1}] = \frac{1}{2(1-a)} \frac{\tau^2}{p_{t+1}\mu_F} \mathbb{E}[||\nabla f_{i_{t+1}}(\mathbf{w}_{t+1}) - \nabla f_{i_{t+1}}(\mathbf{x}^\star)||^2]$$

$$= \frac{\tau^2}{2(1-a)\mu_F} \mathbb{E}[||\nabla f_{i_{t+1}}(\mathbf{x}_t) - \nabla f_{i_{t+1}}(\mathbf{x}^\star)||^2] + \frac{1-p_{t+1}}{2(1-a)} \frac{\tau^2}{p_{t+1}\mu_F} \mathbb{E}[||\nabla f_{i_t}(\mathbf{w}_t) - \nabla f_{i_t}(\mathbf{x}^\star)||^2]$$

$$\stackrel{(A.5)}{\leq} \frac{\tau^2}{2(1-a)\mu_F} \mathbb{E}[2L(f_{i_{t+1}}(\mathbf{x}_t) - f_{i_{t+1}}(\mathbf{x}^\star) - \langle \nabla f_{i_{t+1}}(\mathbf{x}^\star), \mathbf{x}_t - \mathbf{x}^\star\rangle)] + \frac{(1-p_{t+1})p_t}{p_{t+1}} \mathbb{E}[\mathcal{Z}_t]$$

$$= \frac{L}{(1-a)\mu_F} \tau^2 (\mathbb{E}[f(\mathbf{x}_t) - f^\star]) + \frac{(1-p_{t+1})p_t}{p_{t+1}} \mathbb{E}[\mathcal{Z}_t] \; . \tag{B.52}$$

Adding $\sum_{t=0}^{T-1} \mathbb{E}[\mathcal{Z}_{t+1}]$ to both sides of (B.51) and substituting the above upper bound, we get:

$$\left(\sum_{t=0}^{T-1} \mathbb{E}[f(\mathbf{x}_t) - f^\star]\right)^2 + \sum_{t=0}^{T-1} \mathbb{E}[\mathcal{Z}_{t+1}] \leq (1 + \frac{L}{(1-a)\mu_F})\tau^2 \sum_{t=0}^{T-1} \mathbb{E}[f(\mathbf{x}_t) - f^\star]$$

$$+ \sum_{t=0}^{T-1}\left((1-a)p_t + \frac{(1-p_{t+1})p_t}{p_{t+1}}\right)\mathbb{E}[\mathcal{Z}_t] \; . \tag{B.53}$$

Rearranging and dropping $\mathbb{E}[\mathcal{Z}_T] \geq 0$ gives:

$$\left(\sum_{t=0}^{T-1} \mathbb{E}[f(\mathbf{x}_t) - f^\star]\right)^2 \leq (1 + \frac{L}{(1-a)\mu_F})\tau^2 \sum_{t=0}^{T-1} \mathbb{E}[f(\mathbf{x}_t) - f^\star] + \sum_{t=1}^{T-1}\left((1-a)p_t + \frac{(1-p_{t+1})p_t}{p_{t+1}} - 1\right)\mathbb{E}[\mathcal{Z}_t]$$

$$+ \left((1-a)p_0 + \frac{(1-p_1)p_0}{p_1}\right)\mathbb{E}[\mathcal{Z}_0] \; . \tag{B.54}$$

By our choice of $p_t$, we have:

$$(1-a)p_t + \frac{(1-p_{t+1})p_t}{p_{t+1}} - 1 = \frac{p_t}{p_{t+1}} - ap_t - 1 = \frac{at+a+1}{at+1} - \frac{a}{(at+1)} - 1 = \frac{0}{2(at+1)} = 0 \; . \tag{B.55}$$

Therefore, it holds that:

$$\left(\sum_{t=0}^{T-1} \mathbb{E}[f(\mathbf{x}_t) - f^\star]\right)^2 \leq (1 + \frac{L}{(1-a)\mu_F})\tau^2 \sum_{t=0}^{T-1} \mathbb{E}[f(\mathbf{x}_t) - f^\star] + \mathbb{E}[\mathcal{Z}_0] \; . \tag{B.56}$$

Further, by $L$-smoothness and convexity of $f$, we have

$$\mathbb{E}[\mathcal{Z}_0] = \frac{1}{2(1-a)} \frac{\tau^2}{p_0\mu_F} \mathbb{E}[||\nabla f_{i_0}(\mathbf{x}_0) - \nabla f_{i_0}(\mathbf{x}^\star)||^2] \stackrel{(A.5)}{\leq} \frac{L\tau^2}{(1-a)\mu_F}(f(\mathbf{x}_0) - f^\star) \; . \tag{B.57}$$

Hence. we obtain:

$$\left( \sum_{t=0}^{T-1} \mathbb{E}[f(\mathbf{x}_t) - f^\star] \right)^2 \leq (1 + \frac{2L}{(1-a)\mu_F})\tau^2 \sum_{t=0}^{T-1} \mathbb{E}[f(\mathbf{x}_t) - f^\star]. \tag{B.58}$$

It follows that:

$$\sum_{t=0}^{T-1} \mathbb{E}[f(\mathbf{x}_t) - f^\star] \leq (1 + \frac{2L}{(1-a)\mu_F})\tau^2. \tag{B.59}$$

Dividing both sides by $T$ and applying Jensen's inequality concludes the proof.

$\square$

## B.6 Proof of Corollary 8

*Proof.* Algorithm 1 calls the stochastic gradient oracle in expectation $\mathcal{O}(1 + p_t n)$ times at iteration $t$. Therefore, the total number of gradient evaluations is upper bounded by $\mathcal{O}(\sum_{t=0}^{T-1} p_t n + T)$. By our choice of $p_t$, it holds that $\sum_{t=0}^{T-1} p_t \leq \frac{1}{a} \sum_{t=0}^{T-1} \frac{1}{t+2} \leq \frac{1}{a}(\log(T) + 1 - 1) = \frac{1}{a}\log(T)$. Due to the sublinear convergence rate of Algorithm 1, we conclude that the total number of stochastic gradient calls is $\mathcal{O}(\log(1/\varepsilon)n + 1/\varepsilon)$. $\square$

## C  Pseudo-codes for AdaSPS and AdaSLS

In this section, we provide formal pseudo codes for AdaSPS (AdaSPS) and AdaSLS (AdaSLS).

To implement AdaSPS, a lower bound of optimal function value for each minibatch function is required. For machine learning problems where the individual loss functions are non-negative, we can use zero as an input. Apart from that, we need to provide a constant $c_p$ to adjust the magnitude of the stepsize. Theoretically suggested $c_p$ for robust convergence satisfies $c_p^{\text{scale}} = c_p\sqrt{f_{i_0}(\mathbf{x}_0) - \ell_{i_0}^\star} \geq \frac{1}{2}$. Therefore, a common choice is to set $c_p = \frac{1}{\sqrt{f_{i_0}(\mathbf{x}_0) - \ell_{i_0}^\star}}$.

---

**Algorithm 2** AdaSPS

---

**Require:** $\mathbf{x}_0 \in \mathbb{R}^d, T \in \mathbb{N}^+, c_p > 0$
1: set $\eta_{-1} = +\infty$
2: set $\varepsilon = 10^{-10}$
3: **for** $t = 0$ to $T - 1$ **do**
4:     uniformly sample $i_t \subseteq [n]$
5:     provide a lower bound $\ell_{i_t}^\star \leq f_{i_t}^\star$
6:     set $\eta_t = \min\left\{ \frac{f_{i_t}(\mathbf{x}_t) - \ell_{i_t}^\star}{c_p\|\nabla f_{i_t}(\mathbf{x}_t)\|^2} \frac{1}{\sqrt{\sum_{s=0}^t f_{i_s}(\mathbf{x}_s) - \ell_{i_s}^\star + \varepsilon}}, \eta_{t-1} \right\}$
7:     $\mathbf{x}_{t+1} = \Pi_\mathcal{X}(\mathbf{x}_t - \eta_t \nabla f_{i_t}(\mathbf{x}_t))$
    **return** $\bar{\mathbf{x}}_T = \frac{1}{T} \sum_{t=0}^{T-1} \mathbf{x}_t$

---

To implement AdaSLS (AdaSLS), a line-search sub-solver 4 and an input constant $c_l > 0$ are required. Similar to (AdaSPS), we can set $c_l = \frac{1}{\rho\sqrt{\gamma_0\|\nabla f_{i_0}(\mathbf{x}_0)\|^2}}$ according to the theory.

---

**Algorithm 3** AdaSLS

---

**Require:** $\mathbf{x}_0 \in \mathbb{R}^d, T \in \mathbb{N}^+, c_l > 0$
1: set $\eta_{-1} = +\infty$
2: set $\varepsilon = 10^{-10}$
3: **for** $t = 0$ to $T - 1$ **do**
4:     uniformly sample $i_t \subseteq [n]$
5:     obtain $\gamma_t$ via backtracking line-search (4)
6:     set $\eta_t = \min\left\{ \frac{\gamma_t}{c_l\sqrt{\sum_{s=0}^t \gamma_s\|\nabla f_{i_s}(\mathbf{x}_s)\|^2 + \varepsilon}}, \eta_{t-1} \right\}$
7:     $\mathbf{x}_{t+1} = \Pi_\mathcal{X}(\mathbf{x}_t - \eta_t \nabla f_{i_t}(\mathbf{x}_t))$
    **return** $\bar{\mathbf{x}}_T = \frac{1}{T} \sum_{t=0}^{T-1} \mathbf{x}_t$

---

# D  Line-search procedure

In this section, we introduce the classical Armijo line-search method [3, 41]. Given a function $f_{i_t}(\mathbf{x})$, the Armijo line-search returns a stepsize $\gamma_t$ that satisfies the following condition:

$$f_{i_t}(\mathbf{x}_t - \gamma_t \nabla f_{i_t}(\mathbf{x}_t)) \leq f_{i_t}(\mathbf{x}_t) - \rho \gamma_t ||\nabla f_{i_t}(\mathbf{x}_t)||^2 \,, \tag{D.1}$$

where $\rho \in (0, 1)$ is an input hyper-parameter. If $f_{i_t}(\mathbf{x})$ is a smooth function, then backtracking line-search 4 is a practical implementation way to ensure that D.1 is satisfied.

---

**Algorithm 4** Backtracking line-search

---

**Require:** $\beta \in [\frac{1}{2}, 1)$, $\rho \in (0, 1)$, $\gamma_{\max} > 0$ (We fix $\beta = 0.8$ and $\rho = 0.5$ for AdaSLS)
1: $\gamma = \gamma_{\max}$
2: **while** $f_{i_t}(\mathbf{x}_t - \gamma \nabla f_{i_t}(\mathbf{x}_t)) > f_{i_t}(\mathbf{x}_t) - \rho\gamma||\nabla f_{i_t}(\mathbf{x}_t)||^2$ **do**
3:     $\gamma = \beta\gamma$
   **return** $\gamma_t = \gamma$

---

To implement Algorithm 4, one needs to provide the decreasing factor $\beta$, the maximum stepsize $\gamma_{\max}$, and the condition parameter $\rho$. Starting from $\gamma_{\max}$, Algorithm 4 decreases the stepsize iteratively by a constant factor $\beta$ until the condition D.1 is satisfied. Note that checking the condition requires additional minibatch function value evaluations. Fortunately, note that the output $\gamma$ cannot be smaller than $\frac{1-\rho}{L}$ (Lemma 16), and thus the number of extra function value evaluations required is at most $\mathcal{O}\left(\max\{\log_{1/\beta}^{L\gamma_{\max}/(1-\rho)}, 1\}\right)$. In practice, Vaswani et al. [52] suggests dynamic initialization of $\gamma_{\max}$ to reduce the algorithm's running time, that is, setting $\gamma_{\max_t} = \gamma_{t-1}\theta^{1/n}$ where a common choice for $\theta$ is 2. This strategy initializes $\gamma_{\max}$ by a slightly larger number than the last output and thus is usually more efficient than keeping $\gamma_{\max}$ constant or always using $\gamma_{\max_t} = \gamma_{t-1}$. In all our experiments, we use the same $\gamma_{\max}$ at each iteration for AdaSLS to show its theoretical properties.

Goldstein line-search is another line-search method that checks D.1 and an additional curvature condition [41]. We do not study this method in this work and we refer to [52] for more details.

# E  Counter examples of SPS and its variants for SVRG

We provide two simple counterexamples where SVRG with the SPS stepsize and its intuitive variants fail to converge. For simplicity, consider the update rule $\mathbf{x}_{t+1} = \mathbf{x}_t - \eta_t \nabla f(\mathbf{x}_t)$, i.e. $\mathbf{w}_t = \mathbf{x}_t$ for all $t \geq 0$. Consider the function $f(\mathbf{x}) = \frac{f_1(\mathbf{x}) + f_2(\mathbf{x})}{2}$ where $f_1(\mathbf{x}) = a_1(\mathbf{x} - 1)^2$ and $f_2(\mathbf{x}) = a_2(\mathbf{x} + 1)^2$ with $a_1, a_2 > 0$.

**Example 19.** *Individual curvature is not representative. Consider the standard stochastic Polyak stepsize:* $\eta_t = \frac{f_i(\mathbf{x}_t) - f_i^\star}{||\nabla f_i(\mathbf{x}_t)||^2}$ *where $i$ is randomly chosen from $\{1, 2\}$. We now let $a_1 = 1$ and $a_2 < 1$. Note that* $\nabla^2 f(\mathbf{x}) = a_1 + a_2 \in (1, 2)$ *while $\mathbb{E}_i[\eta_t] = \frac{1}{8} + \frac{1}{8a_2} \to +\infty$ as $a_2 \to 0$, which leads to divergence. The reason behind this is that individual curvature does not match the global curvature.*

**Example 20.** *Mismatching quantity. Consider a variant of stochastic Polyak stepsize:* $\eta_t = \frac{f_i(\mathbf{x}_t) - f_i^\star}{||\nabla f_i(\mathbf{x}_t) - \nabla f_i(\mathbf{w}_t) + \nabla f(\mathbf{w}_t)||^2}$ *where $i$ is randomly chosen from $\{1, 2\}$. Let $a_1 = a_2 = 1$. We note* $\mathbb{E}_i[\eta_t \nabla f(\mathbf{x}_t)] = \frac{x_t^2 + 1}{2x_t} \neq 0$ *and thus no stationary point exists. Similar reasoning can exclude a number of other variants such as:* $\eta_t = \frac{f_i(\mathbf{x}_t) - f_i(\mathbf{w}_t) + f(\mathbf{w}_t) - f_i^\star}{||\nabla f_i(\mathbf{x}_t) - \nabla f_i(\mathbf{w}_t) + \nabla f(\mathbf{w}_t)||^2}$. *Indeed, the numerator is not the proper function value difference of a valid function with the gradient defined in the denominator.*

# F  Experimental details and additional experiment results

In this section, we provide a detailed setup of the experiments presented in the main paper.

In practice, we can use a lower bound of $F_{i_t}^\star$ for running AdaSVRPS since convergence is still guaranteed thanks to the property of AdaSPS. By default, we use $\ell_{i_t}^\star + \min_{\mathbf{x}}\{\mathbf{x}^T(\nabla f(\mathbf{w}_t) - \nabla f_{i_t}(\mathbf{w}_t)) + \frac{\mu_F}{2}||\mathbf{x} - \mathbf{x}_t||^2\}$ for all the experiments, where $\ell_{i_t}^\star$ is a lower bound for $f_{i_t}^\star$.

## F.1  Synthetic experiment

We consider the minimization of a quadratic of the form: $f(\mathbf{x}) = \frac{1}{n}\sum_{i=1}^n f_i(\mathbf{x})$ where $f_i(\mathbf{x}) = \frac{1}{2}(\mathbf{x} - \mathbf{b}_i)^T A_i(\mathbf{x} - \mathbf{b}_i)$, $\mathbf{b}_i \in \mathbb{R}^d$ and $A_i \in \mathbb{R}^{d \times d}$ is a diagonal matrix. We use $n = 50$, $d = 1000$. We control the interpolation by either setting all $\{b_i\}$ to be identical or different. Each component of $\{b_i\}$ is generated according

to $\mathcal{N}(0, 10^2)$. We control the complexity of the problems by choosing different matrices $A_i$. For the strongly-convex case, we first generate a matrix $A^N = \text{clip}(\begin{pmatrix} a_{11} & ... & a_{1d} \\ ... & & \\ a_{n1} & ... & a_{nd} \end{pmatrix}, 1, 10)$ where each $a_{ij} \sim N(0, 15^2)$ and the clipping operator clips the elements to the interval between 1 and 10. Then we compute:

$$A = \begin{pmatrix} 1 & 1 & \cdots & \frac{n}{\sum_{i=1}^n A_{i(d-1)}^N} & \frac{10n}{\sum_{i=1}^n A_{id}^N} \\ ... & & ... & & ... \\ 1 & 1 & \cdots & \frac{n}{\sum_{i=1}^n A_{i(d-1)}^N} & \frac{10n}{\sum_{i=1}^n A_{id}^N} \end{pmatrix} \odot A_N \ ,$$

where $\odot$ denotes the Hadamard product. We set the diagonal elements of each $A_i$ using the corresponding row stored in the matrix $A$. Note that $\nabla^2 f(\mathbf{x}) = \frac{1}{n}\sum_{i=1}^n A_i$ has the minimum and the largest eigenvalues being 1 and 10. For the general convex case, we use the same matrix $A_N$ to generate a sparse matrix $A_M$ such that $A_M = A_N \odot M$ where $M$ is a mask matrix with $M_{ij} \sim B(1, p)$ and $(1 \ ... \ 1) \cdot M_{:j} \geq 1, \ \forall j \in [1, d]$. We then compute the matrix $A$ and set each $A_i$ in the same way.

$$A = \begin{pmatrix} \frac{2^{-20}n}{\sum_{i=1}^n A_{i1}^M} & \frac{2^{-19}n}{\sum_{i=1}^n A_{i2}^M} & \cdots & \frac{2^{-1}n}{\sum_{i=1}^n A_{i20}^M} & 1 & \cdots & 1 & \frac{10n}{\sum_{i=1}^n A_{id}^M} \\ & & ... & ... & & ... & & ... \\ \frac{2^{-20}n}{\sum_{i=1}^n A_{i1}^M} & \frac{2^{-19}n}{\sum_{i=1}^n A_{i2}^M} & \cdots & \frac{2^{-1}n}{\sum_{i=1}^n A_{i20}^M} & 1 & \cdots & 1 & \frac{10n}{\sum_{i=1}^n A_{id}^M} \end{pmatrix} \odot A_M \ .$$

Through the construction, the smallest eigenvalues of $\nabla^2 f(\mathbf{x})$ are clustered around zero, and the largest eigenvalue is 10. Additionally, each $\nabla^2 f_i(\mathbf{x})$ is positive semi-definite.

We set the batch size to be 1 and thus we have $f_{i_t}^\star = 0$ with interpolation and $\ell_{i_t}^\star = 0$ without interpolation.

For AdaSPS/AdaSVRPS we fix $c_p^{\text{scale}} = 1$, and for AdaSVRPS we further use $\mu_F = 10$ and $p_t = \frac{1}{0.1t+1}$. We compare against DecSPS [44], SPS [34] and SVRG [22] and tune the stepsize for SVRG by picking the best one from $\{10^i\}_{i=-4,..,3}$.

In addition to these optimizers, we further evaluate the performance of SLS [52], AdaSLS, SGD, SPS$_{\text{max}}$ [34] and AdaSVRLS. We fix $c_l^{\text{scale}} = 1$, $\gamma_{\max} = 10$, $\beta = 0.8$ and $\rho = 1/2$ for both algorithms and for AdaSVRLS, we further use $\mu_F = 10$ and $p_t = \frac{1}{0.1t+1}$. For SGD, we use the best learning rate schedules in different scenarios. Specifically, for both interpolation problems, we keep the stepsize constant and for non-interpolation problems, we apply $\Theta(1/\sqrt{t})$ and $\Theta(1/t)$ decay schedules for convex and strongly-convex problems respectively. We further pick the best stepsize from $\{10^i\}_{i=-4,...,3}$. For SPS$_{\text{max}}$, we use $\gamma_b = 10^{-3}$ and we only showcase its performance in non-interpolated settings. We report the results in Figure F.1. We observe that AdaSLS is comparable if no faster than the best-tuned vanilla SGD. SPS$_{\text{max}}$ is reduced to the vanilla SGD with constant stepsize. AdaSVRLS provides similar performance to AdaSVRPS but due to the cost of additional function evaluations, it is less competitive than AdaSVRPS.

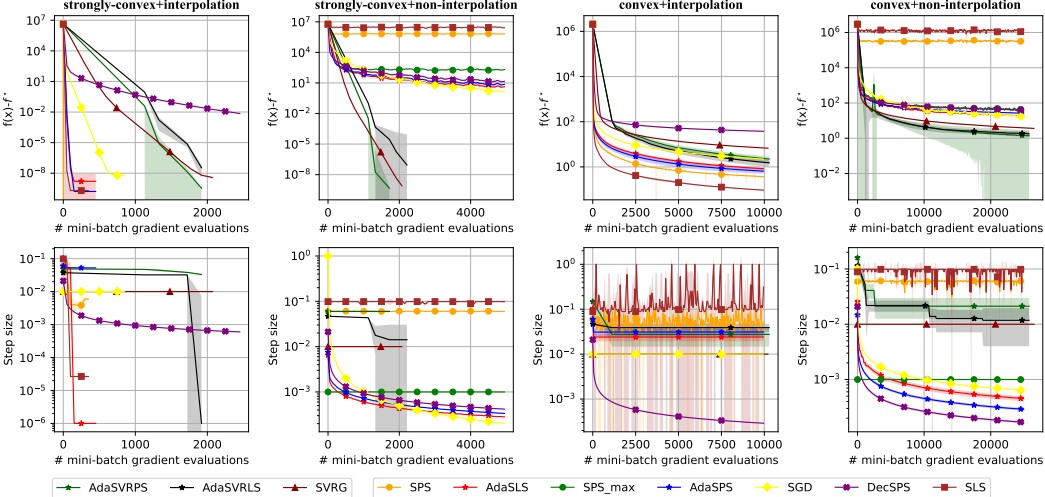

Figure F.1: Comparison of the considered optimizers on synthetic data set with quadratic loss. The left block of the label illustrates the variance-reduced methods and the right represents SGD with different stepsizes. (Repeated 3 times. The solid lines and the shaded area represent the mean and the standard deviation.)

| optimizers | hyper-parameters used for synthetic experiments | | | |
|---|---|---|---|---|
| | st-convex+ip | st-convex+non-ip | convex+ip | convex+non-ip |
| AdaSPS | $c_p^{\text{scale}} = 1$ | | | |
| AdaSLS | $c_l^{\text{scale}} = 1, \beta = 0.8, \rho = 0.5, \gamma_{\max} = 10$ | | | |
| SPS | $c = 0.5$ | | | |
| SPS$_{\max}$ | $c = 0.5, \gamma_b = 10^{-3}$ | | | |
| SLS | $\rho = 0.1, \beta = 0.9, \gamma_{\max} = 10$ | | | |
| DecSPS | $c_0 = 1, \gamma_b = 10$ | | | |
| SGD | constant, $\eta = 10^{-2}$ | $\mathcal{O}(1/t), \eta = 1$ | constant, $\eta = 10^{-2}$ | $\mathcal{O}(1/\sqrt{t}), \eta = 10^{-1}$ |
| AdaSVRPS | $c_p^{\text{scale}} = 1, \mu_F = 10, p_t = \frac{1}{0.1t+1}$ | | | |
| AdaSVRLS | $c_l^{\text{scale}} = 1, \beta = 0.8, \rho = 0.5, \gamma_{\max} = 0.1, \mu_F = 10, p_t = \frac{1}{0.1t+1}$ | | | |
| SVRG | $\eta = 10^{-2}$ | | | |

Table F.1: Hyper-parameters of the considered optimizers used in synthetic experiments. st-convex stands for strongly-convex and ip stands for interpolation.

## F.2 Binary classification

Following the binary classification experiments presented in the main paper, we provide additional experiments for VR algorithms. The chosen hyper-parameters for each algorithm can be found in Table F.3. In particular, we fix $c_l^{\text{scale}} = 1$, $\gamma_{\max} = 10^3$, $\beta = 0.8$ and $\rho = 1/2$ for AdaSLS and AdaSVRLS. We report the best $\mu_F \in \{10^{-4}, 10^2\}$. In Figure F.2, we observe that AdaSVRLS/AdaSVRPS provides similar performance to the other two variance-reduction methods. The details of the four considered datasets are summarized in Table F.2.

We next investigate the impact of the probability schedule on the convergence behaviours. We pick w8a as the dataset and run AdaSVRPS (Alg. 1) with and without probability decay. Specficially, we set $p_t = B/n$ and $p_t = \frac{1}{0.1t+1}$ to separate the cases. We control the level of the interpolation by using $B = 32$ and $B = 128$ since $\sigma_{f,128} \leq \sigma_{f,32}$. From Figure F.3, we observe that decreasing probability schedule is more efficient when the problem is more non-interpolated. This is because for interpolated problems, the frequent computation of the full gradients at the beginning provides no additional convergence benefits.

| | duke | rcv1 | ijcnn | w8a |
|---|---|---|---|---|
| n | 44 | 20242 | 49990 | 49749 |
| d | 7129 | 47236 | 22 | 300 |
| B | 1 | 64 | 64 | 128 |

Table F.2: Number of datapoints, dimension of features, used batch size of four datasets from LIBSVM [10]

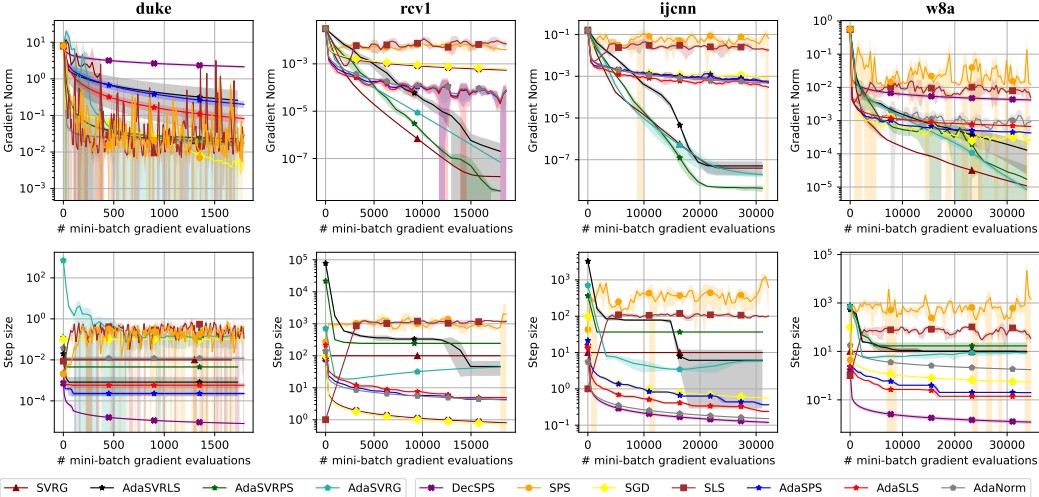

Figure F.2: Comparison of the considered optimizers on four LIBSVM datasets with regularized logistic loss. The left block of the label illustrates the variance-reduced methods and the right represents SGD with different stepsizes. (Repeated 3 times. The solid lines and the shaded area represent the mean and the standard deviation.)

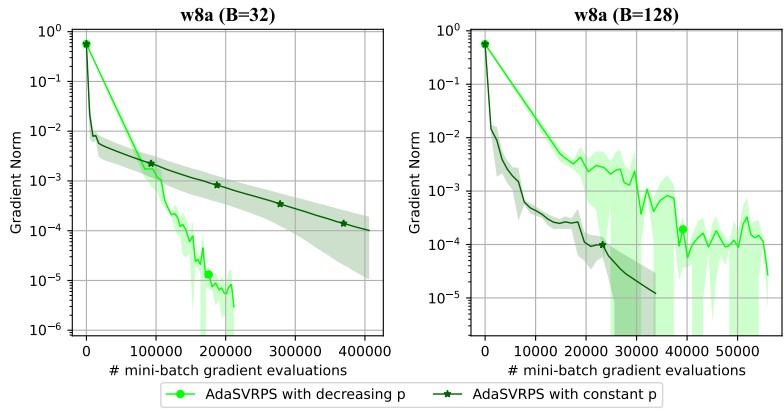

Figure F.3: Comparison of different probability schedules for AdaSVRPS on the w8a dataset with regularized logistic loss. Decreasing probability is more efficient when optimizing highly non-interpolated convex problems (Repeated 3 times. The solid lines and the shaded area represent the mean and the standard deviation.)

| optimizers | hyper-parameters used for binary classification tasks | | | | |
|---|---|---|---|---|---|
| | duke | rcv1 | ijcnn | w8a | for all |
| AdaSPS | | | | | $c_p^{\text{scale}} = 1$ |
| AdaSLS | $c_l^{\text{scale}} = 1, \gamma_{\max} = 10$ | $c_l^{\text{scale}} = 1, \gamma_{\max} = 10^3$ | $c_l^{\text{scale}} = 1, \gamma_{\max} = 10^3$ | $c_l^{\text{scale}} = 1, \gamma_{\max} = 10^3$ | $\beta = 0.8, \rho = 0.5$ |
| SPS | | | | | $c = 0.5$ |
| SLS | $\gamma_{\max} = 10$ | $\gamma_{\max} = 10^3$ | $\gamma_{\max} = 10^3$ | $\gamma_{\max} = 10^3$ | $\beta = 0.9, \rho = 0.1$ |
| DecSPS | $\gamma_b = 200$ | $\gamma_b = 100$ | $\gamma_b = 100$ | $\gamma_b = 100$ | $c_0 = 1$ |
| SGD | constant, $\eta = 10^{-1}$ | $\Theta(1/\sqrt{t}), \eta = 100$ | $\Theta(1/\sqrt{t}), \eta = 100$ | $\Theta(1/\sqrt{t}), \eta = 100$ | |
| AdaNorm | $c_g = 1$ | $c_g = 10$ | $c_g = 10$ | $c_g = 10$ | $b_0 = 10^{-10}$ |
| Adam | lr $= 10^{-3}$ | lr $= 10^{-2}$ | lr $= 10^{-2}$ | lr $= 10^{-2}$ | $\beta_1 = 0.9, \beta_2 = 0.999$ |
| AdaSVRPS | $c_p^{\text{scale}} = 1, \mu_F = 100$ | $c_p^{\text{scale}} = 1, \mu_F = 10^{-4}$ | $c_p^{\text{scale}} = 1, \mu_F = 10^{-4}$ | $c_p^{\text{scale}} = 1, \mu_F = 10^{-4}$ | $p_t = \frac{B}{N}$ |
| AdaSVRLS[a] | $c_l^{\text{scale}} = 1, \mu_F = 100$ | $c_l^{\text{scale}} = 1, \mu_F = 10^{-4}$ | $c_l^{\text{scale}} = 1, \mu_F = 10^{-4}$ | $c_l^{\text{scale}} = 1, \mu_F = 10^{-4}$ | $\beta = 0.8, \rho = 0.5, p_t = \frac{B}{N}$ |
| SVRG | $\eta = 10^{-2}$ | $\eta = 100$ | $\eta = 10$ | $\eta = 10$ | |
| AdaSVRG | | We use the heuristic method provided in Section 5 from [14]. | | | |

[a] $\gamma_{\max} = \frac{1}{\mu_F}$

Table F.3: Hyper-parameters of the considered optimizers used in binary classification.

# G Deep learning task

In this section, we provide a heuristic extension of AdaSPS to over-parameterized non-convex optimization tasks. When training modern deep learning models, Loshchilov and Hutter [35] observe that a cyclic behaviour of the stepsize, i.e., increasing at the beginning and then decreasing up to a constant, can help with fast training and good generalization performance. Since AdaSPS is a non-increasing stepsize, it excludes such a cyclic behaviour. To address this issue, we provide a non-convex version of AdaSPS which incorporates a restart mechanism that allows an increase of the stepsize according to the local curvature. The full algorithm is summarized in Algorithm 5. In practice, we can set $u = \frac{B}{n}$. Algorithm 5 updates the stepsize and $c_p$ at the beginning of each epoch and uses AdaSPS (AdaSPS) for the rest of the epoch.

Following [34, 52], we benchmark the convergence and generalization performance of AdaSPS (DL) 5 for the multi-class classification tasks on CIFAR10 [28] and CIFAR100 [29] datasets using ResNet-34 [21]. We compare against SPS [34], Adam [26], AdaGrad [15], DecSPS [44] and SGD with momentum. We use the smoothing technique and pick $c = 0.02$ for SPS as suggested in [34]. We use the official implementations of Adam, AdaGrad, and SGD with momentum from https://pytorch.org/docs/stable/optim.html. We choose lr $= 10^{-3}$, $\beta_1 = 0.9$ and $\beta_2 = 0.999$ for Adam. We choose lr $= 0.01$ for AdaGrad. We choose lr $= 0.01$ and $\beta = 0.9$ for SGD with momentum. Finally, we pick $c_p^{\text{scale}} = 0.02$ for Algorithm 5. In Figure G.1, AdaSPS (DL) shows competitive performance on both datasets. We leave the study of its theoretical properties to future work.

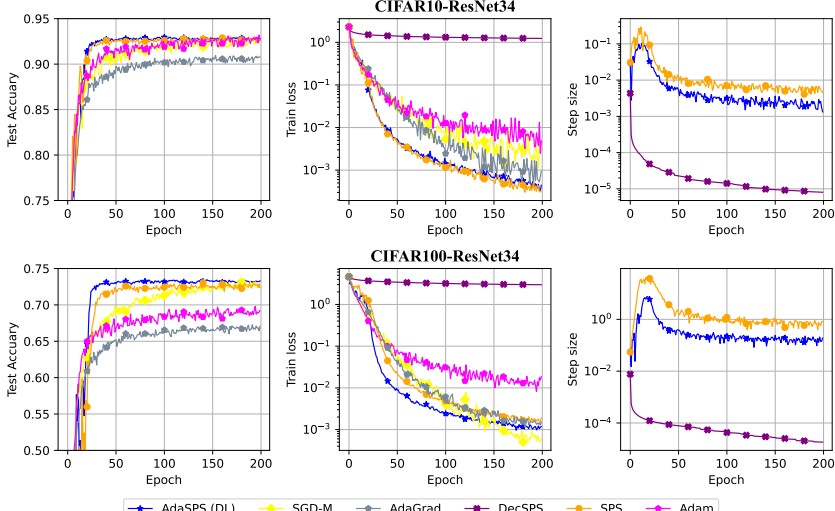

Figure G.1: Comparison of the considered optimizers on multi-class classification tasks with CIFAR10 and CIFAR100 datasets using ResNet34 with softmax loss. AdaSPS (DL) 5 and SPS provide remarkable performance on both datasets.

| optimizers | hyper-parameters used for multi-classification tasks |
|---|---|
| AdaSPS (DS) | $c_p^{\text{scale}} = 0.2$ |
| SPS | $c = 0.2$ + smoothing technique [34] |
| DecSPS | $c_0 = 1, \gamma_b = 1000$ |
| SGD-M | lr $= 0.01, \beta = 0.9$ |
| AdaGrad | lr $= 0.01$ |
| Adam | lr $= 10^{-3}, \beta_1 = 0.9, \beta_2 = 0.999$ |

Table G.1: Hyper-parameters of the considered optimizers used in multi-classification tasks.

---

**Algorithm 5** AdaSPS (DL)

---

**Require:** $\mathbf{x}_0 \in \mathbb{R}^d$, $T \in \mathbb{N}^+$, $c_p^{\text{scale}} > 0$, update frequency $u \in \mathbb{N}^+$
1: set $\eta_{-1} = +\infty$
2: set $\varepsilon = 10^{-10}$
3: **for** $t = 0$ to $T - 1$ **do**
4:      uniformly sample $i_t \subseteq [n]$
5:      provide a lower bound $\ell_{i_t}^\star \leq f_{i_t}^\star$
6:      **if** $t \bmod u$ is 0 **then**
7:          set $c_p = \dfrac{c_p^{\text{scale}}}{\sqrt{\sum_{s=0}^t f_{i_s}(\mathbf{x}_s) - \ell_{i_s}^\star}}$
8:          set $\eta_t = \dfrac{f_{i_t}(\mathbf{x}_t) - \ell_{i_t}^\star}{c_p \lVert \nabla f_{i_t}(\mathbf{x}_t) \rVert^2} \dfrac{1}{\sqrt{\sum_{s=0}^t f_{i_s}(\mathbf{x}_s) - \ell_{i_s}^\star} + \varepsilon}$
9:      **else**
10:         set $\eta_t = \min\left\{\dfrac{f_{i_t}(\mathbf{x}_t) - \ell_{i_t}^\star}{c_p \lVert \nabla f_{i_t}(\mathbf{x}_t) \rVert^2} \dfrac{1}{\sqrt{\sum_{s=0}^t f_{i_s}(\mathbf{x}_s) - \ell_{i_s}^\star} + \varepsilon}, \eta_{t-1}\right\}$
11:      $\mathbf{x}_{t+1} = \mathbf{x}_t - \eta_t \nabla f_{i_t}(\mathbf{x}_t)$
     **return** $\mathbf{x}_T$

---

