# OpenReview forum: "Adaptive SGD with Polyak stepsize and Line-search: Robust Convergence and Variance Reduction"
_NeurIPS.cc/2023/Conference — NeurIPS 2023 poster_

### Official Review · Reviewer_Dgvs · 2023-06-22

**Soundness:** 2 fair
**Presentation:** 2 fair
**Contribution:** 1 poor
**Rating:** 2
**Confidence:** 4

**Summary:**

This paper presents two step-sizes, AdaSPS and AdaSLS, and theoretical analyses of PSGD with AdaSPS/AdaSLS. It also presents numerical results to support the analyses. The contribution of the paper is to provide AdaSPS and AdaSLS step-sizes.

**Strengths:**

The strength of the paper is to provide two step-sizes, AdaSPS and AdaSLS, and to apply them to PSGD. In practice, the modifications, AdaSVRPS and AdaSVRLS, accelerate the existing methods.

**Weaknesses:**

- This paper considers only convex optimization, although optimization problems in deep learning are nonconvex.
- There seem to be some doubts in Lemmas and Theorems. I would like to have solutions for the doubts (Please see Questions).
- Numerical results are insufficient since it does not consider optimization problems in deep neural networks.

**Questions:**

- AdaSLS and Lemma 17 (AdaSPS_main+appendix.pdf): Using the Armijo condition (4) in AdaSLS is interesting. Here, I check Lemma 17 (AdaSPS_main+appendix.pdf). However, I doubt that there exists a lower bound $(1-\rho)/L$ of the step-size $\gamma_t$ satisfying (4). In  fact, we can provide counter-examples such that the step-size $\gamma_t$ satisfying (4) does not have any lower bounds. Consider $f(x) = x^2$ and apply it to the Armijo condition (4). Then, we can set a sufficiently small $\gamma_t$.
- Armijo condition (4): Related to the above comment, Figure 5 in [32] indicates that there is a possibility such that $\gamma_t$ satisfying (4) converges to 0. Hence, we have counter-examples of Lemma 17 such that there does not exist a lower bound of the step-size $\gamma_t$ satisfying (4).
- AdaSPS: AdaSPS uses $\ell_{i_t}^\star$. It would be unrealistic in deep learning. Can the authors provide some practical examples of $\ell_{i_t}^\star$ and $\eta_t$ defined by AdaSPS in deep learning?
- Assumptions: Can the authors provide some examples of satisfying the interpolation condition?
- Convex optimization: Since optimization problems in deep learning are nonconvex, considering only convex optimization is limited in machine learning society. Can the authors rebut this?
- Can the authors provide numerical results for training DNNs on the benchmark datasets (CIFAR-100 and ImageNet)? Using only LIVSBM datasets is insufficient to show usefulness of the proposed algorithms.

**Limitations:**

There is no potential negative societal impact.

---

> ### Author Rebuttal · Authors · 2023-08-09
>
> We thank the reviewer for the remarks and criticisms. We are a bit puzzled by your score. We provide clear answers to your questions below.
>
> Although we do not claim any contribution to the deep learning scenarios, we do provide DL experiments with our practical version of AdaSPS. Please see Appendix G for more details. In Figure G.1, we give results of multi-class classification tasks with CIFAR10 and CIFAR100 using ResNet34 with softmax loss. We compared our stepsize against the popular algorithms including Adam, SGD-M, AdaGrad, and SPS, **among which our proposed AdaSPS gives the best performance**.
>
> We next provide answers to your questions.
>
> > Q1. line-search: lower bound
>
> We kindly disagree. The lower bound should be $\min(\frac{1-\rho}{L},\gamma_{\max})$. We guess you missed the $\min$ operator. (See also Lemma 1 in your mentioned paper). We also provide concrete proof in Lemma 17 in the appendix.
>
> We next answer your second question on why the stepsize in Figure 5 may diminish.
>
> 1. The implementation of SLS shown in Figure 5 uses dynamic initialization of $\gamma_{\max}$ to reduce the algorithm's running time, that is, setting $\gamma_{\max t}=\gamma_{t-1}\theta^{1/n}$ where a common choice for $\theta$ is 2 and $\gamma_{t-1}$ is the previous stepsize which reflects the curvature. Therefore $\gamma_{\max}$ is not fixed anymore.
>
> 2. Note that in ResNet 32, more than 80 percent of the parameters (denoted by $x$) are scale invariant, that is $f(x)=f(cx)$ for any $c>0$. This implies that $||\nabla^2 f(x)||=\frac{1}{||x^2||}||\nabla f^2(\frac{x}{||x||})||$ which is unbounded from above and thus $f(x)$ is **not** smooth. See more discussions in [1].
>
> Combining 1) and 2), we can clearly see that the iterates generated by SLS may converge to some minimizer with locally increasing curvature and therefore the stepsize diminishes (which is the inverse of the curvature).
>
> To sum up, for non-smooth functions, the lower bound might not exist. However, in our work, we consider smooth functions.
>
> > Q2. $\ell_{i_t}^\star$ in deep learning
>
> We kindly disagree with the comment *AdaSPS uses $\ell_{i_t}^\star$ It would be unrealistic in deep learning.*
> First of all, $\ell_{i_t}^\star$ is much more readily available than $f_{i_t}^\star$ and this is already an improvement over SPS. Second of all,  in most deep learning tasks, the loss functions are always lower bounded by zero, i.e. $\ell_{i_t}^\star=0$. The examples include but are not limited to:
> 1. Regression: Mean Square Error, Mean Absolute Error, Huber loss and  log-cosh loss,
> 2. Classification: Cross-entropy loss, Hinge loss, etc.
>
> These losses are commonly used in machine learning and deep learning tasks.
>
> > Q3. Examples satisfying the interpolation condition
>
> If the machine learning model  is highly expressive and can fit the training dataset completely, then normally the interpolation condition holds. For convex problems, a classical example would be binary classification using RBF kernels without regularization and with logistic loss. For non-convex problems, training an overparameterized neural network is a classical example.
>
> > Q4. Since deep learning is non-convex, Why do we still study convex problems?
>
> We thank the author to raise this question.
> First, the development of the 'appropriate' optimization theory for deep learning models remains an elusive challenge. These complicated models are often non-convex, non-smooth, and non-differentiable at their minimizes (e.g. relu activation and normalization). Traditional analysis  investigates convergence towards stationary points with zero gradient norms. However, for deep learning, this conventional framework might not be useful. Therefore, the exploration of optimization theory for deep learning models remains an open question.
> Second, solving open questions in stochastic convex optimization boosts theoretical understanding and scientific development.
> Machine learning is not only about deep learning. Many machine learning problems are indeed convex such as SVM, logistic regression, and linear regression. Even for these simple models, there still exist tricky problems that we cannot solve yet. Providing solutions or insights into these problems will help us understand more complicated models and this is essentially how science should proceed.
>
> > Q5. Numerical results for training DNNs
>
> As mentioned earlier, we provide solid evidence in Appendix G that our proposed stepsize works well on deep learning tasks. Therefore, we think it is a promising stepsize that needs further study in the future.
>
> > Remark
>
> It is important and crucial to fundamentally understand the behavior of adaptive algorithms in st-convex and convex functions before we advance it to a more complicated case. We would like to highlight the contributions in our work. First, we propose two adaptive stepsizes which have both strong theoretical guarantees and practical performance, under the weakest assumptions, which improves upon some of the drawbacks of the previous methods. Essentially, the user can reliably apply our methods to the problems, without the need to know problem-dependent parameters and the underlying interpolation conditions that are often difficult to assess. This serves as a good step towards a totally automatic adaptive stepsize. Besides, they also show strong performance in deep learning experiments. Secondly, Polyak and line-search type methods all fail to be combined with the classical VR framework. However, we break this limitation and manage to accelerate these two complicated stepsizes. We believe our newly proposed framework may encourage more personalized VR techniques in the future.
>
> If you agree that we managed to address all issues, please consider raising your mark. If you believe this is not the case, please let us know so that we have a chance to respond. We really appreciate that.
>
> [1] Robust Training of Neural Networks Using Scale Invariant Architectures, ICML 2022

---

> > ### Comment · Reviewer_Dgvs · 2023-08-10
> > **Q1. line-search: lower bound**
> >
> > Thank you for your replies. I do not understand your replies for the lower bound of step-size. Let us consider $f(x) = x^2$, $\rho = 0.1$, and $\gamma_\max = 1$. Then, $L=2$ and $\min( \frac{1-\rho}{L}, \gamma_\max  ) = \min( 0.45 ,1 )  = 0.45$.
> > The Armijo condition $f(x - \gamma f'(x)) \leq f(x) - \rho \gamma |f'(x)|^2$ implies that $\gamma \leq 1 - \rho = 0.9$, where $x$ is not the global minimizer of $f$.
> > Hence, $\gamma$ satisfying the Armijo condition is, for example, $\gamma = 0.1$. Moreover, $0.1 < \min( \frac{1-\rho}{L}, \gamma_\max  ) = 0.45$.
> > Even if $\gamma_\max = 10^{-10}$, $\gamma = 10^{-11} \leq  1 - \rho = 0.9$ satisfies the Armijo condition.
> > Accordingly, a step-size $\gamma$ satisfying the Armijo condition is not always lower bounded by $\min( \frac{1-\rho}{L}, \gamma_\max  )$ (L528).

---

> > > ### Author Response · Authors · 2023-08-10
> > > **Q1. line-search: lower bound**
> > >
> > > We thank the reviewer for the reply.
> > >
> > > We would like to refer to Algorithm 4 for a more detailed procedure of Armijo line-search.
> > >
> > > If we run Armijo line-search, then we need to provide $\gamma_{\max}$.
> > > We first initialize $\gamma$ as $\gamma_{\max}$ (line 1), then we check if $\gamma$ satisfies the line-search condition (line 2). If not, then we decrease $\gamma$ by a factor of $\beta$ and repeat.
> > >
> > > Note we assume $\beta\in[0.5,1)$. In your example, if you initialize $\gamma_{\max}=1$, then the returned $\gamma$ should be at least 0.5 which is larger than $\min(\frac{1-\rho}{L},\gamma_{\max})$=0.45.
> > >
> > > We would like to highlight that the $\gamma_{t}$ in equation (4) is returned by Armijo line-search procedure rather than an arbitrary number that satisfies Armijo condition.
> > >
> > > We hope this solves your concern.
> > >
> > > Thank you

---

> > > > ### Comment · Reviewer_Dgvs · 2023-08-10
> > > > **Q1. line-search: lower bound**
> > > >
> > > > Thank you for your reply. I doubt your claim such that $\gamma$ is lower bounded by $\min(\frac{1-\rho}{L}, \gamma_\max)$ since the lower bound does not depend on $\beta$. I have believed that the lower bound depends on $\beta$ since I reviewed the paper. Please revise the proof of Lemma 17 and the results of Lemmas and Theorems adequately. At least, the results based on the lower bound $\min(\frac{1-\rho}{L}, \gamma_\max)$ are incorrect.

---

> > > > > ### Author Response · Authors · 2023-08-10
> > > > > **Q1. line-search: lower bound**
> > > > >
> > > > > The lower bound including $\beta$ should be $\min(\frac{2\beta(1-\rho)}{L},\gamma_{\max}). $
> > > > > In Algorithm 4, we clearly state that $\beta\in[0.5,1)$. Therefore the lower bound is at least $\min(\frac{1-\rho}{L},\gamma_{\max})$,
> > > > >
> > > > > **Hence, our lower bound is correct.**
> > > > >
> > > > > We will make it clear in the manuscript.
> > > > > Thank you

---

> > > > > > ### Comment · Reviewer_Dgvs · 2023-08-11
> > > > > > **Q1. line-search: lower bound**
> > > > > >
> > > > > > Could you prove that the lower bound is $\min (\frac{2 \beta (1-\rho)}{L}, \gamma_\max)$? It does not seem that the proof of Lemma 17 implies that the lower bound is $\min (\frac{2 \beta (1-\rho)}{L}, \gamma_\max)$. Moreover, I think that the lower bound is replaced with $\min (\frac{2 \beta (1-\rho)}{L}, \gamma_\max)$ not $\min (\frac{1-\rho}{L}, \gamma_\max)$. Is there a reason why the lower bound should not depend on $\beta$?

---

> > > > > > > ### Author Response · Authors · 2023-08-11
> > > > > > > **Q1. line-search: lower bound**
> > > > > > >
> > > > > > > > The proof for $\min(\frac{2\beta(1-\rho)}{L},\gamma_{\max})$.
> > > > > > >
> > > > > > > As shown in line 525, for any $\gamma\in(0,\frac{2(1-\rho)}{L}]$, the line-search condition is satisfied. If $\gamma_{\max}\le\frac{2\beta(1-\rho)}{L}$, then $\gamma=\gamma_{\max}$ is accepted. Otherwise, consider the extreme case where $\gamma_{\max}$ is arbitrarily close to and larger than $\frac{2(1-\rho)}{L}$. Then $\gamma=\gamma_{\max}$ is rejected but the next $\gamma=\beta\gamma_{\max}$ is accepted. Hence, $\gamma \ge \min(\frac{2\beta(1-\rho)}{L},\gamma_{\max})$.
> > > > > > >
> > > > > > > > Why not including $\beta$ in the lower bound?
> > > > > > > 1. $\beta$ is a decreasing factor which in practice is often set to $[0.5,1)$ (because decreasing too fast might miss a larger stepsize). For instance we set $\beta=0.8$ for AdaSLS. (See Algorithm 4). Therefore, the lower bound with any $\beta\in[0.5,1)$ is always between $\min(\frac{(1-\rho)}{L},\gamma_{\max})$ and $\min(\frac{2(1-\rho)}{L},\gamma_{\max})$. **In other words, the influence of $\beta$ on the lower bound is merely by a factor of constant 2.**
> > > > > > > 2. Following point 1, using $\min(\frac{(1-\rho)}{L},\gamma_{\max})$ makes our convergence results **much cleaner without losing anything.**
> > > > > > > 3. This is a convention that is widely used in optimization literature. For instance, see Lemma 1 in [1] where they do not include this dependence in their results for conciseness.
> > > > > > >
> > > > > > > We will make it clear in the manuscript.
> > > > > > >
> > > > > > > [1] Painless Stochastic Gradient: Interpolation, Line-Search, and Convergence Rates, Neurips 2019.

---

> > > > > > > > ### Comment · Reviewer_Dgvs · 2023-08-11
> > > > > > > > **Q1. line-search: lower bound**
> > > > > > > >
> > > > > > > > Thank you for your valuable comments. I understand almost of your claims. However, I still think that the lower bound depends on $\beta$ since step-sizes are computed by using backtracking method with $\beta$. I also check Lemma 1 in [1]. However, it would be incorrect. Please see Page 15 in [1] (https://arxiv.org/pdf/1905.09997.pdf) and the proof of Lemma 1, in particular, $c \eta_k \geq \eta_k - \frac{L_{ik} \eta_k^2}{2}$, which does not hold in general (I gave counter-example ($f(x) = x^2$) of Lemma 1 in [1] (10 Aug 2023)).
> > > > > > > > Therefore, the following proposition would be correct.
> > > > > > > > **Proposition:**
> > > > > > > > If a step-size $\gamma$ satisfying the Armijo condition is computed by backtracking method, then $\frac{2 \beta (1-\rho)}{L}  \leq \gamma$.

---

> > > > > > > > > ### Author Response · Authors · 2023-08-11
> > > > > > > > > **Q1. line-search: lower bound**
> > > > > > > > >
> > > > > > > > > We would like to highlight again that we assume $\beta\in[0.5,1)$ in our Algorithm 4 and hence our bound is correct.
> > > > > > > > > The dependence of $\beta$ on the convergence rate is not important.
> > > > > > > > >
> > > > > > > > > But we also agree with your comment and appreciate your persistence.
> > > > > > > > > We will add a lower bound that depends on arbitrary $\beta$ in the appendix.
> > > > > > > > >
> > > > > > > > > (PS: in Lemma1 of [1], they do not include the dependence of $\beta$, and thus their proof is still correct. They also mentioned in the last line of section 3.1 that the stepsize will be smaller by at most a factor of $\beta$ if they include this dependence (which is our result).)

---

> > ### Comment · Reviewer_Dgvs · 2023-08-11
> > **Q3. Examples satisfying the interpolation condition**
> >
> > Thank you again for your replies. I might misunderstand the definition of the interpolation condition. Could you provide the definition of the interpolation condition? (Probably, there is no definition of the interpolation condition in the paper)

---

> > > ### Author Response · Authors · 2023-08-11
> > > **Q3. Examples satisfying the interpolation condition**
> > >
> > > **The interpolation condition is clearly presented on line 122.**
> > >
> > > We say a problem is interpolated if $\sigma_{f,1}^2=f^\star-\mathbb{E}[f_{i}^\star]=f^\star-\frac{1}{n}\sum_{i=1}^n f_{i}^\star=0$.
> > >
> > > To make it extra clear, from the definition of interpolation: $\frac{1}{n}\sum_{i=1}^n f_{i}^\star=f(x^\star)=\frac{1}{n}\sum_{i=1}^n f_i(x^\star)$ where $x^\star$ is a minimizer of $f$, we can deduce $f_i(x^\star)=f_i^\star$ for any $i\in[n]$ since $f_i(x^\star)\ge f_i^\star$.
> > >
> > > In other words, interpolation means the global minimizer of $f$ is also a minimizer of each individual function $f_i$.
> > >
> > > Thank you

---

> > > > ### Comment · Reviewer_Dgvs · 2023-08-11
> > > > **Q3. Examples satisfying the interpolation condition**
> > > >
> > > > Thank you again for your comments. I understand the definition of the interpolation. I think that, under the interpolation condition, it is sufficient to consider minimizing only a certain $f_i$. Is this correct?

---

> > > > > ### Author Response · Authors · 2023-08-11
> > > > > **Q3. Examples satisfying the interpolation condition**
> > > > >
> > > > > No. Since each $f_i$ can have a set of minimizers, it is not sufficient to consider minimizing only a certain $f_i$. Thanks.

---

> > > > > > ### Comment · Reviewer_Dgvs · 2023-08-11
> > > > > > **Q3. Examples satisfying the interpolation condition**
> > > > > >
> > > > > > Sorry for my insufficient question. I think that, under the interpolation condition, it is sufficient to consider minimizing only a certain **strongly convex/convex** $f_i$ (Theorems 1 and 5). Is this correct? If it is not correct, it would be grateful that the authors could provide some examples such that we must minimize $f$ not $f_i$ under the interpolation and strongly convex/convex optimization.
> > > > > >
> > > > > > Thank you again for your replies.

---

> > > > > > > ### Author Response · Authors · 2023-08-11
> > > > > > > **Q3. Examples satisfying the interpolation condition**
> > > > > > >
> > > > > > > Thank you for your question. Suppose $f_1(x)=(x_1-1)^2$ and $f_2(x)=(x_2+1)^2$ where $x=(x_1,x_2)$. Then $f(x)=\frac{1}{2}[(x_1-1)^2+(x_2+1)^2]$ is strongly-convex. Note $f_1$ and $f_2$ are both convex and thus minimizing any of them alone is not sufficient to guarantee convergence to the global minimizer $x^\star=(1,-1)$.
> > > > > > >
> > > > > > > (PS: we have relaxed the assumption in Theorem 5 to individual convexity. See attached pdf.)

---

> > > > > > > > ### Comment · Reviewer_Dgvs · 2023-08-11
> > > > > > > > **Q3. Examples satisfying the interpolation condition**
> > > > > > > >
> > > > > > > > Thank you. It would be helpful if you could provide some practical examples in machine learning under strongly convex/convex optimization. The example that you gave is easy to understand, however, I would like to see practical examples in ML.

---

> > > > > > > > > ### Author Response · Authors · 2023-08-11
> > > > > > > > > **Q3. Examples satisfying the interpolation condition**
> > > > > > > > >
> > > > > > > > > First, our simple example is to address your concern ***I think that, under the interpolation condition, it is sufficient to consider minimizing only a certain strongly convex/convex.***
> > > > > > > > >
> > > > > > > > > Second, we have already provided in the answer for Q3 that binary classification using RBF kernels with logistic loss is a convex model. If the kernel bandwidths is selected properly, then this model is interpolated on some dataset such as mushroom from LIBSVM.  If we further add a regularization term to the loss, then the model is non-interpolated. Our algorithms work well in both cases.
> > > > > > > > >
> > > > > > > > > See also section 7.3 in [1] for more details.
> > > > > > > > >
> > > > > > > > > [1] Painless Stochastic Gradient: Interpolation, Line-Search, and Convergence Rates, Neurips 2019.

---

> > ### Comment · Reviewer_Dgvs · 2023-08-11
> > **Q5. Numerical results for training DNNs**
> >
> > Thank you again for your comments. I check Appendix G. Checking https://github.com/weiaicunzai/pytorch-cifar100 ,  the test accuracy of training ResNet-34 on CIFAR-100 using SGD-Momentum is almost 75 \%. The numerical results in the paper are insufficient (I think that the parameter setting would be insufficient in the paper).  Moreover, the authors' proposed methods would not be nice since the scores of the methods are less than 75 \%.

---

> > > ### Author Response · Authors · 2023-08-11
> > > **Q5. Numerical results for training DNNs**
> > >
> > > We kindly disagree. We do no use any weight decay (regularization) in our experiments. We report the performance of each method under the same setting with zero weight decay to show their effectiveness of minimizing the original loss function.

---

> > > > ### Comment · Reviewer_Dgvs · 2023-08-11
> > > > **Q5. Numerical results for training DNNs**
> > > >
> > > > I understand the authors' methods are applied to convex optimization in both theory and practice. However, I still would like to know whether the proposed methods can be applied to nonconvex optimization in ML. I would like to re-check the paper.  Thank you again for your replies.

---

> > > > > ### Author Response · Authors · 2023-08-11
> > > > > **Q5. Numerical results for training DNNs**
> > > > >
> > > > > Since our method has strong performance in numerical experiments on DNN, we believe this method can be extended to non-convex settings (since we have a similar preconditioner as AdaGrad). We leave this as a future work.
> > > > >
> > > > > Thank you for taking more time to read our paper. We would be happy to answer your further questions and concerns.

---

### Official Review · Reviewer_7mHQ · 2023-07-02

**Soundness:** 1 poor
**Presentation:** 4 excellent
**Contribution:** 2 fair
**Rating:** 4
**Confidence:** 3

**Summary:**

The paper aims to propose robust methods that achieves optimal rates in both
strongly-convex or convex and interpolation or non-interpolation settings.
Specifically, they propose AdaSPS, a modification of $SPS_{max}$ with
an AdaGrad-like denominator (replacing the gradient norms with function values) in
the stepsize, and AdaSLS, a combination of AdaGrad-Norm and line search
methods.

- For convex functions,
AdaSPS and AdaSLS achieve a $O(1/\epsilon^2)$ convergence rate assuming individual
smoothness and bounded domain.
When interpolation is assumed, both methods get rid of the bounded domain
assumption (but additionally require individual convexity and AdaSPS requires
the exact minimal function values) and achieve a $O(1/\epsilon)$ convergence
rate.
- For strongly-convex functions, individual strong-convexity and smoothness
are required. AdaSPS and AdaSLS achieve a $O(1/\epsilon^2)$ rate without interpolation.
With interpolation, they achieve linear convergence rates.
- Furthermore, the author combine the proposed methods with variance reduction techniques,
and improves the rate for strongly-convex and convex settings to $O(1/\epsilon)$
without interpolation.

The proposed methods are also evaluated by numerical experiments.

**Strengths:**

1. The paper has clear logic and is well-written, making it easy to follow.
2. In all considered settings (convex/strongly-convex and interpolation/non-interpolation),
the proposed AdaSPS and AdaSLS match the rate of well-tuned SGD in all settings
except for the case when we have strongly-convexity and non-interpolation.
Importantly, AdaSLS achieves this without requiring knowledge of any problem-dependent parameters.
3. The intuition behind the proposed variance reduction framework is interesting,
and could potentially motivate new algorithms.

**Weaknesses:**

  1. The related work on interpolation is somehow insufficient. One very
  relevant work [1] is missing. Importantly, [1] also proposed combining AdaGrad
  with line search, similar to AdaSLS, and provided theoretical guarantees in convex setting.
  The difference is that AdaSLS uses a minimum operator for all past stepsizes.
  Usually, non-diminishing stepsizes work better in practice, so it would also
  make sense to compare them in the experiments.
  2. I cannot find any theorems or proofs that show the results of AdaSVRPS/AdaSVRLS
  in strongly-convex and convex settings with interpolation,
  as listed in Table 1. This is the reason why I give
  a 1 in the soundness assessment.
  3. Without variance reduction, the paper claims that their algorithms match the convergence rate of SGD in
  many settings, however, the assumptions are different. Individual smoothness
  and individual (strongly-)convexity are additionally assumed in many theorems in this work,
  making it hard to compare them to classical results of SGD (also AdaGrad-Norm).
  I wonder if there are lower bounds for the individual (strongly-)convexity
  settings. With variance reduction, Theorem 8 also requires individual convexity,
  while the compared AdaSVRG does not.

  Minor issues:

  4. Table 1 could contain results of AdaSVRG and SARAH for a more comprehensive comparison
  of adaptive variance reduction methods. For the interpolation strongly-convex setting
  where we have linear convergence, the dependence on $\kappa$, the condition number,
  is also important.
  5. In Theorem 5, $T_p$ and $T_l$ are not defined. According to the results in
  the appendix, they depends on $\epsilon$ (of order $1/\log(1+\epsilon)$, or nearly $1/\epsilon$,
  if $c_p$ is chosen poorly).
  6. Usually the scalar version of AdaGrad is referred as AdaGrad-Norm. Calling
  it AdaNorm is not common, especially when there are other algorithms called AdaNorm.


  **References:**

  [1] Vaswani, Sharan, et al. "Adaptive gradient methods converge faster with over-parameterization (but you should do a line-search)." arXiv preprint arXiv:2006.06835 (2020).

**Questions:**

  I would appreciate it if the author could address the concerns listed
  as weaknesses 1-3.

Additionally, I wonder if we could change the added function value
  gaps (under the square root) in the denominator of AdaSPS stepsize to the
  squared norm of gradients (like AdaGrad) and obtain a similar result. Is the use
  of function value gap necessary?


**Limitations:**

  I think the main limitation is the sub-optimality in the non-interpolation
  strongly-convex setting, which is also a known hard problem for adaptive methods.
  This is metioned in the Conclusion and future work section of the paper.

---

> ### Author Rebuttal · Authors · 2023-08-09
>
> We thank the reviewer for the constructive remarks. We provide answers below. (For space limit, We omit the big O notation and write V-SGD for Vanilla SGD, AGN for AdaGrad-Norm, IC for individual convexity, ISC for individual st-convexity and PCV for potential camera-ready version).
>
> > W1. Paper on AdaGrad+line-search
>
> We apologize for missing this paper and will include it in our PCV. Since we study adaptive stepsize for SGD, let us discuss the scalar version of this algorithm.
>
> For AGN with constant stepsize, the best $\eta$ should be of order $\Theta(D)$ which gives the optimal rate of $LD^2/T+\sqrt{L}D\sigma/\sqrt{T}$. Note this method multiplies it by a scalar of order $1/L$ which results in a suboptimal rate $D^4L^3/T+D^2L^{3/2}\sigma/\sqrt{T}$ (See Theorem 2 in this paper and $\eta_{\max}$ is usually larger than $\frac{1}{L}$). In contrast, our carefully designed AdaSLS multiplies the inverse of the curvature $\gamma_t$ by the refined scaling term $1/c_l\sqrt{\sum_{s=0}^t\gamma_s||\nabla f_{i_s}(x_s)||^2}$ instead of $1/\sqrt{\sum_{s=0}^t||\nabla f_{i_s}(x_s)||^2}$ (this paper) which allows to obtain the optimal convergence rate.
>
> In the case of interpolation + IC, they can remove the constraint on $\eta_{t+1}\le\eta_t$ to improve practical performance. For AdaSLS, we can also remove the $\min$-operator, and the fast rate is still preserved (our proof does not require $\min$-operator in this case).  However. since we focus on the setting where the underlying interpolation condition is unknown, it is only fair to compare these methods with conservative constraints. This paper requires $\eta_t\le\eta_0$, and thus the algorithm is no better than a well-tuned AGN in practice. Since we have compared our stepsizes with the best-tuned AGN, we argue that it is not necessary to add this method to the experiments.
>
> > W2. VR, typos
>
> We thank the reviewer for finding these typos. For the VR methods, in all the four cases listed in Table 1, we will refill them with $\tilde{\mathcal{O}}(n+\frac{1}{\epsilon})$. Theorem 8 covers the convex and interpolation setting as well as the two st-convex settings.
>
> For the st-convex problems, it is a known hard problem to prove adaptive VR methods converge linearly. Since we do not claim any contribution of our new VR methods to the st convex setting and the word robustness is referred to the adaptive stepsizes, we really appreciate it if you consider raising your soundness score.
>
> > W3. Assumptions
>
> We thank for your careful discussion. We first agree V-SGD only requires smoothness of $f$. But individual smoothness assumption is also standard and can be satisfied in practice. Under this assumption, for all the cases, **the assumptions required for AdaSP/LS are at least the same or even weaker than any of the previous adaptive methods**.
> 1. $f$ is convex + interp. V-SGD with constant stepsize/AdaSP/LS/AGN converges as  $1/T$. SPS/SLS/DecSPS requires IC and DecSPS converges slowly.
> 2. $f$ is convex + non-interp. V-SGD with decreasing stepsize/AdaSP/LS/AGN converges as $\sigma/\sqrt{T}$. SPS/SLS cannot converge. DecSPS needs IC.
> 3. $f$ is convex + IC + interp. AdaSP/LS can remove bounded iterates assumption while no such result exists for AGN.
> 4. $f$ is st-convex + ITC + non-interp: AdaSP/LS/DecSPS can remove bounded iterates assumption while no such result exists for AGN.
> 5. $f$ is st-convex + interp: (In the attached pdf, we replaced ISC with the classical IC, and the original linear rate is preserved.) **In this case, the current adaptive methods require one more assumption than V-SGD**. V-SGD converges as $\exp(-\mu T/L)$. With additional IC, SPS/SLS converges as $\exp(-\mu T/L)$. AdaSP/LS also converges linearly, the constant of which depends on the first iterate and is usually worse than SPS/SLS (see attached). AdaGrad-Norm cannot converge linearly without knowing $L$.
>
> VR method + $f$ is convex: AdaSVRP/LS requires IC while SARAH/AdaSVRG do not. We will clearly state this requirement  in our PCV. However, since SPS/SLS exactly needs individual convexity in the interpolation setting. we think this assumption cannot be removed and we believe this is a weakness of these methods compared with AdaGrad. But in practice, the IC assumption is often satisfied.
>
> > Q1. Can we replace added function value gaps with the squared norm of gradients
>
> There are a few reasons why we cannot. 1) scaling issue. Suppose the exact Polyak stepsize is used. This quantity is of scale $1/L$. If we multiply it by AGN, then the scaling gives a suboptimal rate. 2) For convex problems, the error term $f_{i_t}(x_t)-f_{i_t}^\star$ cannot be upper bounded by $||\nabla f_{i_t}(x_t)||^2$ which might lead to divergence. 3) If we relax it to $\frac{f_{i_t}(x_t)-\ell_{i_t}^\star}{||\nabla f_{i_t}(x_t)||^2}$. Then the error caused by $\ell_{i_t}^\star$ cannot be compensated by the growing squared norm of the gradients at a correct rate. But it can be controlled by the accumulated $f_{i_t}-\ell_{i_t}^\star$ used in AdaSPS.
>
> > M1. A refined version of Table 1
>
> We thank for the suggestions. We will add the results and its relaxed assumption for AdaSVRG. For SARAH, since it uses a different gradient estimator and its stepsize is constant, we think it is a bit unfair to add it to Table 1 for adaptive stepsizes. For ISC problem, we agree that the constant is important and we will add it for clarity.
>
> > M2. $T_p$ and $T_l$
>
> We agree that if $c_p$/$c_l$ is chosen poorly, the convergence can be sublinear. Therefore, we recommend the theoretically suggested $c_p$/$c_l$ indicated in the Theorem to avoid the potential slowdown. These values only depend on the first iterate. In other words, they are parameter-free.
>
> > M3. A correct name: AdaGrad-Norm
>
> We will do that, thanks.
>
> We thank you again for the great reviews. If you agree we addressed the main concerns, please consider raising your mark. If you believe not, please let us know. We really appreciate that.

---

> > ### Comment · Reviewer_7mHQ · 2023-08-15
> >
> > Thanks for the detailed explanation.
> >
> > If AdaSVRPS/AdaSVRLS can only achieve $\widetilde{\mathcal{O}}(n + \frac{1}{\epsilon})$
> > for all settings, then they seem less compelling. The author claimed that
> > "This is a significant contribution, as trivial combinations of existing
> > variance-reduction techniques with Polyak stepsizes or line-search does not
> > work." Why is it interesting to bring line search or Polyak stepsize to variance-reduction
> > in the first place if the rates are no better than existing methods?
> > Also it seems a bit unconventional to consider variance-reduction and interpolation at the same time,
> > given there is no benefit in improving the rate when compared to considering them separately.
> >
> > In addition, I noticed that in Proposition 1, to achieve the claimed goal of no requirement for unknown parameters,
> > both $c_l$ and $c_p$ should be set according to the initial stochastic gradient or function value.
> > However, in the proof of Theorem 1, when taking expectations, they are treated as constants.
> > I am not sure whether the proof remains valid when they are random variables. This could also
> > be an issue for the theorem presented in the rebuttal PDF.

---

> > > ### Author Response · Authors · 2023-08-16
> > > **Answers to the first paragraph**
> > >
> > > Thanks for the great remarks.
> > >
> > > > If AdaSVRPS/AdaSVRLS can only achieve $\tilde{O}(n+\frac{1}{\epsilon})$ for all settings, then they seem less compelling.
> > >
> > > 1. We thank for the comment. The word "only" seems a bit strong. There are **no** counter examples that show AdaSVRPS/AdaSVRLS **cannot** achieve linear convergence in the strongly-convex settings. In numerical experiments, they show such a linear rate  (for instance, see the second plot in Figure 1). However, the proof itself is a known hard problem due to the many bottlenecks in the current proof framework. That being said, the VR methods themselves including AdaSVRG/AdaSVRLS/AdaSVRPS might still be able to accelerate in this case. We leave this as a future work.
> > >
> > > 2. The only current adaptive VR method is AdaSVRG. First, three methods all show competitive performance in practice. Second, note that AdaSVRPS/AdaSVRLS has freedom to decide how often to update the full gradient depending on the computational power so that convergence might be faster in practice. This can be done by choosing different $p_t$ ($p_t$=1 reduces to GD). AdaSVRG needs to carefully determine the number of stages and the inner-outer-loop size to guarantee convergence. Therefore, AdaSVRPS is no worse than AdaSVRG.
> > >
> > > >  Why are these two methods are still interesting if the rates are no better than existing methods?
> > >
> > > 1. Bringing line-search to VR was an interesting open question in the last decade. Schmidt
> > > et al. [1] and Mairal [2] provide promising empirical results by setting the stepsize in VR using line-search. However, the theoretical guarantees were elusive. Dubois-Taine et al. [3] first provide a counter-example that shows an intuitive line-search method with VR fails to converge, which brings less hope to this method. However, we show that in fact,
> > > doing line-search on the individual biased function $f_{i_t}$ provides misleading curvature information, which makes the classical VR method fail to work. We address this issue by adding a correction term that contains global information to $f_{i_t}$ and then doing line-search on the variance-reduced $F_{i_t}$. This approach breaks the previous limitation and bias, which might encourage faster and better VR methods with line-search and Polyak-stepsize.
> > >
> > > 2. The proposed VR framework is general. Apart from Polyak-stepsize and line-search, one can also apply AdaGrad stepsize to the functions $F_{i_t}$ and the resulting algorithm enjoys the same convergence guarantee. The same approach can be applied to the stepsizes proposed by Malitsky et al.[4],  Lvgi et al.[5], etc. Therefore, it would be interesting and promising to use this framework to develop better and faster VR algorithms (for instance, applying second-order methods or momentum on $F_{i_t}$).
> > >
> > > > Consider variance-reduction and interpolation at the same time
> > >
> > > We apologize for the confusion in Table 1. The separation is used for the adaptive stepsizes for SGD and we do not aim to consider VR and interpolation at the same time. We will remove the VR methods from Table 1 to make it extra clear.
> > >
> > > [1] Minimizing finite sums with the stochastic average gradient. Mathematical Programming
> > >
> > > [2] Optimization with first-order surrogate functions. ICML 2013
> > >
> > > [3] SVRG meets AdaGrad: Painless Variance Reduction, Machine Learning 2022
> > >
> > > [4] Adaptive Gradient Descent without Descent, ICML 2020
> > >
> > > [5] DoG is SGD's Best Friend: A Parameter-Free Dynamic Step Size Schedule, ICML 2023

---

> > > ### Author Response · Authors · 2023-08-16
> > > **Answers to the second paragraph**
> > >
> > > > $c_p$ and $c_l$ depending on the first stochastic information that might make the proof invalid.
> > >
> > > We appreciate the great observations. **We apologize that we did make a mistake in the last step of the proof. But this can be easily fixed.** We take AdaSPS as an example.
> > >
> > > Under individual convexity and interpolation assumptions, we get equation (7) from the rubuttal pdf.
> > >
> > > $||x_{t+1}-x^\star||^2\le||x_t-x^\star||^2-\frac{1}{(2c_pL||x_0-x^\star||)^2}\nabla f_{i_t}(x_t)^T(x_t-x^\star)$
> > >
> > > Denote $\frac{1}{(2c_pL||x_0-x^\star||)^2}$ by $A$ and plug in $c_p=\frac{c_p^{scale}}{\sqrt{f_{i_0}(x_0)-f^\star}}$ where $c_p^{scale}\ge 1$ is a fixed constant. Since the right hand side depend on the inner product of two RVs, we need carefully take the expectations.
> > >
> > > > **$f$ is convex.**
> > > For any $t\ge 1$, we can take expectation conditional on $i_0$ on both sides and get
> > > $E[||x_{t+1}-x^\star||^2|i_0]\le E[||x_t-x^\star||^2|i_0]-AE[\nabla f(x_t)^T(x_t-x^\star)|i_0]$.
> > > Using convexity and summing up from $1$ to $T$, we get
> > > $\sum_{t=1}^TE[f(x_t)-f^\star|i_0]\le\frac{1}{A}||x_1-x^\star||^2\le\frac{1}{A}||x_0-x^\star||^2$.
> > > Take expectation w.r.t $i_0$ on both sides and dividing by $T$, we get
> > > $\frac{1}{T}\sum_{t=1}^T E[f(x_t)-f^\star]\le 4L(c_p^{scale})^2E_{i_0}[\frac{||x_0-x^\star||^2}{(f_{i_0}(x_0)-f^\star)}]\frac{L||x_0-x^\star||^2}{T}$.
> > >
> > > > **$f$ is st-convex.**
> > > For any $t\ge 1$, we have
> > > $E[||x_{t+1}-x^\star||^2|i_0]\le(1-\mu A)E[||x_{t}-x^\star||^2|i_0]$.
> > > Unrolling, for any $T\ge 1$, we get
> > > $E[||x_{T+1}-x^\star||^2|i_0]\le(1-\mu A)^T||x_1-x^\star||^2\le(1-\mu A)^T||x_0-x^\star||^2$.
> > > Take expectation w.r.t. $i_0$, we get
> > > $E[||x_{T+1}-x^\star||^2]\le E_{i_0}[(1-\mu A)^T]||x_0-x^\star||^2$.
> > >
> > > (Note another way to handle this issue is to first use the bound: $c_p\le\frac{c_p^{scale}}{\sqrt{\min_{i_0} f_{i_0}(x_0)-f^\star}}$ before taking the expectation. But this is too pessimistic.)
> > >
> > > We will make it clear in the manuscript.
> > >
> > > We thank again for the great remarks. We hope our answer addresses your concerns.

---

> > > > ### Comment · Reviewer_7mHQ · 2023-08-21
> > > >
> > > > Thank you for your response. After consideration, I have decided to maintain my current score.

---

### Official Review · Reviewer_a9xP · 2023-07-07

**Soundness:** 3 good
**Presentation:** 3 good
**Contribution:** 2 fair
**Rating:** 4
**Confidence:** 4

**Summary:**

This paper introduces Adagrad-norm type update into stochastic line search (SLS) and stochastic Polyak step size (SPS) (namely AdaSLS and AdaSPS) that guarantee convergence in non-interpolating convex scenario with a $\mathcal{O}(\frac{1}{\epsilon^2})$ rate. Then it shows that AdaSLS and AdaSPS  converge linearly when interpolation holds under strong convexity. Finally, it combines variance reduction with AdaSLS/AdaSPS to improve the convergence to $\mathcal{O}(n+\frac{1}{\epsilon})$ for convex losses.

**Strengths:**

1. Overall, the paper is well-written. The comparisons with existing literature results are thorough.

2. The introduction of loopless variance reduction, together with AdaGrad and SPS/SLS into a single framework is novel.

**Weaknesses:**

1. There is no improvement in the convergence rates. DecSPS was introduced to make SPS converge in non-interpolating settings by forcing the step size to be monotonic-decreasing. The proposed AdaSPS essentially replaces the sequence $c_k$ in DecSPS [1] with Adagrad-Norm, and the step size is also monotonically decreasing, resulting in the same rate as DecSPS. In the interpolating settings, SLS  [2] and SPS [3] already converge without the step size being monotonic decreasing. I don’t see the improvements in AdaSPS/AdaSLS over previous methods under interpolation or non-interpolation.

2. In the strongly-convex interpolating case, AdaSPS/AdaSLS converges linearly with some requirements given in Corollary 6. SPS and SLS have the same convergence rate without those additional requirements. In fact, the constants $c_p$ and $c_l$ associated with AdaSPS and AdaSLS respectively depend on the sample at first iteration. I don’t see how this is always satisfied. Does it mean that $c_p$  and $c_l$ need to change every time when running the algorithm? In addition to this, AdaSPS/AdaSLS assumes that each function is strongly convex, this is a very strong assumption that does not appear in the analysis of SPS and SLS. Does it mean only optimizing one function is sufficient as they all share the same unique global optimum under interpolation?

3. The experiments are not promising. Figure 1 just shows that AdaSPS performs worse than SPS under interpolation, and performs worse than or similarly to DecSPS under non-interpolation. If there already exists some better alternatives in either setting, what is the benefit of using AdaSPS?

**Questions:**

I vote for rejection due to the following reasons:

1. No improvements in rates over previous works in either interpolation or non-interpolation scenario.

2. Weak empirical results. The gain in the proposed method is insignificant.

3. The analysis requires some strong assumptions and additional requirements to obtain the same rate as previous works.

[1] Dynamics of SGD with Stochastic Polyak Stepsizes: Truly Adaptive Variants and Convergence to Exact Solution

[2] Painless Stochastic Gradient: Interpolation, Line-Search, and Convergence Rates

[3] Stochastic Polyak Step-size for SGD: An Adaptive Learning Rate for Fast Convergence

---

> ### Author Rebuttal · Authors · 2023-08-09
>
> We thank the reviewer for the remarks and criticisms. Before responding to each point, we kindly want to highlight our first contribution to address your main concern.
>
> **We focus on the settings where the underlying interpolation condition is unknown to the users.** Having a **robust** (that can adapt to the interpolation condition) and theoretically-grounded algorithm is important in many cases such as in the federated learning setting as mentioned in the introduction part. **Furthermore, it is not easy to determine whether or not a model is effectively interpolating the given dataset.** Consider the rcv1 dataset where the dimension of features are twice larger than the number of data points. A logistic regression model is considered overparameterized. But the features are actually sparse and it is not interpolated.  Therefore employing SPS/SLS cannot guarantee convergence. **Developing a robust and reliable algorithm will be of great convenience for the users in practice.**
>
> As summarized in Table 1, our newly proposed AdaSPS/AdaSLS are the **first** adaptive stepsizes to have such strong **robust** theoretical guarantees in all cases without knowledge of the Lipschitz constant.
>
> We now provide answers to each point.
>
> >W1. There is no improvement in the convergence rates.
>
> We kindly disagree. The previous adaptive methods are the best on a certain range class of problems, while our method achieves the best known rates in all these scenarios. From Table 1, AdaSPS/AdaSLS achieve both fast convergence rates in the interpolation settings like SPS/SLS and in non-interpolated settings like DecSPS. **These asymptotic rates are already optimal and cannot be further improved.** (except for strong convexity without interpolation). The denominator defined in AdaSPS/AdaSLS is essentially the key to having such an adaptivity to the underlying interpolation condition. Note that DecSPS artificially incorporates $\mathcal{O}(1/\sqrt{t})$ decreasing rule which results in slow convergence with interpolation while the denominator designed for AdaSPS/AdaSLS actually can be upper bounded by a constant if interpolation holds and will go to infinity if not.
>
> > W2. Assumptions
>
> We thank the reviewer for the comments. In the attached pdf, we replaced the individual strong-convexity assumption in Theorem 5 with the classical individual convexity. The original linear rate complexity is preserved.
>
> We kindly disagree with the comment “The analysis requires some strong assumptions and additional requirements to obtain the same rate as previous works.” **In all the cases, our assumptions are at least the same or even weaker than any of the previous adaptive methods!** .
> 1. Convex + interpolation: SPS/SLS/DecSPS assumes individual convexity while ours and AdaGrad-Norm only assume $f$ is convex (See Theorem 1 with noise = 0).
> 2. Convex + interpolation + individual convexity: we can further remove the bounded iterates assumption while AdaGrad-Norm cannot.
> 3. Convex + non-interpolation: DecSPS assumes individual convexity while ours and AdaGrad-Norm only assume $f$ is convex.
> 4. st-convex + non-interpolation + individual st-convexity: ours and DecSPS remove the bounded iterates assumption while AdaGrad-Norm cannot.
> 5. st-convex + interpolation: SPS/SLS and ours all assume individual convexity. While AdaGrad-Norm cannot show linear convergence without knowledge of $L$. DecSPS only has $\mathcal{O}(1/\epsilon^2)$.
>
> This shows again the **strong robustness and the weakest assumptions required** for our stepsizes.
>
> > W3. How to set $c_p$/$c_l$
>
> As illustrated in the numerical evaluation section, we used the theoretically justified hyperparamter $c_p^{\text{scale}}$ which essentially only depends on the first iterate. For instance, let us fix $c_p^{\text{scale}}=1$. Then $c_p=\frac{1}{2\sqrt{f_{i_0}(x_0)-\ell_{i_0}^\star}}$ and it will not change during the iterations. This quantity provides the right scaling correction to the stepsize and brings much convenience to the user experience since they do not need extra tuning.
>
> > W4. The experiments are not promising?
>
> We kindly disagree: Our synthetic experiments are designed to illustrate the **robustness** of our proposed algorithms. Let us compare these algorithms closely.
>
> We first agree that AdaSPS is not as competitive as SPS with interpolation since AdaSPS is a non-increasing stepsize. However, AdaSPS shows the desired linear and sublinear convergence for st-convex and convex problems while DecSPS converges much more slowly. In the non-interpolation regimes, SPS **cannot** converge and has a big neighborhood error. **AdaSPS outperforms DecSPS in all the experiments** (See Figure 1 and 2) **only except** the second plot in Figure 1 which we can apparently choose a larger $c_p$ to improve its performance. (Note we fix the same $c_p^{\text{scale}}=1$ across these experiments to show the robustness).  As such, AdaSPS can be seen as a direct replacement for DecSPS (both in theory and in practice).
>
> If we are certain that the underlying problem is interpolated and we know the exact optimal function value, then we definitely recommend using SPS. Otherwise, AdaSPS/AdaSLS is always reliable and offers a better choice.
>
> We would like to highlight another importance of introducing AdaSPS/AdaSLS. SPS/SLS/DecSPS all fail to be incorporated into the VR framework because they are not robust. Our work gives a very promising direction that even these difficult stepsizes can also be combined with VR techniques as long as we can 'make them robust'. This may motivate more personalized VR techniques in the future.
>
> We believe that our newly proposed stepsizes and the novel VR framework are good contributions to the community for many potential extensions. If you agree that we managed to address all issues, please consider raising your mark. If you believe this is not the case, please let us know so that we have a chance to respond. We really appreciate that.

---

> ### Comment · Reviewer_a9xP · 2023-08-21
> **Reply**
>
> Thanks for the rebuttal. I have raised my score to 4. But I am not fully convinced by the novelty and significance of this work.

---

### Official Review · Reviewer_NwRJ · 2023-07-07

**Soundness:** 2 fair
**Presentation:** 2 fair
**Contribution:** 3 good
**Rating:** 5
**Confidence:** 4

**Summary:**

This paper proposes two new variants of SPS and SLS, called AdaSPS and AdaSLS, which provide convergence in non-interpolation settings for convex and strongly convex functions when training over-parameterized models. AdaSLS requires no knowledge of problem-dependent parameters, and AdaSPS requires a lower bound of the optimal function value as input. In addition, the paper studies a new variance reduction technique for AdaSPS and AdaSLS and proves the gradient complexity for convex functions, which improves upon the rates of AdaSPS and AdaSLS without variance reduction in the non-interpolation settings. This matches the fast rates of AdaSVRG and SARAH without the inner-outer-loop structure, which is easier to implement and analyze. The authors provides numerical experiments on synthetic and binary classification on LIBSVM datasets.

**Strengths:**

The two algorithms AdaSPS and AdaSLS are interesting. The authors provide the convergence analysis for them, especially in non-interpolation setting the algorithms attained the classical convergence rate for convex functions. The variance reduction technique has the same complexity in expectation for convex functions as the rate of AdaSVRG and SARAH.

**Weaknesses:**

The theoretical rate has comparison with SARAH but it is not included in the experiment. The variance technique and the one loop structure is not entirely new since it already appeared in the PAGE algorithm. The authors may consider discussing that as well. A natural question is why practitioners should use the new variance reduction algorithms while it is more complicated than the well established methods in convex settings. Thus more extensive experiments would help to demonstrate the effectiveness of the algorithms.

**Questions:**

Since the algorithm requires more parameter than classical methods, could you explain how they are chosen in the experiments/ suggested in practical settings? e.g. $\mu_F$, $c_p$, $c_l$, $\gamma_t$ , $p_t$?

How the lower bound information of 0 may affect the performance of the algorithm (theoretically and empirically), since the theory uses the correct lower optimal values.

---
I thank the authors for your rebuttal. My recommendation remains the same.

---

> ### Author Rebuttal · Authors · 2023-08-09
>
> We thank the reviewer for the remarks and suggestions on our paper. You can find our replies below:
>
> > Comparison with SARAH and PAGE
>
> We thank the reviewer for mentioning PAGE. We will include the discussion on PAGE in the potential camera-ready version. We would like to highlight that **there is no direct connection of our methods to SARAH and PAGE.** We classify SVRG, SARAH, and PAGE into the first group, as they propose different gradient estimators for variance reduction. To guarantee convergence, the knowledge of $L$ is required to set their constant stepsizes. Conversely, AdaSVRG/AdaSVRPS/AdaSVRLS belongs to a different group where they fix the gradient estimator first (for instance, they all use SVRG) and the focus is to set the stepsize adaptively without the need to know problem-dependent parameters to guarantee convergence. **Therefore, the works in these two groups are orthogonal.**
>
> > Technical novelty
>
> We do not claim that the one loop structure and probabilistic update of the full gradient is new since it has already been discussed in Dmitry [19] and PAGE as you mentioned. However, our proposed VR framework based on the moving sequence of random functions is new as it allows some difficult adaptive stepsizes such as Polyak and line-search type algorithms to accelerate. With the common VR technique, these stepsizes fail to converge. Another novelty is that with our decreasing probability strategy, the inner-outer-loop structure used in AdaSVRG can be removed and this strategy is the key to providing the optimal convergence rate in the convex setting. In the other works on loopless VR technique, the probability is set to be a constant.
>
> > SARAH experiment
>
> We thank the comment on SARAH and we have included it in the attached pdf under the same experimental setting as in Section 5. But as we mentioned earlier, AdaSVRPS/AdaSVRLS/AdaSVRG/SVRG all use the same gradient estimator but with different stepsizes. Consequently, we think it is fair to compare them in the experiments. Comparison of different gradient estimators is a bit orthogonal to the focus of this work.
>
> > How to set hyper-parameters for AdaSVRPS/AdaSVRLS in practice
>
> For both AdaSVRPS and AdaSVRLS, the key parameters are $\mu_F$ and $c_p^{\text{scale}}/c_l^{\text{scale}}$.
>
> Note that the inverse of the curvature of the random function $F_{i_t}$ is upper bounded by $\mathcal{O}(\frac{1}{\mu_F})$ ($F_{i_t}$ is at least $\mu_F$-st covex). Therefore, the smaller the $\mu_F$, the larger the maximum value of the stepsize could reach. In practice, we can set it to be $10^{-4}$ for a potential aggressive stepsize in the case where $d>n$. Otherwise, we can set it to be $1$ for a conservative stepsize.
>
> $c_p^{\text{scale}}/c_l^{\text{scale}}$ controls the scale of the Polyak stepsize/line-search stepsize. In other words, the adaptive stepsize is upper bounded by the standard  Polyak stepsize/line-search stepsize multiplied by the inverse of $c_p^{\text{scale}}/c_l^{\text{scale}}$. A reasonable choice would simply be $c_p^{\text{scale}}/c_l^{\text{scale}}\in[0.5,1,2]$. The smaller the number, the more aggressive the stepsize.
>
> For the parameters inside the line-search method, we always  fix $\gamma_{\max}=10^3$ or $\frac{1}{\mu_F}$, $\beta=0.8$ and $\rho=1/2$.
>
> The last parameter $p_t=\frac{1}{at+1}$ is very flexible. The smaller $a$, the more frequent of computation of the full gradient. A standard choice would be $a=0.1$. However, depending on the computational power, one can freely choose this number. In constrast, the inner-outer-loop structure of AdaSVRG does not allow arbitrary full gradient update frequency. Indeed, AdaSVRG fails to converge with inner-loop size being one and fixing $g_t = \nabla f(x_t)$.
>
> All the experimental details can be found in Appendix F.
>
> > Impact of  the lower bound of zero for AdaSVRPS in practice
>
> Yes, for AdaSVRPS, we use the exact $F_{i_t}^\star$ in theory. If we replace it with $\ell_{i_t}^\star$ (a lower bound of $F_{i_t}$), then an additional slow down term $\mathcal{O}(\frac{\sigma}{\sqrt{T}})$ will occur in theory. The proof can follow the routine for AdaSPS. However, in practice, it suffices to use $\ell_{i_t}^\star+\min_x\{x^T(\nabla f(w_t)-\nabla f_{i_t}(w_t))+\frac{\mu_F}{2}||x-x_t||^2\}$ where $\ell_{i_t}^\star$ is a lower bound for $f_{i_t}^\star$ which is zero normally. Note that we use this lower bound for all our experiments and AdaSVRPS always shows competitive performance compared with a well tuned SVRG. Therefore, we believe using a lower bound has no impact in practice and sometimes can even make the algorithm behave more aggressively (since the stepsize is larger than needed).
>
> > Practical consideration
>
> The previous well-established VR methods including SVRG/SARAH/PAGE requires the knowledge of Lipschitz constant to guarantee convergence. AdaSVRG needs to predefine the target accuracy $\epsilon$ to design the number of stages and the inner-outer-loop size. Also, the arbitrary full gradient update frequency is not supported. While AdaSVRPS/AdaSVRLS requires more hyper-parameters, one can simply set $\mu_F=1$ and $c_p^{\text{scale}}/c_l^{\text{scale}}=1$, and the algorithms can reliably converge at a correct rate of $\mathcal{O}(1/T)$. The user also has the freedom to adjust these parameters to gain more aggressive practical performance.
> For the two adaptive stepsizes, AdaSPS/AdaSLS brings much convenience to the user experience since they do
> not need extra tuning of the stepsize (simply setting $c_p^{\text{scale}}/c_l^{\text{scale}}=1$ is enough for robust convergence).
>
> We believe that our newly proposed stepsizes and the novel VR framework are important contributions to the community as this may motivate more personalized VR techniques in the future. If you agree that we managed to address all issues, please consider raising your mark. If you believe this is not the case, please let us know so that we have a chance to respond. We really appreciate that.

---

### Author Rebuttal · Authors · 2023-08-09

Thanks to all reviewers for examining our manuscript and help with improving our paper. We appreciate the constructive comments from the reviewers, and we address all raised issues via individual comments.

We like to highlight that:
- **We propose the first adaptive methods that simultaneously achieve optimal asymptotic rates in both strongly-convex or convex and interpolation or non-interpolation settings. This is a significant contribution, as it is not easy to determine whether or not the interpolation condition holds for a given problem (without solving it).**

    We work on smooth and convex optimization. We focus on the practical scenario where the underlying interpolation condition of the considered problem is unknown to the users. We propose two new robust stepsizes based on Polyak stepsize and line-search. These are the first adaptive methods that simultaneously achieve optimal asymptotic rates in both strongly-convex or convex and interpolation or non-interpolation settings (except for the case when we have strongly-convexity and non-interpolation), without requiring knowledge of the Lipschitz constant. Furthermore, AdaSPS only needs a lower bound of the optimal function value and AdaSLS is completely parameter-free. Under the  standard individual smoothness condition, the assumptions required are the weakest compared with all the previous adaptive methods. We provide theoretically suggested hyper-parameters for these stepsizes, which makes it even more convenient and reliable for the users to apply in practice. Moreover, their competitive performance in deep learning experiments show strong potential for non-convex optimization as well.

- **We propose the first variance-reduced methods with Polyak stepsizes or line-search. This is a significant contribution, as trivial combinations of existing variance-reduction techniques with Polyak stepsizes or line-search does not work (see the lower bound proven in [8] for line search, and appendix E for SPS).**

   Polyak and line-search type methods cannot converge with the classical variance-reduction framework. In this work, we successfully break this long-time barrier and manage to accelerate these two complicated stepsizes. We managed to do that by first proposing a novel variance-reduction framework based on the random proxy function sequence and then applying our robust stepsizes to the new proxy function. We prove the optimal rate of our new algorithms for convex problems. We also introduce the new decreasing probability strategy which allows to remove the classical inner-outer-loop structure and make the proof much easier. Numerical experiments show their strong performance in practice.

Reviewer NwRJ found our two algorithms AdaSPS and AdaSLS are interesting. Reviewer a9xP thinks our variance-reduction framework combined with adaptive stepsizes is novel. Reviewer 7mHQ correctly recognizes that our new robust stepsizes have the strong theoretical guarantees and supports AdaSLS as' Importantly, AdaSLS achieves all this without requiring knowledge of any problem-dependent parameters.' Reviewer 7mHQ further agrees that 'the intuition behind the proposed variance reduction framework is interesting, and could potentially motivate new algorithms.' Reviewer Dgvs thinks using these stepsizes and combining them with acceleration can be useful in practice.

Based on the questions of the reviewers, we have made the following major updates.
1) We replaced the individual strong-convexity assumption in Theorem 5 with the classical individual convexity and prove the same linear convergence rate.
2) We added the comparison of SARAH in experiments.
3) We fixed the typos of the last two lines within the interpolation columns in the Table 1.

To summarize, in this work, we propose two new adaptive stepsizes that enjoy robust convergent guarantees, which serves as a good step towards a totally automatic adaptive algorithm. Our novel VR framework is general and may motivate more personalized variance-reduction techniques in the future. We believe both of them are important contributions to the community.

We anticipate an interactive discussion with you, and we will be most happy to answer any remaining questions.

---

> ### Comment · Reviewer_7mHQ · 2023-08-15
>
> Thanks for the summary. I wonder why the author claims, "We propose the first
> adaptive methods that simultaneously achieve optimal asymptotic rates in both
> strongly-convex or convex and interpolation or non-interpolation settings." For
> AdaSPS/AdaSLS, the rate under strongly-convex non-interpolation setting is not
> optimal. For AdaSVRPS/AdaSVRLS, the rate is non-optimal for all strongly-convex
> cases.

---

> > ### Author Response · Authors · 2023-08-16
> >
> > We thank the reviewer for the remark. On the fifth line of the first paragraph with a bullet point, we mentioned in the bracket that our two stepsizes are not optimal in the case when we have strongly-convexity and non-interpolation. We apologize for the confusion. We make the claim clear here: We propose the first adaptive methods that simultaneously achieve optimal asymptotic rates in both strongly-convex or convex and interpolation or non-interpolation settings **except for the case when we have strongly-convexity and non-interpolation.**

---

### Decision · Program_Chairs · 2023-09-21

**Decision:**

Accept (poster)

**Comment:**

Although the reviewers are on average negative, I feel that achieving fast interpolation and non-interpolation rates is an important contribution and I feel that the authors did a reasonable job responding to the criticisms brought up in the reviews. As most of the reviewers did not engage with these rebuttals, I am leaning towards saying that the criticisms were addressed and the paper should be accepted.

Please make sure that the following changes are made (in addition to other comments brought up in the review):
- Fix the table and downplay the claims about the variance-reduced versions. (I would even be fine if you removed the variance-reduced versions from the paper, I feel like they make it more confusing and given the weaker result they downplay interestingness of AdaSLS and AdaSPS.)
- Report numerical results in the main paper where you choose one fixed set of hyper-parameters across datasets. I considered for a long time whether to reject the paper just on this point - truly adaptive methods should not need hyper-parameter tuning for each dataset.
- I feel like a numerical comparison involving SLS and AdaSLS should appear in the main paper, this seems like a weird omission. It probably also makes sense to include the similar "painless SVRG" method.